# Rational $Q$-systems, Higgsing and Mirror Symmetry

**Jie Gu, Yunfeng Jiang, Marcus Sperling**

*School of Physics and Shing-Tung Yau Center, Southeast University, Nanjing, Jiangsu, 210096, China*

ABSTRACT: The rational $Q$-system is an efficient method to solve Bethe ansatz equations for quantum integrable spin chains. We construct the rational $Q$-systems for generic Bethe ansatz equations described by an $A_{\ell-1}$ quiver, which include models with multiple momentum carrying nodes, generic inhomogeneities, generic diagonal twists and $q$-deformation. The rational $Q$-system thus constructed is specified by two partitions. Under Bethe/Gauge correspondence, the rational $Q$-system is in a one-to-one correspondence with a 3d $\mathcal{N} = 4$ quiver gauge theory of the type $T_{\boldsymbol{\rho}}^{\boldsymbol{\sigma}}[\mathrm{SU}(n)]$, which is also specified by the same partitions. This shows that the rational $Q$-system is a natural language for the Bethe/Gauge correspondence, because known features of the $T_{\boldsymbol{\rho}}^{\boldsymbol{\sigma}}[\mathrm{SU}(n)]$ theories readily translate. For instance, we show that the Higgs and Coulomb branch Higgsing correspond to modifying one of the partitions in the rational $Q$-system while keeping the other untouched. Similarly, mirror symmetry is realized in terms of the rational $Q$-system by simply swapping the two partitions - exactly as for $T_{\boldsymbol{\rho}}^{\boldsymbol{\sigma}}[\mathrm{SU}(n)]$. We exemplify the computational efficiency of the rational $Q$-system by evaluating topologically twisted indices for 3d $\mathcal{N} = 4$ U$(n)$ SQCD theories with $n = 1, \ldots, 5$.

## 1 Introduction

Solving Bethe ansatz equations (BAE) is a fundamental and important question in integrability. The solutions of BAE encode the rich structure of the model and are related to the completeness problem of the Bethe ansatz. Therefore, they are of great mathematical interest (see for example [1–6]). Equally important, finding all physical solutions of the BAE is an essential step in computing many physical quantities, either numerically by solving the BAE by numerical approaches or analytically by exploiting the recently developed computational algebraic geometry method [7–11]. Due to the wide applicability of the Bethe ansatz, ranging from statistical mechanics to high energy physics, developing efficient methods for solving BAE is obviously welcome and of great practical value.

However, working directly with BAE has a number of drawbacks such as the generation of non-physical solutions and numerical instability. Therefore, alternative formulations of BAE which are easier to handle have long been sought for. The two most important formulations are the $TQ$ and the $QQ$-relations. The $TQ$-relation stems from Baxter's method of solving integrable ice-type lattice models including the famous six- and eight-vertex models [12]. The idea is to construct an operator $Q$ which commutes with the quantum transfer matrix $T$ and satisfies a specific finite difference equation, called the $TQ$-relation. Working with eigenvalues of both operators, one gets a finite difference equation for Baxter's $Q$-function, whose zeros are the solutions of the BAE. Therefore, one can first solve the $TQ$-relation to find the $Q$-functions and then determine the zeros of the $Q$-functions. This turns out to be more efficient then directly solving BAE, and eliminates part of the non-physical solutions such as the ones with repeated roots.

Baxter's $TQ$-relation is a second order difference equation for the $Q$-function. Therefore it allows two solutions. In addition, the two $Q$-functions satisfy the Wronskian condition, which is called the $QQ$-relation. It turns out that one can solve the $QQ$-relation directly and find both $Q$-functions simultaneously. One then takes the zeros of one of the $Q$-functions, which gives the solution of BAE. In [13], Marboe and Volin proposed an ingenious rewriting of the $QQ$-relation by defining a $Q$-system on a Young tableaux.

This method leads to only physical solutions (*i.e.* all non-physical solutions are automatically eliminated) and is much more efficient to solve compared to the original BAE or $TQ$-relation. It is by far the most efficient approach to find all the physical solutions of the BAE, at least for the rational spin chains with periodic boundary conditions. In order to distinguish the Marboe-Volin $Q$-system, which are defined on a Young tableaux, and the traditional $QQ$-system, which are Wronskian conditions for higher rank $TQ$-relations, we call the former the *rational Q-system*. This method is reviewed in Section 2.

In the original work [13], the authors gave the rational $Q$-system formulation for a $GL(M|N)$ invariant XXX-type spin chain with periodic boundary conditions. Later it has been extended to the non-compact $GL(M, N|L)$ invariant XXX-type spin chains in [14]. Generalizations to XXZ-type spin chain with different boundary conditions (open, twisted) have been investigated in [11, 15–17]. One of the aims of the current work is to take a further step and generalize the formulation of rational $Q$-system for the BAE associated to a generic $A$-type Dynkin diagram, for both XXX- and XXZ-type models with multiple momentum carrying nodes, general inhomogeneities and twists.

In addition, we uncover a beautiful relationship between the rational $Q$-system and supersymmetric 3d $\mathcal{N} = 4$ quiver gauge theories of type $T_{\boldsymbol{\rho}}^{\boldsymbol{\sigma}}[\mathrm{SU}(n)]$. Via the Bethe/gauge correspondence [18–20], the supersymmetric vacua of such theories compactified on $S^1$ are precisely the solutions of the Bethe Ansatz equations. This correspondence builds a one-to-one map between quantities in gauge theory and in the spin chain, which has been studied extensively in the literature [21–25]. We revisit this correspondence from the rational $Q$-system point of view. It turns out that rational $Q$-system seems to be an even more natural formulation than BAE for the Bethe/gauge correspondence. For example, the origin of the Young tableaux, on which the rational $Q$-system is defined, might seem a bit mysterious from the spin chain point of view. On the other hand, it is quite natural in the quiver gauge theory and its brane realisation in Type-IIB superstring theory. The theories $T_{\boldsymbol{\rho}}^{\boldsymbol{\sigma}}[\mathrm{SU}(n)]$ are specified by two partitions $\boldsymbol{\rho}$ and $\boldsymbol{\sigma}$ [26]. It turns out that one of the partitions $\boldsymbol{\rho}$ corresponds precisely to the Young tableaux of the rational $Q$-system. What about the other partition $\boldsymbol{\sigma}$? It also plays an important role in the rational $Q$-system. As we shall explain later, to specify a rational $Q$-system, we need a Young tableaux and also fix boundary conditions. The boundary conditions are encoded by another partition, given precisely by $\boldsymbol{\sigma}^{\mathrm{T}}$, the transposition of $\boldsymbol{\sigma}$.

Important gauge theory phenomena such as Higgsing and mirror symmetry are also reflected nicely in the rational $Q$-system. There are two kinds of partial Higgs mechanisms, *i.e.* Higgs branch Higgsing and Coulomb branch Higgsing. The former corresponds to an operation of the $Q$-system which maintains the shape of the Young tableaux while changing the boundary condition while the latter corresponds to the $Q$-system which preserves the boundary condition while re-arranging the boxes of Young tableaux. Mirror symmetry corresponds to exchanging $\boldsymbol{\rho}$ and $\boldsymbol{\sigma}$. We can see that the solutions of the two $Q$-systems

are indeed in one-to-one correspondence.

The structure of this paper is as follows. In Section 2, we present the construction of the rational $Q$-system for a generic $A$-type quiver. We describe how to solve the rational $Q$-systems in Section 3. In Section 4, we review the 3d $\mathcal{N} = 4$ supersymmetric gauge theories, with an emphasis on brane realization and relations to BAE. In Section 5, we discuss Higgsings of the supersymmetric gauge theories and their realizations in rational $Q$-system. In Section 6, we discuss mirror symmetry in supersymmetric gauge theory and rational $Q$-system. We comment on the Bethe/Gauge correspondence for orthosympletic quivers and the rational $Q$-system for integrable open spin chains in Section 7. We conclude in Section 8. Some detailed technical derivations are delegated to the appendices.

## 2 Rational $Q$-system

In this section, we present the rational $Q$-system for generic Bethe ansatz equation of $A_{\ell-1}$-type.

### 2.1 $QQ$-relations and BAE

#### 2.1.1 $A_{\ell-1}$-type BAE

The $A_{\ell-1}$-type BAE can be encoded in an $A_{\ell-1}$-type Dynkin diagram as is shown in Figure 1. We label the nodes from left to right as $1, 2, \ldots, \ell-1$. Each node-$s$ is associated

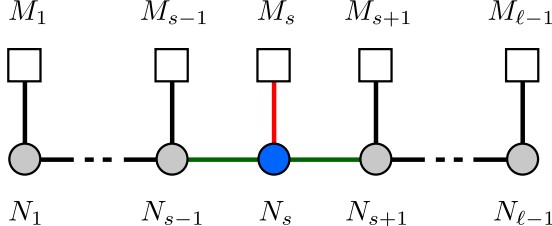

**Figure 1**. An $A_{\ell-1}$-type Dynkin diagram

with two sets of variables. The one associated with each circle is called *Bethe roots*, the number of Bethe roots is denoted by $N_s$; The other associated with the box on top of the circle is called *inhomogeneities*, the number of which is denoted by $M_s$. The inhomogeneities $\theta_j$ are *parameters* of the BAE and can be set to any values freely. On the other hand, Bethe roots are the unknown variables and should be found by solving BAE. At each node-$s$, the BAE is a set of $N_s$ algebraic equations $P_a^{(s)} = 1$ ($a = 1, 2, \ldots, N_s$), where $P_a^{(s)}$ is given by

$$P_a^{(s)} = \tau^{(s)} \prod_{\substack{d=1 \\ d \neq a}}^{N_s} \frac{\varphi\big(u_a^{(s)} - u_d^{(s)} + \eta\big)}{\varphi\big(u_a^{(s)} - u_d^{(s)} - \eta\big)} \prod_{j=1}^{M_s} \frac{\varphi\big(u_a^{(s)} - \theta_j^{(s)} - \frac{\eta}{2}\big)}{\varphi\big(u_a^{(s)} - \theta_j^{(s)} + \frac{\eta}{2}\big)} \tag{2.1}$$

$$\times \prod_{b=1}^{N_{s-1}} \frac{\varphi\big(u_a^{(s)} - u_b^{(s-1)} - \frac{\eta}{2}\big)}{\varphi\big(u_a^{(s)} - u_b^{(s-1)} + \frac{\eta}{2}\big)} \prod_{c=1}^{N_{s+1}} \frac{\varphi\big(u_a^{(s)} - u_c^{(s+1)} - \frac{\eta}{2}\big)}{\varphi\big(u_a^{(s)} - u_c^{(s+1)} + \frac{\eta}{2}\big)}$$

where the function $\varphi(x)$ is given by

$$\varphi(x) = \begin{cases} \sinh(x), & \text{XXZ-type;} \\ x, & \text{XXX-type.} \end{cases} \tag{2.2}$$

The parameter $\eta$ is related to the anisotropy or the quantum deformation parameter of the XXZ-type spin chain. For the XXX-type spin chain, we take $\varphi(x) = x$ and $\eta = \mathrm{i}$[1]. The parameters $\tau^{(s)}$ denote the twists.

For the XXZ-type BAE, it is sometimes more convenient to work with multiplicative variables which are defined as

$$x_j \equiv e^{2u_j}, \qquad y_j \equiv e^{2\theta_j}, \qquad q \equiv e^{\eta}. \tag{2.3}$$

In terms of which (2.1) becomes

$$P_a^{(s)} = \tilde{\tau}^{(s)} \prod_{\substack{d=1 \\ d \neq a}}^{N_s} \frac{x_a^{(s)} q - x_d^{(s)} q^{-1}}{x_d^{(s)} q - x_a^{(s)} q^{-1}} \prod_{j=1}^{M_s} \frac{x_a^{(s)} - y_j^{(s)} q}{y_j^{(s)} - x_a^{(s)} q} \tag{2.4}$$

$$\times \prod_{b=1}^{N_{s-1}} \frac{x_a^{(s)} - x_b^{(s-1)} q}{x_b^{(s-1)} - x_a^{(s)} q} \prod_{c=1}^{N_{s+1}} \frac{x_a^{(s)} - x_b^{(s+1)} q}{x_b^{(s+1)} - x_a^{(s)} q}$$

where

$$\tilde{\tau}^{(s)} = \tau^{(s)} \times (-1)^{N_{s-1} + N_s + N_{s+1} + M_s - 1}. \tag{2.5}$$

### 2.1.2 Rational $Q$-system

The BAE given in the previous subsection can be reformulated in terms of a set of $QQ$-relations, equipped with proper boundary conditions. Let us first describe the rational $Q$-system for XXX-type model following [13]. A $Q$-system is defined on a Young tableaux as is shown in Figure 2. At each point we associate a $Q$-function, which is a rational or hyperbolic function in one variable called the *spectral parameter*[2]. The four $Q$-functions associated to the four corners of each box are related by the $QQ$-relation given in (2.15). Therefore, not all $Q$-functions are independent. By fixing a few $Q$-functions and imposing analytic properties for the $Q$-functions, we can determine all the $Q$-functions on the Young tableaux. The $Q$-functions on the upper boundary are fixed to be 1. We fix the $Q$-functions on the left boundary partially. We call the precise form of the $Q$-functions on the left boundary the *boundary condition* of the rational $Q$-system. As we see later, different choices of boundary conditions lead to different BAEs. We give more detailed derivations

---

[1]The value of $\eta$ is irrelevant as long as it is non-vanishing, because we can always bring $\eta = \mathrm{i}$ by rescaling Bethe roots

[2]The $Q$-function can have more complicated analytic structures in other models.

in what follows.

**Young tableaux**   For each $A_{\ell-1}$-type BAE, the Young tableaux has $\ell$ rows

$$\vec{\lambda} = (\lambda_1, \lambda_2, \ldots, \lambda_\ell) \tag{2.6}$$

where $\lambda_k$ is the number of boxes of the $k$-th row. As a convention, we count the rows from the bottom to the top, as is shown in Figure 2. We require that

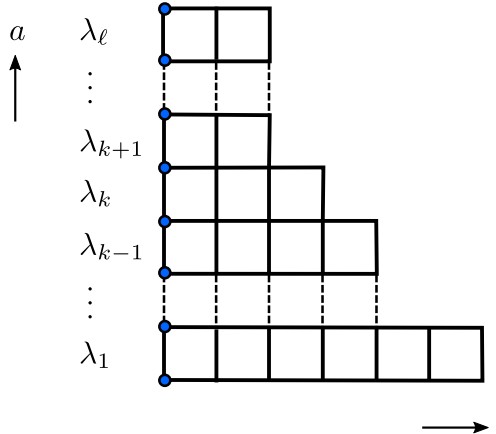

**Figure 2**. Young tableaux associated with an $A_{\ell-1}$-type Dynkin diagram.

$$\lambda_1 \geq \lambda_2 \geq \ldots \geq \lambda_\ell . \tag{2.7}$$

For a $A_{\ell-1}$ Dynkin diagram specified by

$$\vec{M} = (M_1, M_2, \ldots, M_{\ell-1}), \qquad \vec{N} = (N_1, N_2, \ldots, N_{\ell-1}) , \tag{2.8}$$

The number of boxes are given by

$$\lambda_\ell = N_{\ell-1} , \tag{2.9}$$
$$\lambda_a = N_{a-1} - N_a + (M_a + M_{a+1} + \ldots + M_{\ell-1}), \quad a = 2, \ldots, \ell-1 ,$$
$$\lambda_1 = M_1 + M_2 + \ldots + M_{\ell-1} - N_1 .$$

The total number of boxes is thus

$$\sum_{k=1}^{\ell} \lambda_a = M_1 + 2M_2 + \ldots + (\ell-1)M_{\ell-1} , \tag{2.10}$$

which is independent of $N_a$. Two comments are in order. Firstly, for a given set of BAE with multiple momentum carrying nodes, we propose the corresponding Young tableaux is given by (2.9). In the special case $\vec{M} = (M, 0, \ldots, 0)$, we recover the Young tableaux given

in [13, 17] which correspond to BAE with one momentum carrying node. Secondly, for the Young tableaux, we impose the requirement

$$\lambda_1 \geq \lambda_2 \ldots \geq \lambda_\ell \,. \tag{2.11}$$

This requirement results in certain constraints on the choices of $M_i$ and $N_i$. For some simple cases, the physical meaning of such requirements is clear. Let us explain this with two examples. In the SU(2) invariant XXX spin chain, we have $M_1 = M$ which is the length of the spin chain and $N_1 = N$ is the number of magnons. For BAE with length $M$ and magnon number $N$, the corresponding Young tableaux is $\boldsymbol{\lambda} = (M - N, N)$. The requirement becomes

$$M - N \geq N \,. \tag{2.12}$$

In the Bethe state of XXX spin chain, $N$ is the number of down spins and $M - N$ is the number of up spins. This requirement states that the number of up spins should be greater or equal than the number of down spins. The reason that one can impose this restriction is that we can obtain the Bethe states with $N > M - N$ by flipping all the spins simultaneously. Physically, there is nothing wrong to consider Bethe states with $N > M - N$, which corresponds to the solutions of BAE 'beyond the equator' [27, 28]. However, this is not necessary because we can construct all the Bethe states first within the region $M - N \geq N$ and then obtain the rest of the states by flipping all the spins.

As another example, we can consider the SU(3) invariant XXX spin chain. The Bethe equations are given by $\vec{M} = (M, 0)$ and $\vec{N} = (N_1, N_2)$ where $M$ is the length of the spin chain. The corresponding Young tableaux reads $\boldsymbol{\lambda} = (M - N_1, N_1 - N_2, N_2)$. We have the requirement

$$M - N_1 \geq N_1 - N_2 \geq N_2 \,. \tag{2.13}$$

The local Hilbert space of SU(3) invariant spin chain is $\mathbb{C}^3$. We can denote the basis states by $|1\rangle, |2\rangle, |3\rangle$. In the framework of nested Bethe ansatz, $M - N_1$, $N_2 - N_1$, $N_3$ are the number of polarizations $|1\rangle$, $|2\rangle$, $|3\rangle$ respectively of the Bethe state. The requirement (2.13) means $\#1 \geq \#2 \geq \#3$. Similar to the SU(2) case, we can first focus on the Bethe states within this region. The rest of the states can be obtained by permuting the role of $|1\rangle$, $|2\rangle$ and $|3\rangle$ properly.

We expect similar interpretations applies to more general cases. However, since we do not yet have a clear understanding of the nested Bethe ansatz for spin chains with generic $A_{\ell-1}$-type quiver with multiple momentum carrying nodes, we are not able to complete such physical interpretations for the generic case. Interestingly, the requirement (2.11) makes perfect physical sense in quiver gauge theories in Bethe/Gauge correspondence, as

we discuss later. This is a first hint that rational $Q$-system is a natural language for the Bethe/Gauge correspondence.

**$Q$-functions**  At each point $(a, s)$ of the Young tableaux, we define a $Q$-function denoted by $\mathbb{Q}_{a,s}(u)$ where $u$ is the spectral parameter. For the XXX-type BAE, the $Q$-functions are *polynomials* of the spectral parameter $u$. For the XXZ-type BAE, the $Q$-functions are *rational functions* of the multiplicative spectral parameter $x$. The number of boxes on each row is related to the asymptotic behavior of the $Q$-functions at the Southwest corner of the box at the left boundary. More precisely,

$$\lim_{u \to \infty} \mathbb{Q}_{a,0}^{\mathrm{XXX}}(u) = u^{\lambda_{a+1}+\lambda_{a+2}+\dots+\lambda_\ell} \left(1 + \mathcal{O}(u^{-1})\right), \qquad (2.14)$$

$$\lim_{x \to \infty} \mathbb{Q}_{a,0}^{\mathrm{XXZ}}(x) = x^{\lambda_{a+1}+\lambda_{a+2}+\dots+\lambda_\ell} \left(1 + \mathcal{O}(x^{-1})\right).$$

**$QQ$-relation**  The $Q$-functions defined on the box whose Southwest corner is located at $(a, s)$ satisfy the following $QQ$-relation

$$\mathbb{Q}_{a+1,s}\mathbb{Q}_{a,s+1} = \mathbb{Q}_{a+1,s+1}^+\mathbb{Q}_{a,s}^- - \epsilon_a\, \mathbb{Q}_{a+1,s+1}^-\mathbb{Q}_{a,s}^+ \qquad (2.15)$$

where $\epsilon_a$ are constants which are related to the diagonal twists in BAE. $\mathbb{Q}_{a,s}^{\pm}$ is as follows

$$\begin{cases} \mathbb{Q}_{a,s}^{\pm}(u) \equiv \mathbb{Q}_{a,s}(u \pm \tfrac{\mathrm{i}}{2}), & \text{XXX-type model} \\ \mathbb{Q}_{a,s}^{\pm}(x) \equiv \mathbb{Q}_{a,s}\left(xq^{\pm 1}\right), & \text{XXZ-type model} \end{cases} \qquad (2.16)$$

Among all the $Q$-functions, the ones at the left boundary (denoted by blue dots in Figure 2) are the most important because their zeros are related to the Bethe roots and inhomogeneities.

**Boundary condition**  Since $QQ$-relations relate the $Q$-functions at different points, the $Q$-functions are not independent. As a result, we can fix certain $Q$-functions and determine the rest by $QQ$-relations. We fix the $Q$-functions at the boundaries of the Young tableaux. The $Q$-functions at the upper boundary are fixed to be 1. We fix the $Q$-function at the left boundary partially. In what follows, we discuss the XXX-type and the XXZ-type $Q$-system separately.

For the XXX-type $Q$-system, the $Q$-functions at the left boundary take the form

$$\mathbb{Q}_{a,0}(u) = f_a(u)Q_a(u), \qquad a = 0, 1, \dots, \ell - 1 \qquad (2.17)$$

where $f_a(u)$ are some fixed functions whose zeros are related to inhomogeneities, we will discuss these function in more detail shortly. The functions $Q_a(u)$ are Baxter's $Q$-functions

whose zeros are the Bethe roots, namely

$$Q_a(u) = \prod_{k=1}^{N_a} \left( u - u_k^{(a)} \right) , \tag{2.18}$$

where $\{u_k^{(a)}\}$ are the Bethe roots associated to node-$a$ of the Dynkin diagram. For the XXZ-type $Q$-system, we consider the $Q$-functions with multiplicative spectral parameter. The $Q$-functions at the left boundary take the form

$$\mathbb{Q}_{a,0}(x) = f_a(x)Q_a(x) , \tag{2.19}$$

where again $f_a(x)$ is a fix function whose zeros are related to inhomogeneities and $Q_a(x)$ is Baxter's $Q$-function defined by

$$Q_a(x) = \prod_{j=1}^{N_a} \left( (x/x_j^{(a)})^{1/2} - (x_j^{(a)}/x)^{1/2} \right) , \tag{2.20}$$

where $x_j^{(a)}$ are the Bethe roots in the multiplicative variable. The function $f_a(x)$ is a rational function of $x$, which is discussed in more detail in the next subsection.

## 2.2 From $QQ$-relation to BAE

As a consistency check, we show that the $A_{\ell-1}$-type BAE (2.1) can be derived from the rational $Q$-system with proper boundary conditions $f_a(u)$. The following discussions apply to both XXX-type and XXZ-type spin chains. For the XXZ-type model, the spectral parameter of the $Q$-function should be understood as the multiplicative one with the corresponding shifts defined in (2.16). To obtain BAE at the $a$-th node, we consider the $QQ$-relation for node $a-1$ and $a$ with $s=0$

$$\mathbb{Q}_{a,0}\mathbb{Q}_{a-1,1} = \mathbb{Q}_{a,1}^+ \mathbb{Q}_{a-1,0}^- - \epsilon_{a-1}\mathbb{Q}_{a,1}^- \mathbb{Q}_{a-1,0}^+ , \tag{2.21}$$

$$\mathbb{Q}_{a+1,0}\mathbb{Q}_{a,1} = \mathbb{Q}_{a+1,1}^+ \mathbb{Q}_{a,0}^- - \epsilon_a \mathbb{Q}_{a+1,0}^- \mathbb{Q}_{a,0}^+ . \tag{2.22}$$

Taking $u = u_k^{(a)}$ in (2.21), we obtain

$$\mathbb{Q}_{a,1}^+ \left( u_k^{(a)} \right) \mathbb{Q}_{a-1,0}^- \left( u_k^{(a)} \right) - \epsilon_{a-1} \mathbb{Q}_{a,1}^- \left( u_k^{(a)} \right) \mathbb{Q}_{a-1,0}^+ \left( u_k^{(a)} \right) = 0 . \tag{2.23}$$

Assuming none of the factors above vanish[3], we can rewrite it as

$$\frac{\mathbb{Q}_{a-1,0}^- \left( u_k^{(a)} \right)}{\mathbb{Q}_{a-1,0}^+ \left( u_k^{(a)} \right)} \frac{\mathbb{Q}_{a,1}^+ \left( u_k^{(a)} \right)}{\mathbb{Q}_{a,1}^- \left( u_k^{(a)} \right)} \frac{1}{\epsilon_{a-1}} = 1 . \tag{2.24}$$

---

[3]This assumption is necessary especially for the higher rank case. Otherwise there can be unwanted solutions generated by the $Q$-system.

Evaluating (2.22) at Bethe roots with proper shifts, we obtain

$$\mathbb{Q}^+_{a+1,0}\big(u^{(a)}_k\big)\mathbb{Q}^+_{a,1}\big(u^{(a)}_k\big) = -\epsilon_a\,\mathbb{Q}_{a+1,1}\big(u^{(a)}_k\big)\mathbb{Q}^{++}_{a,0}\big(u^{(a)}_k\big)\,, \tag{2.25}$$
$$\mathbb{Q}^-_{a+1,0}\big(u^{(a)}_k\big)\mathbb{Q}^-_{a,1}\big(u^{(a)}_k\big) = \mathbb{Q}_{a+1,1}\big(u^{(a)}_k\big)\mathbb{Q}^{--}_{a,0}\big(u^{(a)}_k\big)\,.$$

Assuming none of the factors vanish, we can take the ratio of these equations and obtain

$$\frac{\mathbb{Q}^+_{a,1}\big(u^{(a)}_k\big)}{\mathbb{Q}^-_{a,1}\big(u^{(a)}_k\big)} = -\epsilon_a\frac{\mathbb{Q}^{++}_{a,0}\big(u^{(a)}_k\big)}{\mathbb{Q}^{--}_{a,0}\big(u^{(a)}_k\big)}\frac{\mathbb{Q}^-_{a+1,0}\big(u^{(a)}_k\big)}{\mathbb{Q}^+_{a+1,0}\big(u^{(a)}_k\big)}\,. \tag{2.26}$$

Inserting this into (2.24), we find

$$\frac{\epsilon_a}{\epsilon_{a-1}}\frac{\mathbb{Q}^-_{a-1,0}\big(u^{(a)}_k\big)}{\mathbb{Q}^+_{a-1,0}\big(u^{(a)}_k\big)}\frac{\mathbb{Q}^{++}_{a,0}\big(u^{(a)}_k\big)}{\mathbb{Q}^{--}_{a,0}\big(u^{(a)}_k\big)}\frac{\mathbb{Q}^-_{a+1,0}\big(u^{(a)}_k\big)}{\mathbb{Q}^+_{a+1,0}\big(u^{(a)}_k\big)} = -1\,. \tag{2.27}$$

The BAE in (2.1) can be written in terms of Baxter's $Q$-functions as

$$\tau^{(a)}\frac{Q^{++}_a\big(u^{(a)}_k\big)}{Q^{--}_a\big(u^{(a)}_k\big)}\frac{B^-_a\big(u^{(a)}_k\big)}{B^+_a\big(u^{(a)}_k\big)}\frac{Q^-_{a-1}\big(u^{(a)}_k\big)}{Q^+_{a-1}\big(u^{(a)}_k\big)}\frac{Q^-_{a+1}\big(u^{(a)}_k\big)}{Q^+_{a+1}\big(u^{(a)}_k\big)} = -1\,. \tag{2.28}$$

with $Q_a$ defined in (2.18) and (2.20) for the XXX-type and XXZ-type model respectively. We have also introduced Baxter's polynomials $B_a(u)$ in (2.28) whose zeros are the inhomogeneities. More explicitly,

$$\begin{cases} B_a(u) = \prod_{j=1}^{M_a}\big(u - \theta^{(a)}_j\big)\,, & \text{XXX-type model} \\ B_a(x) = \prod_{j=1}^{M_a}\Big(\big(x/y^{(a)}_j\big)^{1/2} - \big(y^{(a)}_j/x\big)^{1/2}\Big)\,, & \text{XXZ-type model} \end{cases} \tag{2.29}$$

The shifts in the spectral parameter are defined in the same way as before. Comparing (2.27) and (2.28) and using (2.17), we find that if the functions $f_a$ satisfy

$$\frac{f^-_{a-1}\big(u^{(a)}_k\big)}{f^+_{a-1}\big(u^{(a)}_k\big)}\frac{f^{++}_a\big(u^{(a)}_k\big)}{f^{--}_a\big(u^{(a)}_k\big)}\frac{f^-_{a+1}\big(u^{(a)}_k\big)}{f^+_{a+1}\big(u^{(a)}_k\big)} = \frac{B^-_a\big(u^{(a)}_k\big)}{B^+_a\big(u^{(a)}_k\big)} \tag{2.30}$$

and $\epsilon_a$ satisfy

$$\tau^{(a)} = \frac{\epsilon_a}{\epsilon_{a-1}}\,, \tag{2.31}$$

then (2.27) can be identified with (2.28). The functions $f_a$ satisfying (2.30) can be constructed as

$$f_a(u) = \prod_{k=1}^{\ell-a-1} F_{\ell-a-k}(u|\boldsymbol{\theta}_{\ell-k}) = F_1(u|\boldsymbol{\theta}_{a+1})F_2(u|\boldsymbol{\theta}_{a+2})\dots F_{\ell-a-1}(u|\boldsymbol{\theta}_{\ell-1}) \tag{2.32}$$

where the functions $F_n(x|\boldsymbol{\theta}_a)$ satisfy

$$\frac{F_{n-1}^-(u|\boldsymbol{\theta}_a)}{F_{n-1}^+(u|\boldsymbol{\theta}_a)} \frac{F_n^{++}(u|\boldsymbol{\theta}_a)}{F_n^{--}(u|\boldsymbol{\theta}_a)} \frac{F_{n+1}^-(u|\boldsymbol{\theta}_a)}{F_{n+1}^+(u|\boldsymbol{\theta}_a)} = 1\,, \tag{2.33}$$

and

$$F_1(u|\boldsymbol{\theta}_a) = B_a(u), \qquad F_0(u|\boldsymbol{\theta}_a) = 1. \tag{2.34}$$

The functions $F_n(u|\boldsymbol{\theta}_a)$ can be constructed by $B_a$ defined in (2.29) as

$$F_n(u|\boldsymbol{\theta}_a) = \prod_{k=1}^n B_a^{[2k-n-1]}(u|\boldsymbol{\theta}_a)\,, \tag{2.35}$$

where

$$\begin{cases} B_a^{[m]}(u|\boldsymbol{\theta}_a) \equiv B_a\left(u + \frac{m\mathrm{i}}{2}|\boldsymbol{\theta}_a\right), & \text{XXX-type model} \\ B_a^{[m]}(x|\boldsymbol{y}_a) \equiv B_a\left(xq^{m/2}|\boldsymbol{y}_a\right). & \text{XXZ-type model} \end{cases} \tag{2.36}$$

For example, the first few $F_n(u|\boldsymbol{\theta}_a)$ are given by

$$\begin{aligned} F_1(u|\boldsymbol{\theta}_a) &= B_a(u)\,, \tag{2.37}\\ F_2(u|\boldsymbol{\theta}_a) &= B_a^-(u)B_a^+(u)\,, \\ F_3(u|\boldsymbol{\theta}_a) &= B_a^{[-2]}(u)B_a(u)B_a^{[2]}(u)\,. \\ &\dots \end{aligned}$$

The condition (2.31) can be solved by taking

$$\epsilon_0 = 1, \qquad \epsilon_a = \tau^{(1)}\tau^{(2)}\dots\tau^{(a)}, \quad a = 1,\dots,\ell. \tag{2.38}$$

To sum up, the general $A_{\ell-1}$-type BAE can be obtained from the $QQ$-relations with the boundary condition (2.32) and the choice of the parameter $\epsilon_a$ given in (2.38).

**Physical meaning of parameters** Let us explain the physical meanings of the inhomogeneities, twists and $q$-deformation in the spin chain language with the simplest $A_1$ model.

The XXX-type model with $\theta_j = 0$ and $\tau = 1$ is the famous Heisenberg XXX spin chain, which was proposed by W. Heisenberg and solved by H. Bethe himself, whose Hamiltonian is given by

$$H_{\text{XXX}} = \sum_{n=1}^M (\sigma_n^x \sigma_{n+1}^x + \sigma_n^y \sigma_{n+1}^y + \sigma_n^z \sigma_{n+1}^z) \tag{2.39}$$

with periodic boundary conditions. Introducing a twist $\tau$ means imposing a twisted boundary condition $\sigma_{m+M}^{\pm} = \epsilon_{\pm} \sigma_m^{\pm}$ where $\sigma^{\pm} = (\sigma^x \pm i\sigma^y)/2$ such that $\tau = \epsilon_-/\epsilon_+$.

The XXZ-type $A_1$ model corresponds to the Heisenberg XXZ spin chain whose Hamiltonian is given by

$$H_{\text{XXZ}} = \sum_{n=1}^{M} (\sigma_n^x \sigma_{n+1}^x + \sigma_n^y \sigma_{n+1}^y + \Delta \sigma_n^z \sigma_{n+1}^z) \tag{2.40}$$

where $\Delta$ is the anisotropy. It is related to $\eta$ and $q$ in BAE as follows:

$$\Delta = \cosh \eta = \frac{1}{2}(q + q^{-1}). \tag{2.41}$$

Conventionally, the XXZ spin chain is considered as the $q$-deformation of the XXX spin chain. We shall adopt the same terminology here and view the XXZ-type model as the $q$-deformation of the XXX-type model where the deformation parameter is $q = e^\eta$.

The inhomogeneities are slightly more difficult to explain at the level of Hamiltonian. It is most easily introduced in the framework of Algebraic Bethe ansatz where we shift each Lax operator by different amounts, given by the inhomogeneities, see for example [29, 30] and references therein. The resulting model is still integrable, but the Hamiltonian is no longer a nearest neighboring interacting spin chain and is rather complicated to write down.

**Symmetry enhancement**  The twistless XXX-type models are special because they preserve extra symmetries. As a result, there are extra degeneracies in the spectrum. For example, the twistless Heisenberg XXX spin chain preserves the full SU(2) symmetry of the spin chain. Therefore, the spectrum is organized according to this symmetry. States in the same multiplet have the same energy. The descendant states are characterized by the same set of Bethe roots, but with additional roots at infinity. This fact is also reflected in the rational $Q$-system. Recall that for the XXX-type model, the $Q$-functions are polynomials of the spectral parameter $u$. If we take $\epsilon_a = 1$, from the $QQ$-relation we find that the order of $Q$-functions decreases as we move towards the right boundary. In fact, all the $Q$-functions at the right boundary are simply constants and can be set to 1. This is imposed as a boundary condition in the original Marboe-Volin prescription [13]. However, we would like to point out that it is a consequence of the rational $Q$-system for the twistless XXX-type model. Turning on either the twist, or the $q$-deformation breaks the symmetry. As a result, the $Q$-functions at the right boundary are no longer constants in these cases.

The extra degeneracies in the spectrum is also reflected by the number of solutions of BAE/Q-system. For the SU(2) invariant XXX spin chain with length $M$ and $N$ magnons, the number of solutions is given by $\binom{M}{N} - \binom{M}{N-1}$ [31]. On the other hand, for the twisted or $q$-deformed chain where the symmetry is broken to U(1), the number of solutions is $\binom{M}{N}$.

The 'missing' solutions are compensated by the descendant states.

## 3 Solving rational $Q$-systems

In this section, we discuss how to solve rational $Q$-systems. As we have seen from the previous section, the Bethe roots, which we are after, are the zeros of $Q_a(u)$. The idea of rational $Q$-system is first determining the functions $Q_a(u)$ and then finding their zeros. We first parameterize $Q_a(u)$ by $N_a$ parameters denoted by $\{c_k^{(a)}\}$, which are basically the elementary symmetric polynomials of the Bethe roots $\{u_k^{(a)}\}$. As discussed before, after fixing the $Q$-functions on the left boundary, we can use $QQ$-relation to determine the rest of the $Q$-functions. In general, such a procedure does not guarantee that the resulting $Q$-functions are polynomials (or Laurent polynomials in the XXZ case). Imposing this condition leads to a set of algebraic equations for $\{c_k^{(a)}\}$, which are called *zero remainder conditions*. We then solve the zero remainder conditions, which turns out to be more advantageous than directly working with original BAE.

### 3.1 The XXX-type $QQ$-relation

We first illustrate the basic strategy in detail for the XXX-type $Q$-system. For a given Young tableaux, we parameterize the $Q$-functions on the left boundary $\mathbb{Q}_{a,0}(u) = f_a(u)Q_a(u)$. The polynomial $f(u)$ is completely fixed by the inhomogeneities and is given in (2.32). We parameterize $Q_a(u)$ as

$$Q_a(u) = u^{N_a} + \sum_{k=0}^{N_a-1} c_k^{(a)} u^k \, . \tag{3.1}$$

Using the fact that

$$Q_a(u) = \prod_{k=1}^{N_a} \left(u - u_k^{(a)}\right) \tag{3.2}$$

we find that $c_k^{(a)}$ are essentially elementary symmetric polynomials of $\{u_k^{(a)}\}$, *e.g.* $c_0^{(a)} = (-1)^{N_a} u_1^{(a)} u_2^{(a)} \dots u_{N_a}^{(a)}$. After parameterizing $Q_a(u)$, we view them as 'known' functions and solve for the rest of the $Q$-functions on the Young tableaux. We solve the $Q$-functions row by row, from top to bottom.

1. The $Q$-functions on the upper boundary is fixed by the boundary condition, *i.e.* $\mathbb{Q}_{\ell,s}(u) = 1$. Therefore we start solving the $Q$-system from $a = \ell - 1$. The $QQ$-relation (2.15) becomes

$$\mathbb{Q}_{\ell-1,s+1} = \mathbb{Q}_{\ell-1,s}^- - \epsilon_{\ell-1}\mathbb{Q}_{\ell-1,s}^+ \tag{3.3}$$

This can be seen as a recursion relation and we can use it to compute all $\mathbb{Q}_{\ell-1,s}$ from $\mathbb{Q}_{\ell-1,0}$ as

$$\mathbb{Q}_{\ell-1,s}(u) = D^s_{\epsilon_{\ell-1}} \mathbb{Q}_{\ell-1,0} \, , \tag{3.4}$$

where the operator $D_\epsilon$ is defined by

$$D_\epsilon g(u) = g(u - \tfrac{i}{2}) - \epsilon \, g(u + \tfrac{i}{2}) \, . \tag{3.5}$$

2. We then consider the next row with $a = \ell - 2$. The $QQ$-relation reads

$$\mathbb{Q}_{\ell-2,s+1}\mathbb{Q}_{\ell-1,s} = \mathbb{Q}^+_{\ell-1,s+1}\mathbb{Q}^-_{\ell-2,s} - \epsilon_{\ell-2}\,\mathbb{Q}^-_{\ell-1,s+1}\mathbb{Q}^+_{\ell-2,s} \tag{3.6}$$

where the blue colored $Q$-functions are already determined from the previous step. This equation can be used to determine all $\mathbb{Q}_{\ell-2,s}$ from $\mathbb{Q}_{\ell-2,0}$ by writing it as

$$\mathbb{Q}_{\ell-2,s+1} = \frac{\mathbb{Q}^+_{\ell-1,s+1}\mathbb{Q}^-_{\ell-2,s} - \epsilon_{\ell-2}\,\mathbb{Q}^-_{\ell-1,s+1}\mathbb{Q}^+_{\ell-2,s}}{\mathbb{Q}_{\ell-1,s}} \, . \tag{3.7}$$

If we do not impose any constraints, the right hand side of (3.7) is in general a rational function of $u$ instead of a polynomial. The key point of the rational $Q$-system is that we require all the $Q$-functions to be polynomials in $u$. To impose this condition, we perform the polynomial division on the right hand side of (3.7), which gives a quotient and a remainder, both are polynomials in $u$. We then require the remainders to be zero, which leads to a set of algebraic equations for $\{c_k^{(\ell-1)}\}$.

3. Repeat the above procedure for all $a$ until we reach $a = 0$. Collect all the zero remainder conditions[4], which are the equivalence of BAE.

4. Solve the zero remainder conditions or manipulate it by other means such as computational algebraic geometry methods [7, 8, 11].

## 3.2 The XXZ-type $QQ$-relation

For the XXZ-type model, the $Q$-functions $\mathbb{Q}_{a,s}(x)$ are Laurent polynomials in the multiplicative variables $x$. However, it is rather inefficient to work with Laurent polynomials when solving the $QQ$-relations. Therefore, we first rewrite the $QQ$-relation in an equivalent polynomial form. The main idea of the rewriting is extracting proper global factors from the Laurent polynomials. After doing so, the $QQ$-relation

$$\mathbb{Q}_{a+1,s}(x)\mathbb{Q}_{a,s+1}(x) = \mathbb{Q}^+_{a+1,s+1}(x)\mathbb{Q}^-_{a,s}(x) - \epsilon_a\,\mathbb{Q}^-_{a+1,s+1}(x)\mathbb{Q}^+_{a,s}(x) \tag{3.8}$$

---

[4]We would like to point out that in practice, not all zero remainder conditions are needed. There exists a set of minimal choices of such relations which allows us to find the solutions of BAE. See [32] for related discussions.

can be rewritten as

$$\widetilde{Q}_{a+1,s}(x)\widetilde{Q}_{a,s+1}(x) = \widetilde{Q}^+_{a+1,s+1}(x)\widetilde{Q}^-_{a,s}(x) - \kappa_a(q)\,\widetilde{Q}^-_{a+1,s+1}(x)\widetilde{Q}^+_{a,s}(x) \qquad (3.9)$$

where $\widetilde{Q}_{a,s}(x)$ are *polynomials* in $x$ and the $q$-deformed twist $\kappa(q)$ is given by

$$\kappa_a(q) = \epsilon_a q^{-\lambda_{a+1}}\,. \qquad (3.10)$$

Recall that

$$\lambda_{a+1} = (N_a - N_{a+1}) - (M_{a+1} + M_{a+2} + \ldots + M_{\ell-1})\,. \qquad (3.11)$$

is the number of boxes of the $a+1$-th row. The boundary conditions become

$$\widetilde{Q}_{a,0}(x) = \tilde{f}_a(x)\widetilde{Q}_a(x) \qquad (3.12)$$

where

$$\tilde{f}_a(x) = \prod_{k=1}^{\ell-a-1} \widetilde{F}_k(x|\boldsymbol{y}_{a+k}) \qquad (3.13)$$

and

$$\widetilde{F}_n(x|\boldsymbol{y}_a) = \prod_{k=1}^{n} \widetilde{B}_a^{[2k-n-1]}(x|\boldsymbol{y}_a), \qquad \widetilde{B}_a^{[m]}(x|\boldsymbol{y}_a) = \prod_{j=1}^{M_a} \left(xq^m - y_j^{(a)}\right) \qquad (3.14)$$

are polynomials in $x$. We parameterize $\widetilde{Q}_a(x)$ by

$$\widetilde{Q}_a(x) = \prod_{j=1}^{N_a} \left(x - x_j^{(a)}\right) = x^{N_a} + \sum_{j=0}^{N_a-1} c_j^{(a)} x^j\,. \qquad (3.15)$$

As before, we obtain a system of algebraic equations for the variables $\{c_j^{(a)}\}$ by requiring all $Q$-functions $\widetilde{Q}_{a,s}$ to be polynomials in $x$. The procedure for deriving the zero remainder conditions are the same as in the XXX case and we shall not repeat it here.

## 3.3   Examples

In this section, we give three examples for rational $Q$-systems of $A_3$-type. They corresponds to the BAE of spin chains which are useful in various contexts. The Dynkin diagrams of the three $A_3$-type BAEs are given in Figure 3, we denote the three Dynkin diagrams by $A_3^{(1)}$, $A_3^{(2)}$ and $A_3^{(3)}$ respectively. We consider the homogeneous XXX-type model with periodic boundary condition, namely we take $\theta_k^{(a)} = 0$ and $\epsilon_a = 1$. In all these models, we distinguish between two kinds of nodes. The one which is connected to a box, meaning that

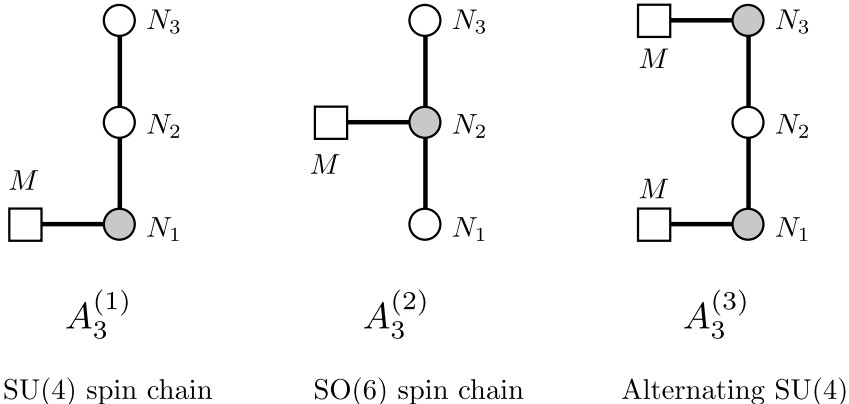

$$A_3^{(1)} \qquad\qquad A_3^{(2)} \qquad\qquad A_3^{(3)}$$

SU(4) spin chain      SO(6) spin chain      Alternating SU(4)

**Figure 3**. Three different rank-3 Dynkin diagrams.

it has non-zero number of inhomogeneities, are called *momentum carrying* while the rest are called *auxiliary*. The reason is that, it turns out the conserved charges such as momentum and energy of the state only depends on the Bethe roots of the momentum carrying nodes explicitly, while Bethe roots of auxiliary nodes only enter implicitly through solving BAE.

**SU(4) spin chain**   This is the simplest $A_3$-type spin chain. Let us denote the Bethe roots by $\{u_k^{(a)}\}$, $a = 1, 2, \ldots, N_a$ respectively. The corresponding BAE read

$$\left(\frac{u_k^{(1)} + \frac{\mathrm{i}}{2}}{u_k^{(1)} - \frac{\mathrm{i}}{2}}\right)^M = \prod_{j \neq k}^{N_1} \frac{u_k^{(1)} - u_j^{(1)} + \mathrm{i}}{u_k^{(1)} - u_j^{(1)} - \mathrm{i}} \prod_{l=1}^{N_2} \frac{u_k^{(1)} - u_l^{(2)} - \frac{\mathrm{i}}{2}}{u_k^{(1)} - u_l^{(2)} + \frac{\mathrm{i}}{2}}, \tag{3.16}$$

$$1 = \prod_{l=1}^{N_1} \frac{u_k^{(2)} - u_l^{(1)} - \frac{\mathrm{i}}{2}}{u_k^{(2)} - u_l^{(1)} + \frac{\mathrm{i}}{2}} \prod_{j \neq k}^{N_2} \frac{u_k^{(2)} - u_j^{(2)} + \mathrm{i}}{u_k^{(1)} - u_j^{(2)} - \mathrm{i}} \prod_{l=1}^{N_3} \frac{u_k^{(2)} - u_l^{(3)} - \frac{\mathrm{i}}{2}}{u_k^{(2)} - u_l^{(3)} + \frac{\mathrm{i}}{2}},$$

$$1 = \prod_{j \neq k}^{N_3} \frac{u_k^{(3)} - u_j^{(3)} + \mathrm{i}}{u_k^{(3)} - u_j^{(3)} - \mathrm{i}} \prod_{l=1}^{N_2} \frac{u_k^{(3)} - u_l^{(2)} - \frac{\mathrm{i}}{2}}{u_k^{(3)} - u_l^{(2)} + \frac{\mathrm{i}}{2}}.$$

Let us briefly explain the origin of Bethe equations. The SU(4) spin chain is a quantum integrable model. At each site of the spin chain, the local Hilbert space has 4 polarizations. We can denote the corresponding states by $|1\rangle, \ldots, |4\rangle$. The Hamiltonian of the spin chain is given by

$$H_{\mathrm{SU}(4)} = \sum_{n=1}^{M} (\mathrm{I}_{n,n+1} - \mathrm{P}_{n,n+1}) \tag{3.17}$$

where $\mathrm{I}_{n,n+1}$ and $\mathrm{P}_{n,n+1}$ are the identity and permutation operators that act on sites $n$ and $n+1$, *i.e.*

$$\mathrm{I}_{n,n+1}|a\rangle_n \otimes |b\rangle_{n+1} = |a\rangle_n \otimes |b\rangle_{n+1}, \qquad \mathrm{P}_{n,n+1}|a\rangle_n \otimes |b\rangle_{n+1} = |b\rangle_n \otimes |a\rangle_{n+1}. \tag{3.18}$$

We impose periodic boundary condition. The Hamiltonian (3.17) can be diagonalized by *nested* Bethe ansatz (see for example [33]). In the coordinate Bethe ansatz, the Bethe equations arise as the quantization conditions for the rapidities at different nesting levels because we have imposed periodic boundary condition.

The rational $Q$-system corresponding to the BAE (3.16) has 4 rows $\vec{\lambda} = (\lambda_1, \lambda_2, \lambda_3, \lambda_4)$ with the number of boxes given by

$$
\begin{aligned}
\lambda_1 &= M - N_1 \,, \\
\lambda_2 &= N_1 - N_2 \,, \\
\lambda_3 &= N_2 - N_3 \,, \\
\lambda_4 &= N_3 \,.
\end{aligned}
\tag{3.19}
$$

The boundary conditions are given by

$$
\mathbb{Q}_{0,0}(u) = u^M \,,
\tag{3.20}
$$

$$
\mathbb{Q}_{1,0}(u) = Q_1(u) = u^{N_1} + \sum_{k=0}^{N_1-1} c_k^{(1)} u^k \,,
$$

$$
\mathbb{Q}_{2,0}(u) = Q_2(u) = u^{N_2} + \sum_{k=0}^{N_2-1} c_k^{(2)} u^k \,,
$$

$$
\mathbb{Q}_{3,0}(u) = Q_3(u) = u^{N_3} + \sum_{k=0}^{N_3-1} c_k^{(3)} u^k \,.
$$

**SO(6) spin chain**  The SO(6) spin chain plays an important role in integrability of planar $\mathcal{N} = 4$ SYM theory. In the seminal paper of Minahan and Zarembo [34], they calculated the one-loop dilation operator of the scalar sector, which turns out to be identical to the Hamiltonian of the SO(6) spin chain. The BAE reads

$$
1 = \prod_{j \neq k}^{N_1} \frac{u_k^{(1)} - u_j^{(1)} + \mathrm{i}}{u_k^{(1)} - u_j^{(1)} - \mathrm{i}} \prod_{l=1}^{N_2} \frac{u_k^{(1)} - u_l^{(2)} - \frac{\mathrm{i}}{2}}{u_k^{(1)} - u_l^{(2)} - \frac{\mathrm{i}}{2}} \,,
\tag{3.21}
$$

$$
\left( \frac{u_k^{(2)} + \frac{\mathrm{i}}{2}}{u_k^{(2)} - \frac{\mathrm{i}}{2}} \right)^M = \prod_{l=1}^{N_1} \frac{u_k^{(2)} - u_l^{(1)} - \frac{\mathrm{i}}{2}}{u_k^{(2)} - u_l^{(1)} - \frac{\mathrm{i}}{2}} \prod_{j \neq k}^{N_2} \frac{u_k^{(2)} - u_j^{(2)} + \mathrm{i}}{u_k^{(2)} - u_j^{(2)} - \mathrm{i}} \prod_{l=1}^{N_3} \frac{u_k^{(2)} - u_l^{(3)} - \frac{\mathrm{i}}{2}}{u_k^{(2)} - u_l^{(3)} - \frac{\mathrm{i}}{2}} \,,
$$

$$
1 = \prod_{l=1}^{N_2} \frac{u_k^{(3)} - u_l^{(2)} - \frac{\mathrm{i}}{2}}{u_k^{(3)} - u_l^{(2)} - \frac{\mathrm{i}}{2}} \prod_{j \neq k}^{N_3} \frac{u_k^{(3)} - u_j^{(3)} + \mathrm{i}}{u_k^{(3)} - u_j^{(3)} - \mathrm{i}} \,.
$$

At each site of the spin chain, there are 6 possible polarizations, denoted by $|1\rangle, \ldots, |6\rangle$. The Hamiltonian of the SO(6) spin chain is given by

$$H_{\mathrm{SO}(6)} = \sum_{n=1}^{M} (\mathrm{K}_{n,n+1} + 2\mathrm{I}_{n,n+1} - 2\mathrm{P}_{n,n+1}) \tag{3.22}$$

where periodic boundary condition has been imposed and the operator $K_{n,n+1}$ acts on sites $n$ and $n+1$ as

$$K_{n,n+1}|a\rangle_n \otimes |b\rangle_{n+1} = \delta_{a,b} \sum_{c=1}^{6} |c\rangle_n \otimes |c\rangle_{n+1}. \tag{3.23}$$

The Young tableaux has four rows $\vec{\lambda} = (\lambda_1, \ldots, \lambda_4)$ with the number of boxes given by

$$\lambda_1 = M - N_1, \tag{3.24}$$
$$\lambda_2 = M + N_1 - N_2,$$
$$\lambda_3 = N_2 - N_3,$$
$$\lambda_4 = N_3.$$

The boundary condition is given by

$$\mathbb{Q}_{0,0}(u) = \left(u - \tfrac{\mathrm{i}}{2}\right)^M \left(u + \tfrac{\mathrm{i}}{2}\right)^M, \tag{3.25}$$

$$\mathbb{Q}_{1,0}(u) = u^M Q_1(u) = u^M \left(u^{N_1} + \sum_{k=0}^{N_1-1} c_k^{(1)} u^k\right),$$

$$\mathbb{Q}_{2,0}(u) = Q_2(u) = u^{N_2} + \sum_{k=0}^{N_2-1} c_k^{(2)} u^k,$$

$$\mathbb{Q}_{3,0}(u) = Q_3(u) = u^{N_3} + \sum_{k=0}^{N_3-1} c_k^{(2)} u^k.$$

**Alternating SU(4) spin chain**  The last example has two momentum carrying nodes. It plays an important role in the study of integrability of ABJM theory [35] where it was identified with the planar two-loop dilatation operator of ABJM theory in the scalar sector. The BAE reads

$$\left(\frac{u_k^{(3)} + \tfrac{\mathrm{i}}{2}}{u_k^{(3)} - \tfrac{\mathrm{i}}{2}}\right)^M = \prod_{j \neq k}^{N_3} \frac{u_k^{(3)} - u_j^{(3)} + \mathrm{i}}{u_k^{(3)} - u_j^{(3)} + \mathrm{i}} \prod_{l=1}^{N_2} \frac{u_k^{(3)} - u_j^{(2)} - \tfrac{\mathrm{i}}{2}}{u_k^{(3)} - u_j^{(2)} + \tfrac{\mathrm{i}}{2}}, \tag{3.26}$$

$$1 = \prod_{l=1}^{N_3} \frac{u_k^{(2)} - u_j^{(3)} - \tfrac{\mathrm{i}}{2}}{u_k^{(2)} - u_j^{(3)} + \tfrac{\mathrm{i}}{2}} \prod_{j \neq k}^{N_2} \frac{u_k^{(2)} - u_j^{(2)} + \mathrm{i}}{u_k^{(2)} - u_j^{(2)} + \mathrm{i}} \prod_{l=1}^{N_1} \frac{u_k^{(2)} - u_j^{(1)} - \tfrac{\mathrm{i}}{2}}{u_k^{(2)} - u_j^{(1)} + \tfrac{\mathrm{i}}{2}},$$

$$\left(\frac{u_k^{(1)} + \frac{\mathrm{i}}{2}}{u_k^{(1)} - \frac{\mathrm{i}}{2}}\right)^M = \prod_{l=1}^{N_2} \frac{u_k^{(1)} - u_j^{(2)} - \frac{\mathrm{i}}{2}}{u_k^{(1)} - u_j^{(2)} + \frac{\mathrm{i}}{2}} \prod_{j \neq k}^{N_1} \frac{u_k^{(1)} - u_j^{(1)} + \mathrm{i}}{u_k^{(1)} - u_j^{(1)} + \mathrm{i}}.$$

The Hamiltonian of the alternating SU(4) spin chain is given by

$$H_{\mathrm{ABJM}} = \sum_{n=1}^{2M} \left(2\mathrm{I}_{n,n+1} - 2\mathrm{P}_{n,n+2} + \mathrm{P}_{n,n+2}\mathrm{K}_{n,n+1} + \mathrm{K}_{n,n+1}\mathrm{P}_{n,n+2}\right) \tag{3.27}$$

where we impose the periodic boundary condition as before. It is called alternating spin chain because we distinguish even and odd sites of the spin chain. At each site, there are 4 possible polarizations $|1\rangle, \ldots, |4\rangle$.

The rational $Q$-system has 4 rows $\vec{\lambda} = (\lambda_1, \ldots, \lambda_4)$ with

$$\lambda_1 = 2M - N_1 \,, \tag{3.28}$$
$$\lambda_2 = M + N_1 - N_2 \,,$$
$$\lambda_3 = M + N_2 - N_3 \,,$$
$$\lambda_4 = N_3 \,.$$

The boundary condition is given by

$$\mathbb{Q}_{0,0}(u) = (u - \mathrm{i})^M u^{2M} (u + \mathrm{i})^M \,, \tag{3.29}$$
$$\mathbb{Q}_{1,0}(u) = \left(u - \tfrac{\mathrm{i}}{2}\right)^M \left(u + \tfrac{\mathrm{i}}{2}\right)^M Q_1(u) \,,$$
$$\mathbb{Q}_{2,0}(u) = u^M Q_2(u) \,,$$
$$\mathbb{Q}_{3,0}(u) = Q_3(u) \,,$$

where

$$Q_a(u) = u^{N_a} + \sum_{k=0}^{N_a - 1} c_k^{(a)} u^k \qquad a = 1, 2, 3 \,. \tag{3.30}$$

### 3.4 Efficiency in solving $Q$-systems

It is far more efficient to use rational $Q$-systems instead of Bethe ansatz equations to solve for the Bethe roots. First of all, although both are algebraic equations, $Q$-systems are simpler and faster to solve. In Table 1, we compare the time required to solve numerically with working precision 100 digits for the Bethe roots of an array of spin chains with generic inhomogeneities and twists using Bethe ansatz equations and $Q$-systems, respectively. In each example, the $Q$-systems take much less time to solve, and the discrepancy in time consumption becomes even greater when the spin chain is longer.

Secondly, Bethe equations can be plagued with various problems, for instance, there

| $(L, N)$ | BAE | $Q$-system |
|---|---|---|
| $(4, 2)$ | 0.238 | 0.105 |
| $(5, 2)$ | 0.512 | 0.155 |
| $(6, 3)$ | 199.7 | 0.385 |
| $(7, 3)$ | 1803 | 2.092 |
| $(8, 4)$ | – | 6.322 |
| $(9, 4)$ | – | 29.85 |
| $(10, 5)$ | – | 1145 |

**Table 1**. Time (in seconds) required to solve for the Bethe roots using either BAEs or rational $Q$-systems, on a computer with Intel Xeon Gold 6248R CPU. We set inhomogeneities and twists to arbitrary natural numbers and set anisotropy $q$ to $1/3$. $L, N$ are respectively the length of spin chain and the number of magnons. "−" means not solvable in reasonable time (more than four hours).

are non-physical solutions which need to be discarded. This problem is most pronounced in the special cases where inhomogeneities and twists are trivial. On the other hand, rational $Q$-systems solve both problems automatically: the unphysical solutions are automatically avoided and the singular solutions are automatically included. In other words, $Q$-systems know how to pick all the physical solutions. We illustrate these features in Section 4.4 for the evaluation of the topologically indices for 3d $\mathcal{N} = 4$ U($N$) SQCD theories with $L$ fundamental hypermultiplets.

## 4   3d $\mathcal{N} = 4$ theories

3-dimensional supersymmetric gauge theories with $\mathcal{N} = 4$ supersymmetry which flow to an interacting conformal theory in the IR have allowed to gain insights in dualities like 3d mirror symmetry [36]. In this Section, the field theory properties and the brane realisation in Type IIB superstring theory are recalled. Thereafter, the relation to Bethe Ansatz equations is reviewed by considering the equations for the supersymmetric vacua of the theory compactified to 2d. The connection between 3d $\mathcal{N} = 4$ theories and spin chain BAE has been discussed in [20, 21].

### 4.1   Brane realisations

The relevant class of 3d $\mathcal{N} = 4$ theories can be constructed via a D5-D3-NS5 brane system in Type IIB superstring theory [37]. Suppose the branes are arranged as in Figure 4 and occupy space-time directions as in Table 2. The 3d low-energy world-volume theory on the D3s is an $A$-type quiver gauge theory. Given $\ell$ NS5 branes which are separated along $x^6$, there are $N_i$ D3 branes suspended between the $i$-th and $(i+1)$-th NS5 brane. In addition, there are $M_i$ D5 branes with $x^6$ position in between the $i$-th and $(i+1)$-th NS5 brane. The

|      | 0 | 1 | 2 | 3 | 4 | 5 | 6 | 7 | 8 | 9 |
|------|---|---|---|---|---|---|---|---|---|---|
| NS5  | × | × | × | × | × | × |   |   |   |   |
| D3   | × | × | × |   |   |   | × |   |   |   |
| D5   | × | × | × |   |   |   |   | × | × | × |

**Table 2**. Space-time occupation of branes. Each brane individually breaks half of the original supercharges. However, the three different types of branes are arranged such that any subset of two branes allows to include the third type of branes without reducing supersymmetry further. Thus, the D5-D3-NS5 system has 8 supercharges. The branes break the $SO(1,9)$ space-time symmetry to $SO(1,2) \times SO(3)_{3,4,5} \times SO(3)_{7,8,9}$.

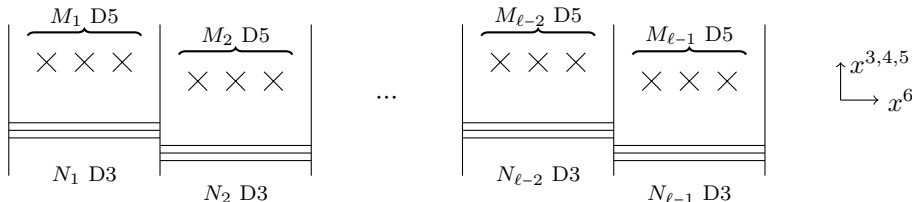

**Figure 4**. D5-D3-NS5 brane configuration. The vertical lines denote NS5 branes, the horizontal lines denote D3 branes, and the crosses are D5 branes.

resulting 3d $\mathcal{N} = 4$ gauge theory is conveniently encoded in the following quiver diagram

$$\tag{4.1}$$

where round nodes denote dynamical $U(N_i)$ vector multiplets and square nodes are background $U(M_i)$ vector multiplets. A solid line between two nodes encodes a hypermultiplet which transforms in the bifundamental representation of the two groups associated to the nodes.

Next, some fundamental properties of the 3d $\mathcal{N} = 4$ theory are recalled. The quiver gauge theory (4.1) flows to an interacting 3d $\mathcal{N} = 4$ SCFT in the IR if each $U(N_i)$ gauge node satisfies

$$e_i = N_{i-1} + N_{i+1} + M_i - 2N_i \geq 0 \tag{4.2}$$

for all $i = 1, 2, \ldots, \ell - 1$. Then (4.1) is referred to as *good* in the sense of [26]. The $\mathcal{N} = 4$ R-symmetry $SO(4)_R \cong SU(2)_H \times SU(2)_C$ is geometrically realised as rotation groups $SO(3)_{7,8,9} \subset SU(2)_H$ and $SO(3)_{3,4,5} \subset SU(2)_C$ in the brane system. The global (non-R) symmetry is a product of the form $G_H \times G_C$. The flavour symmetry $G_H$ is explicit in the

UV Lagrangian description of (4.1)

$$G_H = S\left(\prod_{j=1}^{\ell-1} \mathrm{U}(M_j)\right) = \left(\prod_{j=1}^{\ell-1} \mathrm{U}(M_j)\right)/\mathrm{U}(1)_{\mathrm{diag}}. \qquad (4.3)$$

In contrast, $G_C$ is less obvious. The UV description accounts for $\mathrm{U}(1)^{\ell-1}$, because each $\mathrm{U}(N_i)$ gauge group can be used to construct a conserved current. In the IR, the Coulomb branch symmetry might be enhanced to a non-abelian group $G_C \supset \mathrm{U}(1)^{\ell-1}$. A criterion for symmetry enhancement is given by the notion of *balance*, *i.e.* the node $\mathrm{U}(N_i)$ is balanced if $e_i = 0$. Then certain monopole operators act as ladder operators for the Coulomb branch symmetry, which becomes non-abelian. The reader is referred to [26, 38–41] for details on monopole operators and their role in symmetry enhancement. For linear quiver theories (4.1) the subset of balanced gauge nodes yields the Dynkin diagram of the non-abelian part of $G_C$ in the IR.

The 3d $\mathcal{N} = 4$ SCFT has two types of deformation parameters: (i) a triplet of masses $\vec{m}$ which correspond to Cartan elements of $G_H$ and transform as $[0] \otimes [2]$ under $\mathrm{SU}(2)_H \times \mathrm{SU}(2)_C$; and (ii) a triplet of FI parameters $\vec{w}$ which are Cartan elements of $G_C$ and transform as $[2] \otimes [0]$ under $\mathrm{SU}(2)_H \times \mathrm{SU}(2)_C$. In the brane setup, the masses $\vec{m}$ are realised by the D5 positions in $x^{3,4,5}$, which are acted on by $\mathrm{SO}(3)_{3,4,5}$. Similarly, the FI parameter for the gauge group between two adjacent NS5 branes is realised by the relative position along $x^{7,8,9}$, being acted on by $\mathrm{SO}(3)_{7,8,9}$.

**Repacking into partitions.** The linear quiver (4.1) falls into the well-known class of $T_{\boldsymbol{\rho}}^{\boldsymbol{\sigma}}[\mathrm{SU}(n)]$ theories [26], which are labelled by two partitions $\boldsymbol{\rho}, \boldsymbol{\sigma}$ of $n$:

$$\boldsymbol{\rho} = (\rho_1, \ldots, \rho_\ell) \quad \text{with} \quad \rho_1 \geq \ldots \geq \rho_\ell > 0, \qquad \sum_{i=1}^{\ell'} \rho_i = n, \qquad (4.4a)$$

$$\boldsymbol{\sigma} = (\sigma_1, \ldots, \sigma_{\ell'}) \quad \text{with} \quad \sigma_1 \geq \ldots \geq \sigma_{\ell'} > 0, \qquad \sum_{i=1}^{\ell} \sigma_i = n. \qquad (4.4b)$$

The two sets of integers $(N_1, \ldots, N_{\ell-1})$, $(M_1, \ldots, M_{\ell-1})$ are defined in terms of the partitions as follows:

$$M_j = \hat{\sigma}_j - \hat{\sigma}_{j+1} \quad \text{with} \quad \hat{\sigma}_i = 0, \quad i \geq \hat{l}' + 1 \qquad (4.5a)$$

$$N_j = \sum_{k=j+1}^{\ell} \rho_k - \sum_{i=j+1}^{\hat{\ell}'} \hat{\sigma}_i \quad \text{with} \quad \hat{\ell}' = \ell - 1, \qquad (4.5b)$$

wherein the transposed partition $\sigma^{\mathrm{T}} = (\hat{\sigma}_1, \ldots, \hat{\sigma}_{\hat{\ell}'})$ appears. For convenience, one can obtain partitions $\boldsymbol{\rho}, \boldsymbol{\sigma}$ from the integers $N_j$ and $M_j$ as follows:

$$\hat{\sigma}_j = \sum_{i=j}^{\ell-1} M_i \,, \qquad \rho_i = \begin{cases} \hat{\sigma}_1 - N_1 & i = 1 \,, \\ N_{i-1} - N_i + \hat{\sigma}_i & 1 < i \leq \ell - 1 \,, \\ N_{\ell-1} & i = \ell \,. \end{cases} \tag{4.6}$$

In terms of the brane system of Figure 4, the partition data appears naturally after a sequence of brane moves, including brane creation and annihilation [37], such that all D5 branes are on one side of all of the NS5 branes. The brane realisation of $T_{\boldsymbol{\rho}}^{\boldsymbol{\sigma}}[\mathrm{SU}(n)]$ is then given by $n$ D3 branes suspended between $\ell$ NS5 and $\ell'$ D5 branes. The parts of $\boldsymbol{\rho}$ are the net number of D3s ending in the NS5 branes going from the interior to exterior; likewise, the parts of $\boldsymbol{\sigma}$ are the net number of D3s ending on D5 branes going from interior to exterior.

## 4.2   3d $\mathcal{N} = 2^*$ theories on $\mathbb{R}^2 \times S^1$

Consider a 3d $\mathcal{N} = 4$ linear quiver gauge theory $\mathcal{T}$ on $\mathbb{R}^2 \times S^1$. To be more precise, consider the 3d $\mathcal{N} = 2^*$ theory on $\mathbb{R}^2 \times S^1$ that results from the 3d $\mathcal{N} = 4$ theory by turning on a mass for the adjoint chiral multiplet in the $\mathcal{N} = 4$ vector multiplet. To proceed, two steps are required: (i) the SUSY breaking to $\mathcal{N} = 2^*$ and (ii) the compactification to the 2d KK theory. The reader is referred to [18, 21] for references and details.

In terms of the supersymmetry algebra, one selects a $\mathcal{N} = 2$ subalgebra of the $\mathcal{N} = 4$ algebra. Denote the Cartan generators of the $\mathrm{SU}(2)_H \times \mathrm{SU}(2)_C$ R-symmetry by $j_H^3$ and $j_C^3$, respectively. Without loss of generality, the R-symmetry generator of the $\mathcal{N} = 2$ subalgebra can be chosen to be proportional to $j_H^3 + j_C^3$. However, the orthogonal combination $j_H^3 - j_C^3$ generates a global (non-R) symmetry $\mathrm{U}(1)_\eta$ from the $\mathcal{N} = 2$ perspective. Therefore, the 3d theory $\mathcal{T}$ has an $G_H \times G_C \times \mathrm{U}(1)_\eta$ global symmetry, viewed as $\mathcal{N} = 2$ theory. Turning on a real mass term $\mathrm{U}(1)_\eta$, via coupling a $\mathcal{N} = 2$ background $\mathrm{U}(1)_\eta$ vector multiplet, leads to the desired SUSY breaking $\mathcal{N} = 4 \to \mathcal{N} = 2^*$. The deformation parameters split naturally into real and complex: denote the third components of the triplets $\vec{m}$ and $\vec{w}$ simply by $m \equiv m^3$ and $w \equiv w^3$, respectively. The remaining components, which could be arranged in a complex linear combination like $m^1 + \mathrm{i}m^2$, are not relevant as they do not affect the low-energy effective 2d theory. Besides the real parameters $m$ and $w$, there is also the real mass $\frac{\tilde{\eta}}{2}$ for the $\mathrm{U}(1)_\eta$ symmetry.

Next, compactifying the 3d $\mathcal{N} = 2^*$ theory $\mathcal{T}$ on a circle of radius $R$, allows to combine the real deformation parameters $(m, w, \frac{\tilde{\eta}}{2})$ with arising flavour Wilson lines $a_0^F$ for $G_H$, $G_C$, and $\mathrm{U}(1)_\eta$, respectively, into complex deformation parameters

$$\theta_j = \mathrm{i}R\left(m_j + \mathrm{i}a_{0,j}^H\right) \,, \quad t_s = \mathrm{i}R\left(w_s + \mathrm{i}a_{0,s}^C\right) \,, \quad \eta = \mathrm{i}R\left(\tilde{\eta} + \mathrm{i}a_0^\eta\right) \,. \tag{4.7}$$

From the 2d $\mathcal{N} = (2,2)$ perspective, these correspond to twisted masses. As the flavour Wilson lines $a_0^F = \frac{1}{2\pi R} \int_{S^1} A_\mu^F \mathrm{d}z^\mu$ are periodic, it is more convenient to consider the exponentiated variables

$$y_j = e^{2\pi i \theta_j} , \quad \epsilon_s = e^{2\pi t_s} , \quad q = e^{\pi \eta} . \tag{4.8}$$

Similarly, the 3d $\mathcal{N} = 2$ vector multiplet contains a real scalar field $\sigma_a \equiv \phi_{3,a}$ with $a = 1, \ldots, \mathrm{rk}(G)$, which combines with a flat connection $a_{0,a}$ for the gauge field along $S^1$ into a complex scalar field $u_a$ and the single valued fugacity is obtained by exponentiation

$$x_a = e^{2\pi i u_a} \qquad u_a = iR(\sigma_a + ia_{0,a}) . \tag{4.9}$$

The 2d KK theory is best described by a low-energy effective description, wherein all massive fields have been integrated out. Assuming that the twisted masses are sufficiently generic, the 2d theory at low energies becomes effectively abelian. The field strength multiplet of the $\mathcal{N} = (2,2)$ vector multiplets are twisted chiral multiplets, whose dynamics is governed by the twisted superpotential $\widetilde{\mathcal{W}}$. The low-energy effective action is then determined by the low-energy effective twisted superpotential $\widetilde{\mathcal{W}}_{\text{eff}}$, which receives corrections from integrating out massive fields. Crucially, $\widetilde{\mathcal{W}}_{\text{eff}}$ is independent of the superpotential and the gauge coupling of the original 3d $\mathcal{N} = 2^*$ theory. This is the reason why the complex deformation parameters of the 3d theory can be neglected from the start, because they are superpotential deformations.

The contribution to $\widetilde{\mathcal{W}}_{\text{eff}}$ of a 3d $\mathcal{N} = 2$ chiral multiplet with twisted mass $u$ is given by [18, 42]

$$\widetilde{\mathcal{W}}_{\text{eff}}^{\text{chiral}} = \frac{1}{(2\pi i)^2} \mathrm{Li}_2 \left( e^{2\pi i u} \right) + \frac{1}{4} u^2 \equiv \ell(u) \tag{4.10}$$

and it follows that a 3d hypermultiplet contributes as

$$\widetilde{\mathcal{W}}_{\text{eff}}^{\text{hyper}} = \ell \left( u + \tfrac{1}{2} \eta \right) + \ell \left( -u + \tfrac{1}{2} \eta \right) . \tag{4.11}$$

From the 3d $\mathcal{N} = 4$ vector multiplet, only the adjoint chiral contributes to the effective twisted superpotential

$$\widetilde{\mathcal{W}}_{\text{eff}}^{\text{vector}} = \ell(u - \eta) . \tag{4.12}$$

Besides the contributions from the supermultiplets, the twisted superpotential may receive contributions from Chern-Simons interactions. As the origin is a 3d $\mathcal{N} = 4$ theory, pure gauge Chern-Simons term are not relevant; however, mixed gauge-flavour Chern-Simons interactions appear. For instance, the FI coupling is understood as such a mixed CS term

between the gauge symmetry and the topological symmetry. One finds

$$\widetilde{\mathcal{W}}_{\text{eff}}^{\text{FI}} = t \sum_{a=1}^{k} u_a \tag{4.13}$$

for $G = \mathrm{U}(k)$ which has a single $\mathrm{U}(1)$ topological symmetry.

Finally, the supersymmetric vacua of the compactified theory $\mathcal{T}$ with generic twisted masses are determined by the critical points

$$e^{2\pi\mathrm{i}\frac{\partial \widetilde{\mathcal{W}}_{\text{eff}}}{\partial u_a}} = 1 \qquad \text{for } a = 1, \ldots, \mathrm{rk}(G). \tag{4.14}$$

For theories $\mathcal{T}$ with sufficiently many flavours (and generic twisted masses) the set of supersymmetric vacua are a finite number of discrete points.

**Example.** To exemplify, consider $\mathrm{U}(k)$ SQCD with $N$ fundamental hypermultiplets. The low-energy effective twisted superpotential is given by

$$\widetilde{\mathcal{W}}_{\text{eff}} = \sum_{a=1}^{k}\sum_{j=1}^{N} \left[ \ell(u_a - \theta_j + \tfrac{1}{2}\eta) + \ell(-u_a + \theta_j + \tfrac{1}{2}\eta) \right] \tag{4.15}$$

$$+ \sum_{a,b=1}^{k} \ell(u_a - u_b - \eta) + (t_2 - t_1) \sum_{a=1}^{k} u_a$$

wherein the first line encodes the hypermultiplet in the bifundamental of $\mathrm{U}(k) \times \mathrm{SU}(N)$ with gauge parameter $u_a$ and twisted flavour masses $\theta_j$. The second line entails the contribution of the adjoint chiral and the FI coupling. The physical FI parameter is parametrised by $t_2 - t_1$, as motivated by the brane realisation.

The massive supersymmetric vacua can be evaluated by using

$$-2\pi\mathrm{i}\partial_u \ell(u) = \log\left[ 2\sinh\left(-\mathrm{i}\pi u\right) \right] \quad \Longleftrightarrow \quad \partial_u \ell(u) = \frac{\mathrm{i}}{2\pi} \log\left[ x^{-\frac{1}{2}} - x^{\frac{1}{2}} \right] \tag{4.16}$$

for $x = e^{2\pi\mathrm{i}u}$, as in (4.9). One verifies straightforwardly

$$e^{2\pi\mathrm{i}\partial_{u_a}\widetilde{\mathcal{W}}_{\text{eff}}} = (-1)^\delta \frac{\epsilon_2}{\epsilon_1} \prod_{\substack{d=1 \\ d \neq a}}^{k} \frac{x_a q - x_d q^{-1}}{x_d q - x_a q^{-1}} \prod_{j=1}^{N} \frac{x_a - y_j q}{y_j - x_a q} \tag{4.17}$$

using the complex fugacities (4.8). Here, the additional sign $(-1)^\delta$ can introduced by shifting the fugacities $\frac{\epsilon_2}{\epsilon_1} \to (-1)^\delta \frac{\epsilon_2}{\epsilon_1}$ for the $\mathrm{U}(1)$ topological symmetry. This sign ambiguity was noted in [21, 42, 43]. Here, $\delta = k + N - 1$ is used.

**A-type quiver.** For the general class of $A$-type quivers (4.1), the Bethe Ansatz equations for the $s$-th node

$$(4.18)$$

are given by $(a = 1, \ldots, N_s)$

$$P_a^{(s)} = (-1)^{\delta_s} \frac{\epsilon_{s+1}}{\epsilon_s} \prod_{\substack{d=1 \\ d \neq a}}^{N_s} \frac{x_a^{(s)} q - x_d^{(s)} q^{-1}}{x_d^{(s)} q - x_a^{(s)} q^{-1}} \cdot \prod_{i=1}^{M_s} \frac{x_a^{(s)} - y_i^{(s)} q}{y_i^{(s)} - x_a^{(s)} q} \tag{4.19a}$$

$$\cdot \prod_{b=1}^{N_{s-1}} \frac{x_a^{(s)} - x_b^{(s-1)} q}{x_b^{(s-1)} - x_a^{(s)} q} \prod_{c=1}^{N_{s+1}} \frac{x_a^{(s)} - x_c^{(s+1)} q}{x_c^{(s+1)} - x_a^{(s)} q}$$

$$\delta_s = N_s + N_{s-1} + N_{s+1} + M_s - 1 \,. \tag{4.19b}$$

where the blue parts originate from the $\mathrm{U}(N_s)$ vector multiplet, red parts denote the $M_s$ fundamental hypermultiplets, and green parts are due to the bifundamental hypermultiplets between the $\mathrm{U}(N_s)$ gauge node and the adjacent gauge nodes. The black terms are the classical contributions from the FI-parameter and the associated sign-shift.

## 4.3  3d $\mathcal{N} = 2^*$ theories on $\Sigma_g \times S^1$

As a next step, one can place the resulting 2d $\mathcal{N} = (2,2)^*$ KK theory on a curved background, *i.e.* a Riemann surface $\Sigma_g$ of genus $g$. The curved background does not preserve all supersymmetries, but topological twisting [44] renders the situation manageable. There are two well-known possibilities: the $\mathcal{N} = (2,2)$ R-symmetry contains a vector and an axial U(1) symmetry. The $\mathrm{SO}(2)_L$ Lorentz symmetry of $\Sigma_g$ can be topologically twisted with either the axial or the vector U(1) R-symmetry. The A-twist denotes the twist of $\mathrm{SO}(2)_L$ with the axial U(1) R-symmetry such that the vector part is preserved. Conversely, the B-twist locks $\mathrm{SO}(2)_L$ and vector U(1) R-symmetry rotations such that the axial U(1) is preserved. As a result from the four original supercharges, only two become scalar supercharges after the twisting procedure. These scalar supercharges can be preserved on the curved background and are subsequently used for supersymmetric localisation of the partition functions, see for example [45–54].

From the 3d perspective, the Lorentz group of a Riemannian manifold is $\mathrm{SO}(3)_L \cong \mathrm{SU}(2)_L$, while the $\mathcal{N} = 4$ R-symmetry is $\mathrm{SO}(4)_R \cong \mathrm{SU}(2)_H \times \mathrm{SU}(2)_C$. Then there exist two distinct choices for topological twisting [55–57]:

- A-twist: The novel A-twisted symmetry group is $\mathrm{SU}(2)_A \times \mathrm{SU}(2)_C$ with $\mathrm{SU}(2)_A = \mathrm{diag}\,(\mathrm{SU}(2)_L, \mathrm{SU}(2)_H)$, which is the new Lorentz group after the twist. From the

original 8 supercharges, four become scalar supercharges with respect to $\mathrm{SU}(2)_A$. The preserved R-symmetry is $\mathrm{U}(1)_H \times \mathrm{SU}(2)_C$, while the $\mathrm{U}(1)_R$ symmetry of the $\mathcal{N} = 2$ subalgebra is generated by $R_A = 2j_H^3$ [56], see also [42] for example.

- B-twist: The symmetry group $\mathrm{SU}(2)_B \times \mathrm{SU}(2)_H$ defined by the new Lorentz group after twist $\mathrm{SU}(2)_B = \mathrm{diag}\,(\mathrm{SU}(2)_L, \mathrm{SU}(2)_C)$ leads to four scalar supercharges, with respect to $\mathrm{SU}(2)_B$. The preserved R-symmetry is $\mathrm{SU}(2)_H \times \mathrm{U}(1)_C$, while the $\mathrm{U}(1)_R$ symmetry of the $\mathcal{N} = 2$ subalgebra is generated by $R_B = 2j_C^3$. This is also known as Rozansky-Witten twist [55].

Both, A and B-twist, preserve 4 supercharges each, but not necessarily the same four. One can show that 2 supercharges are same in each set of four, such that these supercharges, preserved by both A and B-twist, are used for the localisation of the partition functions. Most importantly, these supercharges commute with the global symmetry $\mathrm{U}(1)_\eta = 2\,[\mathrm{U}(1)_H - \mathrm{U}(1)_C]$ with charge $Q_\eta = R_A - R_B = 2j_H^3 - 2j_C^3$. The A and B-twisted index is defined as [42, 43, 58, 59]

$$I_{g,A/B}(q, z_i) = \mathrm{Tr}_{\Sigma_g^{A/B}} \left( (-1)^F q^{Q_\eta} \prod_i z_i^{Q_i} \right) \quad \text{with } \mathrm{U}(1)_R \text{ charge } R = R_{A/B} \qquad (4.20)$$

where $z_i$ are fugacities for all global symmetries. Via supersymmetric localisation, the twisted indices reduce to a contour integral over the complexified Cartan subalgebra of the gauge group. This formulation is summarised in Appendix B. Remarkably, the integral expression is equivalent to evaluating a certain function on the set of Bethe roots, *cf.* (B.7)–(B.8).

## 4.4 Examples of twisted index computations

Having introduced the topologically twisted indices, written as sum over Bethe vacua, it is time to demonstrate the efficiency of rational $Q$-systems. From the gauge theory point of view [42], the A and B-twisted indices should agree with the Coulomb and Higgs branch Hilbert series, respectively. We use this as a consistency check for the rational $Q$-system. To be specific, consider $\mathrm{U}(k)$ SQCD with $N_f$ fundamental hypermultiplets. The Coulomb branch Hilbert series is known from [60], while the Higgs branch Hilbert series are, for example, given in [61]. We have verified the index results derived from solving the rational $Q$-system in the following cases: $k = 2$, $N_f = 4$; $k = 3$, $N_f = 6, 7, 8$; $k = 4$, $N_f = 8, 9, 10$; $k = 5$, $N_f = 10$. Some of the results are illustrated in Tables 4, 5, 6. Here we have set twisted masses $y_i$ and FI parameters $\epsilon_s$ to 1, corresponding to trivial inhomogeneities and twists in the language of spin chains, to emphasise the usefulness of rational $Q$-systems in these special situations[5]. The relation between A/B-twisted indices and Coulomb/Higgs

---

[5]Even though the rational $Q$-system can produce all the correct and physical Bethe roots when all the inhomogeneities and twists are trivial, some of the summands in the commputation of twisted indices become the 0/0 indefinite type. We regularise these summands by giving a very small deformation to one

branch Hilbert series is given by

$$I_{g=0,A}(q, z_i) = q^{-\dim_H \mathcal{C}} \cdot \text{HS}_C(q^{-2}, z_i)$$

$$I_{g=0,B}(q, z_i) = q^{-\dim_H \mathcal{H}} \cdot \text{HS}_H(q^{-2}, z_i)$$

(4.21)

assuming that $\text{HS}_{C/H}(t, z_i)$ is Hilbert series graded with respect to the half-integer spins of the third component of $\text{SU}(2)_{C/H}$ using the formal variable $t$. The $z_i$ are the fugacities of the Coulomb/Higgs branch isometries, and $\dim_H \mathcal{C}$, $\dim_H \mathcal{H}$ are the quaternionic Coulomb/Higgs branch dimensions, respectively.

We comment that the Bethe roots here are solved numerically from the rational $Q$-systems, and consequently the A/B-twisted indices of the gauge theories are also computed numerically. Even at this numerical stage, it is clear that the rational $Q$-systems outperforms BAE, as evident from Table 1. In a separate publication, we will use the algebraic geometrical methods to compute the twisted indices analytically, and the comparison with Hilbert series can be made exactly.

| precision | A-twisted index |
|---|---|
| 30 | 0.000287521564055931768363452209678234651728270155754218442756664857 |
| 40 | 0.000287521564055931768363451318442337444821446525532748735091985510 |
| 50 | 0.000287521564055931768363451318442337444821329677884380913431677921 |
| 60 | 0.000287521564055931768363451318442337444821329865556929710052909568 |
| $\text{HS}_C$ | 0.000287521564055931768363451318442337444821329865556929710052909428 |

(a)

| precision | B-twisted index $(/10^{-8})$ |
|---|---|
| 30 | 8.288254353220655138092936969618263612769886734780947 |
| 40 | 8.2882543532199696376393769549620971697496512058973677 |
| 50 | 8.2882543532199696376393769548731493242132365317167792 |
| 60 | 8.2882543532199696376393769548732936836186920889942863 |
| $\text{HS}_H$ $(/10^{-8})$ | 8.2882543532199696376393769548732936836186920889932196 |

(b)

**Table 3**. A and B-twisted index of 3d $\mathcal{N} = 4$ U(2) SQCD with $N_f = 4$ hypermultiplet computed by summing up Bethe roots solved from the Q-systems, compared with Coulomb and Higgs branch Hilbert series $\text{HS}_{C/H}$, respectively. Both twisted masses $y_j$ and FI parameters $\epsilon_s$ are set to 1, and we choose the real mass $q = 59$. "precision" is the working precision to solve numerically the Q-systems. Underlined are matching digits with the Hilbert series.

---

of the twists.

| precision | twisted A-index ($/10^{-6}$) |
|---|---|
| 30 | 4.8732468488677554916247345325593390724774052367581890 |
| 40 | 4.8732468488677554918004120386636296497335115841627159 |
| 50 | 4.8732468488677554918004120386112808487038322399706690 |
| 60 | 4.8732468488677554918004120386112808485997596123091081 |
| HS$_C$ ($/10^{-6}$) | 4.8732468488677554918004120386112808485997596123108992 |

(a)

| precision | twisted B-index ($/10^{-16}$) |
|---|---|
| 50 | 1.1659982057780843613099912930687872041142455898 |
| 60 | 1.1659982057766504232087694161792776514947491793 |
| 70 | 1.1659982057766504232087915341756647902797499693 |
| 80 | 1.1659982057766504232087915341756537366375069393 |
| HS$_H$ ($/10^{-16}$) | 1.1659982057766504232087915341756537366375082331 |

(b)

**Table 4**. A and B-twisted index of 3d $\mathcal{N} = 4$ U(3) SQCD with $N_f = 6$ hypermultiplet computed by summing up Bethe roots solved from the $Q$-systems, compared with the Coulomb and Higgs branch Hilbert series HS$_{C/H}$, respectively. Twisted masses $y_j$ and FI parameters $\epsilon_s$ are set to 1, and the real mass is chosen $q = 59$.

| precision | twisted A-index ($/10^{-8}$) |
|---|---|
| 40 | 8.2597404218099800852151904168033381652758692330277150858865873074 |
| 50 | 8.2597404218099800852404504832303660131368187458206676411560614866 |
| 60 | 8.2597404218099800852404504832303660289741112721141419574556209987 |
| 70 | 8.2597404218099800852404504832303660289741112721141416671652178833 |
| HS$_C$ ($/10^{-8}$) | 8.2597404218099800852404504832303660289741112721141416671652164055 |

(a)

| precision | twisted B-index ($/10^{-29}$) |
|---|---|
| 110 | 4.7230943236941808279598061464232253451472514625969858470737 |
| 120 | 4.7230943236941808279598061464234517300990118052257989496630 |
| 130 | 4.7230943236941808279598061464234517055465812280518352189910 |
| 140 | 4.7230943236941808279598061464234517055465812280518342351030 |
| HS$_H$ ($/10^{-29}$) | 4.7230943236941808279598061464234517055465812280518342354282 |

(b)

**Table 5**. A and B-twisted index of 3d $\mathcal{N} = 4$ U(4) SQCD with $N_f = 8$ hypermultiplets computed by summing up Bethe roots solved from the $Q$-systems, compared with the Coulomb and Higgs branch Hilbert series HS$_{C/H}$, respectively. Twisted masses $y_j$ and FI parameters $\epsilon_s$ are set to 1, and the real mass is chosen $q = 59$.

| precision | twisted A-index ($/10^{-9}$) |
|---|---|
| 70 | 1.3999560036966068050646270947749620937952226021717715400137605519061684 |
| 80 | 1.3999560036966068050646074052335646207373112302059669718203954234379008 16 |
| 90 | 1.3999560036966068050646074052335646207373088342395776258223680243772272 17 |
| 100 | 1.3999560036966068050646074052335646207373088342395776258223678170608474 32 |
| HS$_C$ ($/10^{-9}$) | 1.3999560036966068050646074052335646207373088342395776258223678170608472 31 |

(a)

| precision | twisted B-index ($/10^{-45}$) |
|---|---|
| 100 | 5.5087020127965160145995025812227782446094945377714329 02 |
| 110 | 5.5087020128708323562234075516117939670947763756016333 73 |
| 120 | 5.5087020128708323562234075516088493198467448632147672 87 |
| 130 | 5.5087020128708323562234075516088493198467447364543599 12 |
| HS$_H$ ($/10^{-45}$) | 5.5087020128708323562234075516088493198467447364542514 68 |

(b)

**Table 6**. A and B-twisted index of 3d $\mathcal{N} = 4$ U(5) SQCD with $N_f = 10$ hypermultiplets computed by summing up Bethe roots solved from the $Q$-systems, compared with the Coulomb and Higgs branch Hilbert series HS$_{C/H}$, respectively. Twisted masses $y_j$ and FI parameters $\epsilon_s$ are set to 1, and the real mass is chosen $q = 59$.

## 5 Higgsing $Q$-systems

Under Bethe/Gauge correspondence, 3d $\mathcal{N} = 4$ quiver gauge theories are in one-to-one correspondence to BAE/$Q$-system labelled by the same quiver. Supersymmetric gauge theories have rich structures and different theories can be related to each other by various mechanisms. Due to the correspondence between quiver gauge theories and rational $Q$-systems, operations on one side should be reflected on the other.

One important class of relations comes from the Higgs mechanism. Given a 3d $\mathcal{N} = 4$ linear quiver gauge theory $T_\rho^\sigma[\mathrm{SU}(n)]$ as in (4.1), the Higgs mechanism allows for a rich phase structure. As it is well-known, the moduli space of vacua splits into roughly three distinct types of branches: (i) the Higgs branch, where only hypermultiplet scalars acquire a non-trivial VEV, (ii) the Coulomb branch, parametrised by VEVs of the vector multiplets scalars, and (iii) mixed branches. Consequently, there exist the corresponding three types of Higgs transitions.

In the language of BAE, Higgsing is an operation which reduces the number of Bethe roots, either by fixing some of the Bethe roots at values related to the inhomogeneities, or by taking them to infinity. As we will see, in the $Q$-system, Higgsing corresponds to the operations which reduces the number of Bethe roots while *keeping the the number of boxes fixed*. There are two ways to achieve this, one is changing boundary conditions and the other is moving boxes around. Intriguingly, they correspond to Higgs branch and Coulomb

branch Higgsing respectively.

## 5.1 Higgs branch Higgsing: gauge theory

A generic gauge-invariant Higgs branch operator can be constructed from any path that starts and ends in some flavour node. For example, Figure 5a shows a typical case in the brane system. Suppose one has chosen flavour nodes $s$ and $r$, in order to open up a Higgs branch direction between a D5 brane in the $s$-th interval and a D5 brane in the $r$-th interval, one needs to align the $x^{3,4,5}$ positions of the following branes:

- The D5 labelled by $\theta_a^{(s)} \sim m_a^{(s)}$ needs to align with a D3 brane, *i.e.* one tunes the vector multiplet scalar $u_{N_s}^{(s)} \sim \sigma_{N_s}^{(s)}$.
- In the adjacent interval on the right-hand side, a single D3 brane needs to align with the adjusted D3 brane in the $s$-th interval, *i.e.* $u_{N_{s+1}}^{(s+1)}$ has to be tuned.
- This alignment of a single D3 brane continues for all intervals $j \in \{s+1, \ldots, r\}$.
- Lastly, the position of the D5 brane, labelled by $\theta_b^{(r)} \sim m_b^{(r)}$, needs to align with the position of the D3, which corresponds to the vector multiplet scalar $\sigma_{N_r}^{(r)} \subset u_{N_r}^{(r)}$.

Once all these branes are aligned, the D3 ending on the NS5s can join to form a single D3 that spans from the left NS5 in the $s$-th interval to the right NS5 in the $r$-th interval. Since this single D3 intersects the two D5 branes, the D3 can split on the D5 and the resulting D3 segment is free to move along the D5 branes. This realises the Higgs branch Higgsing of the gauge invariant displayed in the quiver in Figure 5b, because the motion of D3 branes suspended between D5 precisely are the Higgs branch directions. The residual D3 brane segments, which are suspended between an NS5 and a D5, have no dynamical degrees of freedom and can be eliminated moving the D5 through the NS5, due to brane annihilation. The resulting theory is shown in Figure 5c.

While the 3d $\mathcal{N} = 4$ brane systems provides a natural intuition for which parameter need to be adjusted, the precise choices need to take the $\mathcal{N} = 2^*$ deformation $\eta$ into account. One finds [21]

$$\theta_a^{(s)} = u_{N_s}^{(s)} + \frac{\eta}{2}, \quad u_{N_s}^{(s)} = u_{N_{s+1}}^{(s+1)} + \frac{\eta}{2}, \quad \ldots, \quad u_{N_{r-1}}^{(r-1)} = u_{N_r}^{(r)} + \frac{\eta}{2}, \quad u_{N_r}^{(r)} = \theta_b^{(r)} + \frac{\eta}{2} \quad (5.1)$$

and the Bethe Ansatz equation of the theory in Figure 5b reduce to the BAE of the Higgsed theory in Figure 5c upon this tuning of variables, due to telescopic cancellation. See Appendix A.1 for details.

Strictly speaking, it is not necessary to consider such a general Higgs branch Higgsing, as it is sufficient to consider the two minimal Higgsing transitions [62, 63]:

1. $\underline{A_{M-1} \text{ transitions}}$: for a gauge node with $(N_i \geq 1, M_i \geq 2)$ one specialises the transition in Figure 4 to $r = s = i$. After the transition, the gauge label $N_i$ and the flavour labels $M_{i-1}, M_i, M_{i+1}$ are changed respectively to $N_i - 1, M_{i-1} + 1, M_i - 2, M_{i+1} + 1$, while the other gauge/flavour labels are not changed.

This is called $A_{M-1}$ transition with $M = M_i$.

2. $\underline{a_k \text{ transition}}$: Suppose there exists a sequence of nodes such that $(N_s \geq 1, M_s = 1)$, $(N_r \geq 1, M_r = 1)$ with $s < r$, and $(N_i \geq 1, M_i = 0)$ for all $s < i < r$. A VEV to the gauge invariant stretched from $M_s$ to $M_r$ leads to a Higgs mechanism that is known as $A_k$ transition with $k = r - s + 1$. After the transition, the gauge labels $N_s, N_{s+1}, \ldots, N_r$ as well as the flavor labels $M_s, M_r$ are reduced by one, and the flavour labels $M_{s-1}, M_{r+1}$ are increased by one. The remaining gauge/flavour labels are unchanged.

These minimal transitions, also known as Kraft-Procesi transitions [62], are sufficient to describe any Higgsing via a sequence of elementary steps. We note that the balancing conditions $e_i$ as well as the partition $\boldsymbol{\rho}$ are not affected by the Higgs branching Higgsing.

## 5.2  Higgs branch Higgsing: $Q$-system

In the previous sections, we have seen Higgs branch Higgsing from gauge theory and at the level of BAE. Now let us see the correspondence in the rational $Q$-system. We consider a Higgs branch Higgsing along a path from flavor node $M_s$ to $M_r$ with $r > s$. Under this operation, the Dynkin diagram labelled by $\vec{M} = (M_1, \ldots, M_{\ell-1})$, $\vec{N} = (N_1, \ldots, N_{\ell-1})$ becomes

$$\vec{M}' = (M_1, \ldots, M_{s-2}, M_{s-1} + 1, M_s - 1, M_{s+1}, \ldots, M_{r-1}, M_r - 1, M_{r+1} + 1, M_{r+2}, \ldots, M_{\ell-1}),$$
$$\vec{N}' = (N_1, \ldots, N_{s-1}, N_s - 1, N_{s-1} - 1, \ldots, N_r - 1, N_{r+1}, \ldots, N_{\ell-1}).$$

From $\vec{M}, \vec{N}$ and $\vec{M}', \vec{N}'$, we can compute the corresponding Young tableaux by

$$\lambda_a = (N_{a-1} - N_a) + (M_a + M_{a+1} + \ldots + M_{\ell-1}) \tag{5.2}$$

with $N_0 = 0$ and $N_\ell = 0$. Let us denote the Young tableaux by $\boldsymbol{\lambda}$ and $\boldsymbol{\lambda}'$. It is clear that $\lambda_a = \lambda'_a$ for $a = 1, 2, \ldots, s - 2$. For $a = s - 1$, we have

$$\lambda'_{s-1} = (N'_{s-2} - N'_{s-1}) + (M'_{s-1} + M'_s + \ldots + M'_{\ell-1}) \tag{5.3}$$
$$= (N_{s-2} - N_{s-1}) + (M_{s-1} + 1 + M_s - 1 + \ldots + M_{\ell-1}) = \lambda_{s-1}.$$

Similarly, we can check that $\lambda'_a = \lambda_a$ for all $a = 1, 2, \ldots, \ell - 1$. Therefore, we see that although $\vec{M}', \vec{N}'$ and $\vec{M}, \vec{N}$ are different, the corresponding Young tableaux is the same.

The case $r = \ell - 1$ is special. In this case, naively we have

$$\vec{M}' = (M_1, \ldots, M_{s-2}, M_{s-1} + 1, M_s - 1, M_{s+1}, \ldots, M_{\ell-1} - 1), \tag{5.4}$$
$$\vec{N}' = (N_1, \ldots, N_{s-1}, N_s - 1, N_{s-1} - 1, \ldots, N_{\ell-1} - 1)$$

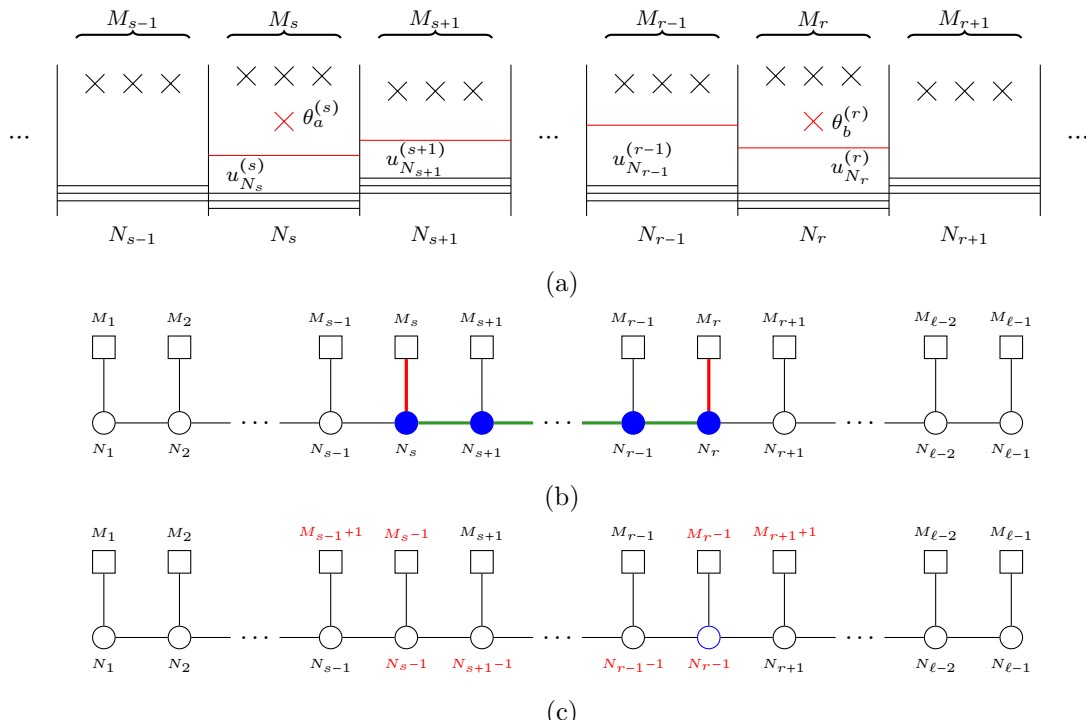

(a)

(b)

(c)

**Figure 5**. Higgs branch Higgsing. (a) displays which branes need to align with each other to open up a Higgs branch direction. (b) shows the corresponding gauge invariant operator in the quiver as path from flavour node $M_s$ to $M_r$. The resulting theory after the Higgs transition is shown in (c).

which leads to

$$\lambda'_a = \lambda_a - 1, \qquad a = 1, \dots, \ell. \tag{5.5}$$

Notice that in (5.4), the total number of inhomogeneities is reduced by 1. Taking into account this missing inhomogeneity, we consider

$$\vec{M}'' = (M_1, \dots, M_{s-2}, M_{s-1} + 1, M_s - 1, M_{s+1}, \dots, M_{\ell-1} - 1, 1), \tag{5.6}$$

$$\vec{N}'' = (N_1, \dots, N_{s-1}, N_s - 1, N_{s-1} - 1, \dots, N_{\ell-1} - 1, 0).$$

This corresponds to adding a floating flavor node attached to the $\ell$-th empty gauge node. The corresponding Young tableaux is

$$\lambda''_a = \lambda_a, \qquad a = 1, \dots, \ell, \qquad \text{and} \quad \lambda''_{\ell+1} = 0 \tag{5.7}$$

which is the same Young tableaux before Higgsing. This again confirms that the Higgs branch Higgsing does not change the Young tableaux of the rational $Q$-system.

Another way to see the Young tableaux does not change is to notice that the numbers

of boxes are related to the balancing conditions

$$\lambda_s = e_s + e_{s+1} + \ldots e_{\ell-1} + e_\ell, \tag{5.8}$$

where we have taken into account the $\ell$-th empty gauge node, and the latter are not changed under Higgs branch Higgsing.

### 5.2.1    Examples

In this subsection, we consider examples of Higgsing for $A_3$-type rational $Q$-system, as is shown in Figure 6 For simplicity, we consider the XXX-type model. All the Dynkin

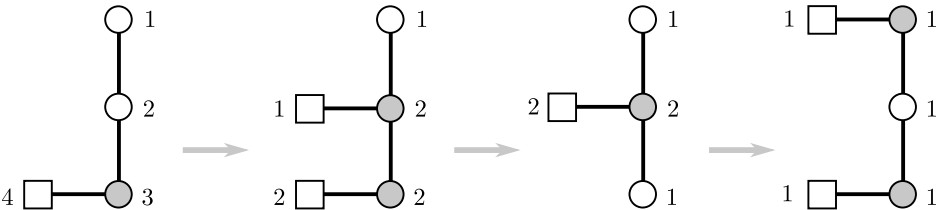

**Figure 6**. Higgs branch Higgsing for $A_3$-type rational $Q$-system. All quivers correspond to the Young tableaux $\boldsymbol{\lambda} = (1, 1, 1, 1)$

diagrams in Figure 6 corresponds to the Young tableaux $\boldsymbol{\lambda} = (1, 1, 1, 1)$. We see that from left to right, the numbers of Bethe roots are reducing. This is due to the different boundary conditions of the rational $Q$-systems. The corresponding boundary conditions for the four Dynkin diagrams $\mathbb{Q}_{a,0} = f_a(u) Q_a(u)$ are given by

1. For $\vec{M} = (4, 0, 0)$ and $\vec{N} = (3, 2, 1)$, we have

$$f_0(u) = \prod_{j=1}^{4} \left(u - \theta_j^{(1)}\right), \qquad f_1(u) = 1, \qquad f_2(u) = 1, \qquad f_3(u) = 1 \tag{5.9}$$

   and

$$Q_0(u) = 1, \tag{5.10}$$
$$Q_1(u) = u^3 + c_2^{(1)} u^2 + c_1^{(1)} u + c_0^{(1)},$$
$$Q_2(u) = u^2 + c_1^{(2)} u + c_0^{(2)},$$
$$Q_3(u) = u + c_0^{(3)}.$$

   The zero remainder conditions have 6 variables.

2. For $\vec{M} = (2, 1, 0)$ and $\vec{N} = (2, 2, 1)$, we have

$$f_0(u) = \left(u - \theta_1^{(1)}\right)\left(u - \theta_2^{(1)}\right)\left(u - \theta_1^{(2)} - \tfrac{i}{2}\right)\left(u - \theta_1^{(2)} + \tfrac{i}{2}\right), \tag{5.11}$$

$$f_1(u) = \left(u - \theta_1^{(2)}\right), \qquad f_2(u) = f_3(u) = 1,$$

and

$$Q_0(u) = 1, \tag{5.12}$$
$$Q_1(u) = u^2 + c_1^{(1)} u + c_0^{(1)},$$
$$Q_2(u) = u^2 + c_1^{(2)} u + c_0^{(2)},$$
$$Q_3(u) = u + c_0^{(3)}.$$

The zero remainder conditions now have 5 variables, and we have 1 less Bethe root.

3. For $\vec{M} = (0, 2, 0)$ and $\vec{N} = (1, 2, 1)$, we have

$$f_0(u) = \left(u - \theta_1^{(2)} - \tfrac{i}{2}\right)\left(u - \theta_2^{(2)} - \tfrac{i}{2}\right)\left(u - \theta_1^{(2)} + \tfrac{i}{2}\right)\left(u - \theta_2^{(2)} + \tfrac{i}{2}\right), \tag{5.13}$$
$$f_1(u) = \left(u - \theta_1^{(2)}\right)\left(u - \theta_2^{(2)}\right), \qquad f_2(u) = f_3(u) = 1.$$

and

$$Q_0(u) = 1, \tag{5.14}$$
$$Q_1(u) = u + c_0^{(1)},$$
$$Q_2(u) = u^2 + c_1^{(2)} u + c_0^{(2)},$$
$$Q_3(u) = u + c_0^{(3)}.$$

The zero remainder conditions have 4 variables now.

4. For $\vec{M} = (1, 0, 1)$ and $\vec{N} = (1, 1, 1)$, we have

$$f_0(u) = \left(u - \theta_1^{(3)} - i\right)\left(u - \theta_1^{(3)}\right)\left(u - \theta_1^{(3)} + i\right)\left(u - \theta_1^{(1)}\right), \tag{5.15}$$
$$f_1(u) = \left(u - \theta_1^{(3)} - \tfrac{i}{2}\right)\left(u - \theta_1^{(3)} + \tfrac{i}{2}\right),$$
$$f_2(u) = \left(u - \theta_1^{(3)}\right), \qquad f_3(u) = 1.$$

and

$$Q_0(u) = 1, \tag{5.16}$$
$$Q_1(u) = u + c_0^{(1)},$$
$$Q_2(u) = u + c_0^{(2)},$$
$$Q_3(u) = u + c_0^{(3)}.$$

The zero remainder condition has 3 variables.

### 5.2.2 A heuristic explanation

To have a better intuition about Higgs branch Higgsing in the spin chain language, let us give a heuristic explanation using periodic rank-1 XXX spin chain. The Bethe roots enter the spin chain via the Bethe ansatz. For a length-$M$ spin chain with $N$ magnons whose rapidities are given by the $N$ Bethe roots, we have inhomogeneities $\{\theta_a\}$ and Bethe roots $\{u_j\}$. The BAE reads

$$\prod_{a=1}^{M} \frac{u_j - \theta_a + \frac{i}{2}}{u_j - \theta_a - \frac{i}{2}} = \prod_{k \neq j}^{N} \frac{u_j - u_k + i}{u_j - u_k - i}, \qquad j = 1, 2, \ldots, N. \tag{5.17}$$

The Higgs branch Higgsing in the spin chain language corresponds to fixing one of the Bethe root, say $u_1$, to a value corresponding to one of the inhomogeneities, say $\theta_1$. We set

$$u_1 = \theta_1 - \tfrac{i}{2}. \tag{5.18}$$

At the same time, to avoid divergences, we need to set another inhomogeneity, say $\theta_2$ to be

$$\theta_2 = \theta_1 - i. \tag{5.19}$$

Making this choice, the BAE for $u_1$ trivializes because it is already fixed. For the rest of the rapidities $u_j$, $j = 2, \ldots, N$, the BAE becomes

$$\frac{u_j - \theta_1 + \frac{i}{2}}{u_j - \theta_1 - \frac{i}{2}} \frac{u_j - \theta_1 + \frac{3i}{2}}{u_j - \theta_1 + \frac{i}{2}} \prod_{a=3}^{M} \frac{u_j - \theta_a + \frac{i}{2}}{u_j - \theta_a - \frac{i}{2}} = \frac{u_j - \theta_1 + \frac{3i}{2}}{u_j - \theta_1 + \frac{i}{2}} \prod_{k \neq 1, j}^{N} \frac{u_j - u_k + i}{u_j - u_k - i}. \tag{5.20}$$

Cancelling common factors from both sides leads to

$$\prod_{a=3}^{M} \frac{u_j - \theta_a + \frac{i}{2}}{u_j - \theta_a - \frac{i}{2}} = \prod_{k \neq 1, j}^{N} \frac{u_j - u_k + i}{u_j - u_k - i}, \tag{5.21}$$

which is the BAE of a spin chain of length $M - 2$ with $N - 1$ magnons.

Heuristically, the physical picture is as follow. In coordinate Bethe ansatz, Bethe roots are rapidities of a kind of particles called magnons. Each time a magnon with rapidity $u_j$ passes site-$a$ with inhomogeneity $\theta_a$, it picks up a phase

$$e^{i \Delta p_a(u_j)} = \frac{u_j - \theta_a + \frac{i}{2}}{u_j - \theta_a - \frac{i}{2}}. \tag{5.22}$$

Making the choice (5.18), (5.19), we have

$$e^{i \Delta p_1(u_1)} = 0, \qquad e^{i \Delta p_2(u_1)} = \infty. \tag{5.23}$$

Effectively, the first and second sites become infinite high barrier and the magnon with rapidity $u_1$ is trapped between them and can no longer move freely. For magnons with other rapidities, the combined effect of the choice (5.18), (5.19) is trivial

$$e^{i\Delta p_1(u_j)} e^{i\Delta p_2(u_j)} S(u_j, u_1) = \frac{u_j - \theta_1 + \frac{i}{2}}{u_j - \theta_1 - \frac{i}{2}} \frac{u_j - \theta_1 + \frac{3i}{2}}{u_j - \theta_1 + \frac{i}{2}} \frac{u_j - \theta_1 - \frac{i}{2}}{u_j - \theta_1 + \frac{3i}{2}} = 1 \,, \qquad (5.24)$$

where $S(u_j, u_1)$ is the scattering phase between two magnons with rapidities $u_j$ and $u_1$, which for the XXX chain is given by

$$S(u_j, u_1) = \frac{u_j - u_1 - i}{u_j - u_1 + i} \,. \qquad (5.25)$$

## 5.3 Coulomb branch Higgsing: gauge theory

Besides turning on VEVs for scalar in the hypermultiplet, also vector multiplet scalars can acquire a non-trivial VEV. The simplest Coulomb branch Higgsing is realised by partial break $U(N_s) \to U(N_s - 1)$ realised by a VEV to, say, $\sigma_{N_s}^{(s)}$, which is then taken to infinity. In the brane system, a single D3 brane from the stack of $N_s$ D3s in between the $s$-th and $(s+1)$-th NS5 is moved off to infinity.

For later purposes, it is necessary to consider a more fundamental Coulomb branch Higgs transition, displayed in Figure 7. The significance of this transition stems from the fact that there exist two fundamental Coulomb branch Higgsing transitions for the $A$-type quiver considered here. To approach the Coulomb branch deformations, one can follow the minimal Higgs branch transitions and revert the logic. That means: The signal for minimal Higgs branch transitions are the presence of flavour nodes (in general non-abelian factors in $G_H$), while the balance of the gauge nodes is preserved in any Higgs branch transitions. Thus, for the Coulomb branch Higgsing the "smoking gun" is the presence of balanced gauge nodes (as these lead to enhance non-abelian factors on $G_C$), while the flavour symmetry is preserved. Consequently, the minimal Coulomb branch transitions are given by:

1. Dual of $A_{M-1}$ transition: Recall that in the $A_{M-1}$ transition, a single gauge node had a $M$ flavour node attached. In the brane system, this translates to $M$ D5 branes in the same NS5 brane interval, and these D5s have identical linking number $L_i = \#\text{D3}_{\text{LHS}} - \#\text{D3}_{\text{RHS}} + \#\text{NS5}_{\text{LHS}}$. Upon S-duality, the D5s become NS5s. The NS5 linking number are related to balance of gauge nodes in the mirror theory (see for instance [21, 37]), $i.e.$ $L_i - L_{i+1} = e_i^\vee = M_i^\vee + N_{i-1}^\vee + N_{i+1}^\vee - 2N_i^\vee$. Here $(M_i^\vee, N_i^\vee)$ are the integers defining the mirror theory. It follows that $M$ consecutive NS5 branes with identical linking numbers imply $M - 1$ consecutive gauge groups with vanishing balance $e_i^\vee = 0$. Hence, the mirror should have a connected set of $M - 1$ balanced nodes.

Suppose there exists a connected sequence of $M - 1$ balanced gauge nodes $U(N_s)$ for $s = r, \ldots, r + M - 2$. Then, the Coulomb branch minimal transition leads to a breaking $U(N_s) \to U(N_s - 1)$ for all $s = r, \ldots, r + M - 2$, while all other gauge and flavour nodes are unaffected. As far as the balances are concerned, $e_r$ and $e_{M-2}$ increase by one, and $e_{r-1}$ and $e_{M-1}$ of the connected nodes reduce by one.

2. Dual of $a_k$ transition: recall that the $a_k$ transition appeared between two single flavours at different gauge nodes. Hence, non-abelian flavour node are not required. Without loss of generality, the flavours are at node $s$ and $r$ such that $r > s$ and $k = r - s + 1$. The two D5 branes differ in their linking numbers as follows: $L_s = \#\text{NS5}_{\text{LHS}}(\text{at } s)$, $L_r = \#\text{NS5}_{\text{LHS}}(\text{at } r)$, but $\#\text{NS5}_{\text{LHS}}(\text{at } r) = \#\text{NS5}_{\text{LHS}}(\text{at } s) + r - s + 1$. This is because the D5 in the $r$-th interval perceives $r - s + 1$ more NS5 branes to its left-hand side compared to the D5 in the $s$-th interval. Upon S-duality, the D5s becomes N5s and their difference in linking number translates to the balance of the gauge theory living on the world-volume of the D3s stretched between them. One finds $e^\vee = L_r - L_s = r - s + 1 = k$. Therefore, in the mirror, this transition is not associated with balanced nodes, but with a node of balance $k$.

Consider a gauge node $U(N_s)$ that is good, but not balanced, $i.e.$ $e_s > 0$ and the connected adjacent nodes also have strictly positive balance $e_i > 0$. Then, a minimal Coulomb branch transitions is simply a breaking of a $U(N_s) \to U(N_s - 1)$, where any $s$ that satisfies the assumptions. This implies that the balance $e_i$ of the connected nodes is reduced by 1, while the balance $e_s$ of node $s$ is increased by 2.

Note that neither of the two scenarios can change the partition $\sigma$, while the partition $\rho$ related to the balancing conditions $e_i$ is changed.

Returning to the scenario of Figure 7, this Higgsing can be realised in the BAE by the following procedure: Firstly, for each partially broken gauge group $U(N_j)$, a single complex

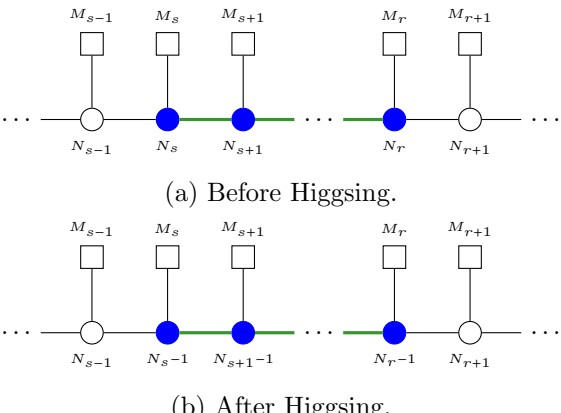

(a) Before Higgsing.

(b) After Higgsing.

**Figure 7**. Coulomb branch Higgsing.

gauge fugacity, say, $x_{N_j}^{(j)}$ is selected. These need to be aligned

$$x_{N_s}^{(s)} = \ldots = x_{N_j}^{(j)} = \ldots = x_{N_r}^{(r)} \equiv \chi \to \infty \qquad (5.26a)$$

and send to infinity simultaneously. In addition, the transition is only meaningful if the $\epsilon_j$ parameter of the affected gauge nodes take specific values

$$\epsilon_{s+1} = -q\epsilon_s, \qquad \epsilon_{j+1} = \epsilon_j \text{ for } s < j < r, \qquad \epsilon_{r+1} = -q\epsilon_r. \qquad (5.26b)$$

Upon this tuning of parameters, the BAE for the theory in Figure 7a reduce to the BAE of the theory in Figure 7b. The detailed analysis is delegated to Appendix A.2.

## 5.4 Coulomb branch Higgsing: $Q$-system

Now we consider the Coulomb branch Higgsing. After the Coulomb branch Higgsing, the Dynkin diagram $\vec{M} = (M_1, \ldots, M_{\ell-1})$, $\vec{N} = (N_1, \ldots, N_{\ell-1})$ becomes $\vec{M} = (M_1, \ldots, M_{\ell-1})$, $\vec{N}' = (N_1', \ldots, N_{\ell-1}')$ where the numbers $\vec{M}$ do not change. If we choose the path of the Higgsing along a path from flavor node $s$ to $r$, the numbers $\vec{N}'$ become

$$N_a' = \begin{cases} N_a - 1, & s \le a \le r \\ N_a, & \text{others} \end{cases} \qquad (5.27)$$

Recalling that

$$\lambda_a = (N_{a-1} - N_a) + (M_a + M_{a+1} + \ldots + M_{L-1}), \qquad (5.28)$$

we find that the Young tableaux $\vec{\lambda}' = (\lambda_1', \ldots, \lambda_\ell')$ becomes

$$\lambda_s' = \lambda_s + 1, \qquad \lambda_{r+1}' = \lambda_{r+1} - 1 \qquad (5.29)$$

where $\lambda_a' = \lambda_a$ for the rest $a$. This amounts to moving a box from the $r + 1$-th row to the $s$-th row. The total number of boxes is the same. At the same time, the boundary condition $f_a(u)$ is not changed.

### 5.4.1 Examples

Let us now consider examples of the Coulomb branch Higgsing. More concretely, we consider the examples in Figure 8. We present the rational $Q$-systems from left to right.

1. For $\vec{M} = (4, 0, 0)$ and $\vec{N} = (3, 2, 1)$, we have

$$f_0(u) = \prod_{j=1}^{4}(u - \theta_j), \qquad f_1(u) = f_2(u) = f_3(u) = 1. \qquad (5.30)$$

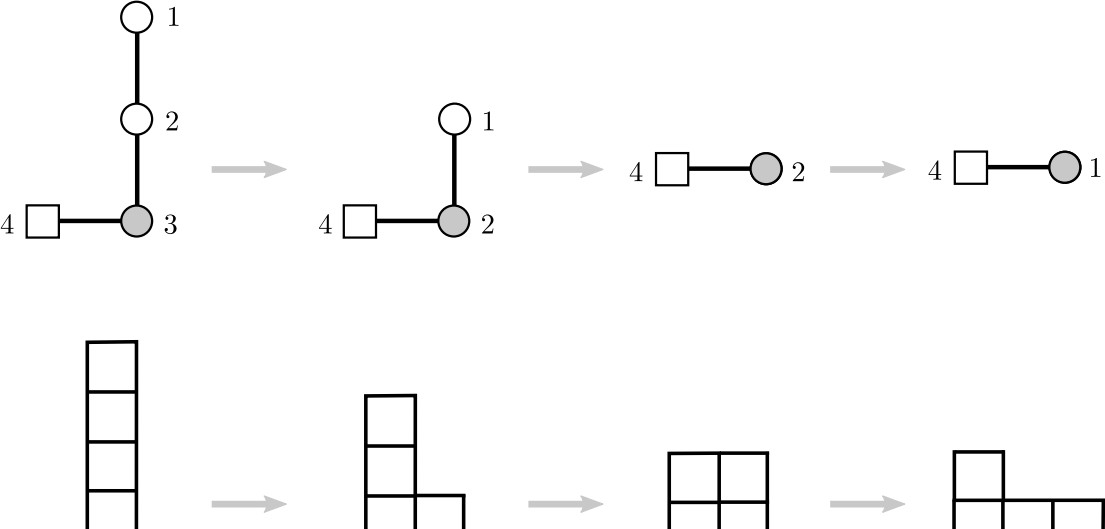

**Figure 8**. Coulomb branch Higgsing for $A_3$-type rational $Q$-system.

and

$$Q_0(u) = 1\,, \tag{5.31}$$
$$Q_1(u) = u^3 + c_2^{(1)}u^2 + c_1^{(1)}u + c_0^{(1)}\,,$$
$$Q_2(u) = u^2 + c_1^{(2)}u + c_0^{(2)}$$
$$Q_3(u) = u + c_0^{(3)}\,.$$

2. For $\vec{M} = (4, 0)$ and $\vec{N} = (2, 1)$, we have

$$f_0(u) = \prod_{j=1}^{4}(u - \theta_j), \qquad f_1(u) = f_2(u) = 1\,, \tag{5.32}$$

and

$$Q_0(u) = 1\,, \tag{5.33}$$
$$Q_1(u) = u^2 + c_1^{(1)}u + c_0^{(1)}\,,$$
$$Q_2(u) = u + c_0^{(2)}\,.$$

3. For $\vec{M} = (4)$ and $\vec{N} = (2)$, we have

$$f_0(u) = \prod_{j=1}^{4}(u - \theta_j), \qquad f_1(u) = 1\,, \tag{5.34}$$

and

$$Q_0(u) = 1, \tag{5.35}$$
$$Q_1(u) = u^2 + c_1^{(1)} u + c_0^{(1)}.$$

4. For $\vec{M} = (4)$ and $\vec{N} = (1)$,

$$f_0(u) = \prod_{j=1}^{4} (u - \theta_j), \qquad f_1(u) = 1, \tag{5.36}$$

and

$$Q_0(u) = 1, \tag{5.37}$$
$$Q_1(u) = u + c_0^{(1)}.$$

We see that from left to right, the total numbers of Bethe roots are reducing, while the number of boxes are fixed. At the same time, the boundary conditions are not modified.

## 6  Mirror symmetry

As we have discussed before, there is deep connection between gauge theories and Bethe ansatz. 3d $\mathcal{N} = 4$ has been crucial in understanding dualities in supersymmetric gauge theories. Most notably, they provide the first examples of 3D mirror symmetry. The incarnation of mirror symmetry at the level of Bethe ansatz equation has been discussed in the literature [21] under the name of bispectral duality. In this section, we discuss the meaning of mirror symmetry for rational $Q$-system. In addition, we give explicit examples for the duality. We shall see that mirror symmetry is more naturally described in the $Q$-system language.

**Partitions**  To start with, the origin of the Young tableaux of $Q$-system might seem a bit mysterious from the spin chain point of view. However, it emerges very naturally from quiver gauge theories. To see this, let us consider quiver gauge theories $T_{\boldsymbol{\rho}}^{\boldsymbol{\sigma}}[\mathrm{SU}(n)]$. These theories are labelled by two partitions $\boldsymbol{\rho}$ and $\boldsymbol{\sigma}$. Both $\boldsymbol{\rho}$ and $\boldsymbol{\sigma}$ are partitions of the integer $n$ given in (6.1). The partition $\boldsymbol{\rho}$ (represented by a Young tableaux) is identified with the Young tableaux of the rational $Q$-system. The total number of boxes is

$$n = \sum_{a=1}^{\ell} \rho_a = \sum_{a=1}^{\ell-1} a M_a. \tag{6.1}$$

recall that $\rho_a$ is given by

$$\rho_a = (N_{a-1} - N_a) + (M_a + M_{a+1} + \ldots + M_{\ell-1}). \tag{6.2}$$

We have seen in the previous sections that the Young tableaux alone is not sufficient to specify the theory labelled by $\vec{M}, \vec{N}$. We still have the freedom to choose different boundary conditions, which can be fixed by the other partition $\boldsymbol{\sigma}$. Let us denote the transpose of $\boldsymbol{\sigma}$ by

$$\boldsymbol{\sigma}^{\mathrm{T}} = (\hat{\sigma}_1, \hat{\sigma}_2, \ldots, \hat{\sigma}_\ell) \tag{6.3}$$

The elements $\hat{\sigma}_j$ are related to $\vec{M}$ by

$$M_j = \hat{\sigma}_j - \hat{\sigma}_{j+1}. \tag{6.4}$$

With the additional constraint

$$\sum_{j=1}^{\ell} \hat{\sigma}_j = n = \sum_{a=1}^{\ell-1} a M_a, \tag{6.5}$$

we find that

$$\hat{\sigma}_a = M_a + M_{a+1} + \ldots + M_{\ell-1}, \quad a = 1, \ldots, \ell - 1, \tag{6.6}$$
$$\hat{\sigma}_\ell = 0.$$

The partition $\boldsymbol{\sigma}^{\mathrm{T}}$ is related to the boundary conditions $f_a(u)$ by

$$\hat{\sigma}_a = \deg(f_{a-1}) - \deg(f_a), \qquad \deg(f_\ell) = 0. \tag{6.7}$$

Together with $\boldsymbol{\rho}$, we see that the correspondence between $\vec{M}, \vec{N}$ and $\boldsymbol{\rho}, \boldsymbol{\sigma}$ is one-to-one.

**Brane construction**   Recall from Section 4.1, the brane realisation of $T_{\boldsymbol{\rho}}^{\boldsymbol{\sigma}}[\mathrm{SU}(n)]$ is given by $n$ D3 branes suspended between $\ell$ NS5 and $\ell'$ D5 branes. The parts of $\boldsymbol{\rho}$ are the net number of D3s ending in the NS5 branes going from the interior to exterior; likewise, the parts of $\boldsymbol{\sigma}$ are the net number of D3s ending on D5 branes going from interior to exterior. As demonstrated in [37], mirror symmetry for 3d $\mathcal{N} = 4$ quiver gauge theories is realised in the brane system by a combination of S-duality transformation and space-time rotation. The S-transformation exchanges D5s with NS5s, F1s with D1s, while D3 branes are invariant. Thus, S-duality acting on the brane configuration for $T_{\boldsymbol{\rho}}^{\boldsymbol{\sigma}}[\mathrm{SU}(n)]$ produces the brane configuration for $T_{\boldsymbol{\sigma}}^{\boldsymbol{\rho}}[\mathrm{SU}(n)]$. By a series of standard brane moves, one transitions the brane system into a phase resembling that of Figure 4.

The brane system gives clear explanations for the mapping of parameters. Coulomb

branch moduli, captured by D3s suspended between NS5 branes, are mapped to Higgs branch degrees of freedom, represented by D3s suspended between D5 branes, and vice versa. Likewise, D5 brane positions transverse to D3 and NS5 branes give rise to mass parameters, which are mapped to NS5 brane positions transverse to D3 and D5 branes defining FI parameters.

**Mirror symmetry**  Mirror symmetry states that the following two theories are the same

$$T_{\boldsymbol{\rho}}^{\boldsymbol{\sigma}}[\mathrm{SU}(n)] \quad \longleftrightarrow \quad T_{\boldsymbol{\sigma}}^{\boldsymbol{\rho}}[\mathrm{SU}(n)]. \tag{6.8}$$

At the level of Bethe ansatz, mirror symmetry can be seen in different ways

- There is a one-to-one correspondence of the solutions of the two sets of seemingly quite different sets of BAE;

- The handle-gluing operator, evaluated at the dual solutions yield the same result.

- The Higgs branch Higgsing in one theory corresponds to the Coulomb branch Higgsing in the mirror symmetry.

To make such identifications, we also need to identify the corresponding parameters including inhomogeneities and twists.

In the gauge theory setup, mirror symmetry relates parameters as follows: let $y_i$ be the mass and $\epsilon_a$ the FI parameter of $T_{\boldsymbol{\rho}}^{\boldsymbol{\sigma}}[\mathrm{SU}(n)]$, and denote by $q$ the $\mathcal{N} = 2^*$ deformation parameter. Likewise $y_a^{\vee}$ and $\epsilon_i^{\vee}$ are the mass and FI parameter of $T_{\boldsymbol{\sigma}}^{\boldsymbol{\rho}}[\mathrm{SU}(n)]$, and the SUSY breaking parameter $q^{\vee}$. Then the mirror map is simply

$$y_i \leftrightarrow \epsilon_i^{\vee}, \qquad \epsilon_a \leftrightarrow y_a^{\vee}, \qquad q \leftrightarrow \frac{1}{q^{\vee}} \tag{6.9}$$

*i.e.* FI and mass parameters are exchange, while the $\mathcal{N} = 2^*$ parameter is inverted.

## 6.1 Explicit examples

Suppose two theories are mirror dual to each other, then partition functions and supersymmetric indices of these two theories need to agree upon using the mirror map between the parameters. Thus, starting from topologically twisted indices written as sum over Bethe vacua, there is also a one-to-one correspondence between Bethe roots of two utterly different set of Bethe ansatz equations.

**Example 1**  Consider SQED with $N_f = 3$ and its mirror $\mathrm{U}(1) \times \mathrm{U}(1)$ quiver gauge theory.

$$
\begin{array}{c}
\overset{3}{\square}\ y_1,y_2,y_3 \\[4pt]
| \\[2pt]
\underset{1}{\bigcirc} \\[4pt]
x \\[2pt]
\frac{\eta_2}{\eta_1}
\end{array}
\qquad \leftrightarrow \qquad
\begin{array}{c}
z_1\ \overset{1}{\square}\quad \overset{1}{\square}\ z_2 \\[4pt]
|\qquad\ | \\[2pt]
\underset{1}{\bigcirc}\!-\!\underset{1}{\bigcirc} \\[4pt]
x^{(1)}\quad x^{(2)} \\[2pt]
\frac{\epsilon_2}{\epsilon_1}\quad \frac{\epsilon_3}{\epsilon_2}
\end{array}
\tag{6.10}
$$

The $\mathcal{N}=2^*$ parameter is denoted by $q$ for the SQED theory and by $p$ for the mirror quiver.

Suppose one solves the BAE for the quiver theory with $\epsilon_i = 1$, such that the mass parameters relate to the physical mass via $z_1 = \sqrt{z}$, $z_2 = 1/\sqrt{z}$. One finds

$$
\mathcal{S}_{\mathrm{BE}}^{\mathrm{quiver}} = \left\{ \left\{ \begin{array}{l} x^{(1)} \to \sqrt[6]{z} \\ x^{(2)} \to \frac{1}{\sqrt[6]{z}} \end{array} \right\},\ \left\{ \begin{array}{l} x^{(1)} \to -\sqrt[3]{-1}\sqrt[6]{z} \\ x^{(2)} \to \frac{(-1)^{2/3}}{\sqrt[6]{z}} \end{array} \right\},\ \left\{ \begin{array}{l} x^{(1)} \to (-1)^{2/3}\sqrt[6]{z} \\ x^{(2)} \to -\frac{\sqrt[3]{-1}}{\sqrt[6]{z}} \end{array} \right\} \right\}
\tag{6.11}
$$

Likewise, solving the BAE for the SQED theory with $y_i = 1$ yields

$$
\mathcal{S}_{\mathrm{BE}}^{\mathrm{SQED}} = \left\{ \left\{ x \to \frac{\sqrt[3]{\varepsilon}q - 1}{\sqrt[3]{\varepsilon} - q} \right\},\ \left\{ x \to \frac{2\varepsilon^{2/3}q + \sqrt[3]{\varepsilon}\left((1 - i\sqrt{3})q^2 + i\sqrt{3} + 1\right) + 2q}{2\left(\varepsilon^{2/3} + \sqrt[3]{\varepsilon}q + t^2\right)} \right\}, \right.
\tag{6.12}
$$
$$
\left. \left\{ x \to \frac{2\varepsilon^{2/3}q + \sqrt[3]{\varepsilon}\left((1 + i\sqrt{3})q^2 - i\sqrt{3} + 1\right) + 2q}{2\left(\varepsilon^{2/3} + \sqrt[3]{\varepsilon}q + q^2\right)} \right\} \right\}
$$

where the FI parameter is denoted with $\varepsilon$, i.e. $\eta_2 = \sqrt{\varepsilon}$, $\eta_1 = \frac{1}{\sqrt{\varepsilon}}$.

Now, one can identify the mirror pairs of corresponding Bethe roots by evaluating the A or B-twist handle-gluing operator $\mathcal{H}_{A/B}$. For the A-twist of the quiver theory, one computes

$$
\frac{1}{H_A^{\mathrm{quiver}}(\boldsymbol{x}_1)} = \frac{p^2\,(\sqrt[3]{z} - p)^2\,(p\sqrt[3]{z} - 1)^2}{3\,(p^2 - 1)^4\,z^{2/3}}
\tag{6.13a}
$$

$$
\frac{1}{H_A^{\mathrm{quiver}}(\boldsymbol{x}_2)} = \frac{1}{3\,(p^2 - 1)^4\,z^{2/3}\left(-p + \sqrt[3]{-1}pz^{2/3} + (-1)^{2/3}(p^2 + 1)\sqrt[3]{z}\right)} \cdot
\tag{6.13b}
$$
$$
p^2\Big(-(-1)^{2/3}p^3 - 3\,(p^5 + 3p^3 + p)\,z^{2/3} + 3\sqrt[3]{-1}p\,(p^4 + 3p^2 + 1)\,z^{4/3}
$$
$$
+ 3\,(p^4 + p^2)\,z^{5/3} - (-1)^{2/3}p^3z^2 - 3\sqrt[3]{-1}p^2\,(p^2 + 1)\sqrt[3]{z}
$$
$$
+ (-1)^{2/3}\,(p^6 + 9p^4 + 9p^2 + 1)\,z\Big)
$$

$$
\frac{1}{H_A^{\mathrm{quiver}}(\boldsymbol{x}_3)} = -\frac{1}{3\,(p^2 - 1)^4\,z^{2/3}\left(-\sqrt[3]{-1}p + pz^{2/3} - (-1)^{2/3}(p^2 + 1)\sqrt[3]{z}\right)} \cdot
\tag{6.13c}
$$
$$
p^2\Big(-(-1)^{2/3}p^3 - 3\sqrt[3]{-1}p^2\,(p^2 + 1)\,z^{5/3} - 3\,(p^5 + 3p^3 + p)\,z^{4/3}
$$
$$
+ 3\sqrt[3]{-1}p\,(p^4 + 3p^2 + 1)\,z^{2/3} - (-1)^{2/3}p^3z^2 + 3\,(p^4 + p^2)\sqrt[3]{z}
$$

$$+ (-1)^{2/3} \left(p^6 + 9p^4 + 9p^2 + 1\right) z \Bigg)$$

while the B-twist in SQED yields

$$\frac{1}{H_B^{\text{SQED}}(\boldsymbol{x}_1)} = \frac{q^2 \left(\sqrt[3]{\varepsilon} - q\right)^2 \left(\sqrt[3]{\varepsilon} q - 1\right)^2}{3\varepsilon^{2/3} \left(q^2 - 1\right)^4} \tag{6.14a}$$

$$\frac{1}{H_B^{\text{SQED}}(\boldsymbol{x}_2)} = \frac{-1}{\left(-12 i \varepsilon^{2/3} q^2 + \left(\sqrt{3} + i\right) \varepsilon^{4/3} - 4 \left(\sqrt{3} + i\right) \sqrt[3]{\varepsilon} q^3 + 4 \left(\sqrt{3} - i\right) \varepsilon q - \left(\sqrt{3} - i\right) q^4\right)}$$

$$\cdot \frac{q^2 \left(\varepsilon^{2/3} + \sqrt[3]{\varepsilon} q + q^2\right)^2}{3\varepsilon^{2/3} \left(q^2 - 1\right)^4} \cdot \left(2 i \varepsilon^{4/3} q^2 + \varepsilon^{2/3} \left(\left(\sqrt{3} - i\right) q^4 + 8 i q^2 - i - \sqrt{3}\right)\right.$$

$$\tag{6.14b}$$

$$\left. + 2 \sqrt[3]{\varepsilon} q \left(\left(\sqrt{3} + i\right) q^2 + i - \sqrt{3}\right) + 2 \varepsilon q \left(\left(\sqrt{3} + i\right) q^2 + i - \sqrt{3}\right) + 2 i q^2\right)$$

$$\frac{1}{H_B^{\text{SQED}}(\boldsymbol{x}_3)} = \frac{1}{\left(12 i \varepsilon^{2/3} q^2 + \left(\sqrt{3} - i\right) \varepsilon^{4/3} - 4 \left(\sqrt{3} - i\right) \sqrt[3]{\varepsilon} q^3 + 4 \left(\sqrt{3} + i\right) \varepsilon q - \left(\sqrt{3} + i\right) q^4\right)}$$

$$\cdot \frac{q^2 \left(\varepsilon^{2/3} + \sqrt[3]{\varepsilon} q + q^2\right)^2}{3\varepsilon^{2/3} \left(q^2 - 1\right)^4} \cdot \left(2 i \varepsilon^{4/3} q^2 + \varepsilon^{2/3} \left(- \left(\sqrt{3} + i\right) q^4 + 8 i q^2 - i + \sqrt{3}\right)\right.$$

$$\tag{6.14c}$$

$$\left. + 2 \sqrt[3]{\varepsilon} q \left(- \left(\sqrt{3} - i\right) q^2 + i + \sqrt{3}\right) + 2 \varepsilon q \left(- \left(\sqrt{3} - i\right) q^2 + i + \sqrt{3}\right) + 2 i q^2\right)$$

Next, one compares the expression using the mirror map of the parameters. This results in

$$\frac{1}{H_A^{\text{quiver}}(\boldsymbol{x}_a)} = \left.\frac{1}{H_B^{\text{SQED}}(\boldsymbol{x}_a)}\right|_{\substack{\varepsilon \to z \\ q \to \frac{1}{p}}} \qquad \forall a = 1, 2, 3 \tag{6.15}$$

$$\sum_a \frac{1}{H_A^{\text{quiver}}(\boldsymbol{x}_a)} = p^2 \frac{1 + 4p^2 + p^4}{(1 - p^2)^4} = \left.\sum_a \frac{1}{H_B^{\text{SQED}}(\boldsymbol{x}_a)}\right|_{\substack{\varepsilon \to z \\ q \to \frac{1}{p}}} = \left. q^2 \frac{1 + 4q^2 + q^4}{(1 - q^2)^4}\right|_{q \to \frac{1}{p}} \tag{6.16}$$

and, therefore, the Bethe roots are in one-to-one correspondence. Also, the genus-0 index (6.16) agrees precisely with the known Coulomb/Higgs branch Hilbert series [60, 61].

For completeness, one evaluates the B-twist for the quiver theory

$$\frac{1}{H_B^{\text{quiver}}(\boldsymbol{x}_1)} = -\frac{p \sqrt[3]{z}}{3 \left(\sqrt[3]{z} - p\right) \left(p \sqrt[3]{z} - 1\right)} \tag{6.17a}$$

$$\frac{1}{H_B^{\text{quiver}}(\boldsymbol{x}_2)} = -\frac{p \sqrt[3]{z}}{3 \left(-\sqrt[3]{-1} p + (-1)^{2/3} p z^{2/3} - p^2 \sqrt[3]{z} - \sqrt[3]{z}\right)} \tag{6.17b}$$

$$\frac{1}{H_B^{\text{quiver}}(\boldsymbol{x}_3)} = \frac{p \sqrt[3]{z}}{3 \left(-(-1)^{2/3} p + \sqrt[3]{-1} p z^{2/3} + p^2 \sqrt[3]{z} + \sqrt[3]{z}\right)} \tag{6.17c}$$

as well as the A-twist of the SQED theory

$$\frac{1}{H_A^{\text{SQED}}(\boldsymbol{x}_1)} = -\frac{\sqrt[3]{\varepsilon}q}{3\left(\sqrt[3]{\varepsilon}-q\right)\left(\sqrt[3]{\varepsilon}q-1\right)} \tag{6.18a}$$

$$\frac{1}{H_A^{\text{SQED}}(\boldsymbol{x}_2)} = \frac{4\varepsilon^{2/3}q^2 + \left(1-i\sqrt{3}\right)\sqrt[3]{\varepsilon}q^3 + \varepsilon\left(q+i\sqrt{3}q\right)}{3\left(\varepsilon^{2/3}+\sqrt[3]{\varepsilon}q+q^2\right)\left(2\varepsilon^{2/3}q+\sqrt[3]{\varepsilon}\left(\left(1-i\sqrt{3}\right)q^2+i\sqrt{3}+1\right)+2q\right)} \tag{6.18b}$$

$$\frac{1}{H_A^{\text{SQED}}(\boldsymbol{x}_3)} = \frac{4\varepsilon^{2/3}q^2 + \left(1+i\sqrt{3}\right)\sqrt[3]{\varepsilon}q^3 + \varepsilon\left(q-i\sqrt{3}q\right)}{3\left(\varepsilon^{2/3}+\sqrt[3]{\varepsilon}q+q^2\right)\left(2\varepsilon^{2/3}q+\sqrt[3]{\varepsilon}\left(\left(1+i\sqrt{3}\right)q^2-i\sqrt{3}+1\right)+2q\right)} \tag{6.18c}$$

Again, explicitly comparing the expressions using the mirror map yields

$$\left(\mathcal{H}_B^{\text{quiver}}\right)^{-1}(\boldsymbol{x}_a) = \left(\mathcal{H}_A^{\text{SQED}}\right)^{-1}(\boldsymbol{x}_a)\Big|_{\substack{\varepsilon\to z\\q\to\frac{1}{p}}} \qquad \forall a = 1,2,3 \tag{6.19}$$

$$\sum_a \left(\mathcal{H}_B^{\text{quiver}}\right)^{-1}(\boldsymbol{x}_a) = \frac{p\left(1-p^6\right)}{\left(1-p^2\right)\left(1-\frac{p^3}{z}\right)\left(1-p^3 z\right)} \tag{6.20}$$

$$= \sum_a \left(\mathcal{H}_A^{\text{SQED}}\right)^{-1}(\boldsymbol{x}_a)\Big|_{\substack{\varepsilon\to z\\q\to\frac{1}{p}}} = \frac{q\left(1-q^6\right)}{\left(1-q^2\right)\left(1-\frac{q^3}{\varepsilon}\right)\left(1-\varepsilon q^3\right)}\Big|_{q\to\frac{1}{p}}$$

which confirms the one-to-one correspondence. Again, the computed genus-0 index (6.20) agrees with known Hilbert series [60, 61].

**Example 2.** Consider U(2) SQCD with 4 fundamentals and its mirror quiver.

$$\tag{6.21}$$

and the $\mathcal{N} = 2^*$ parameter for SQCD is denoted with $q$, while it is called $p$ in the mirror quiver. Due to the complexity of BAE, one needs to specify the fugacities. The parameter of the mirror quiver are chosen as

$$\left\{\epsilon_1 \to 2, \epsilon_2 \to 3, \epsilon_3 \to 5, \epsilon_4 \to 7, p \to \frac{1}{29}, z_1 \to \frac{1}{11}, z_2 \to 13\right\} \tag{6.22}$$

and the corresponding mirror parameter in the SQCD theory are obtained via

$$y_i \leftrightarrow \epsilon_i \text{, for } i = 1,\ldots,4 \text{,} \qquad \eta_a \leftrightarrow z_{3-a} \text{, for } a = 1,2 \text{,} \qquad q \leftrightarrow \frac{1}{p} \text{.} \tag{6.23}$$

For the quiver theory one finds the Bethe roots as displayed in Table 7, while the Bethe roots for SQCD are summarised in Table 8. By evaluating the valued of $\mathcal{H}_{A/B}^{-1}$ one can establish a one-to-one correspondence between the Bethe roots in the two theories.

| | $x^{(1)}$ | $x_1^{(2)}$ | $x_1^{(2)}$ | $x^{(3)}$ |
|---|---|---|---|---|
| $\boldsymbol{x}_1$ | $0.29214$ | $-13.264$ | $-0.014713$ | $0.31343$ |
| $\boldsymbol{x}_2$ | $-0.30412$ | $12.872$ | $0.016048$ | $-0.32564$ |
| $\boldsymbol{x}_3$ | $-0.56070$ | $0.27390 + 0.63429i$ | $0.27390 - 0.63429i$ | $-0.58099$ |
| $\boldsymbol{x}_4$ | $0.58302$ | $0.35993 + 0.61120i$ | $0.35993 - 0.61120i$ | $0.60279$ |
| $\boldsymbol{x}_5$ | $0.00158 + 0.57019i$ | $0.84342 + 0.00011i$ | $-0.57878 + 0.00019i$ | $0.00124 - 0.59025i$ |
| $\boldsymbol{x}_6$ | $0.00158 - 0.57019i$ | $0.84342 - 0.00011i$ | $-0.57878 - 0.00019i$ | $0.00124 + 0.59025i$ |

(a) Bethe roots

| | $\mathcal{H}_A^{-1}$ | $\mathcal{H}_B^{-1}$ |
|---|---|---|
| $\boldsymbol{x}_1$ | $3.7916 \cdot 10^{-7}$ | $2.9932 \cdot 10^{-7}$ |
| $\boldsymbol{x}_2$ | $3.4321 \cdot 10^{-7}$ | $4.0216 \cdot 10^{-7}$ |
| $\boldsymbol{x}_3$ | $2.1491 \cdot 10^{-10}$ | $1.7344 \cdot 10^{-4}$ |
| $\boldsymbol{x}_4$ | $1.3796 \cdot 10^{-10}$ | $1.8503 \cdot 10^{-4}$ |
| $\boldsymbol{x}_5$ | $1.7649 \cdot 10^{-10}$ | $1.7909 \cdot 10^{-4} - 7.5453 \cdot 10^{-9}i$ |
| $\boldsymbol{x}_6$ | $1.7649 \cdot 10^{-10}$ | $1.7909 \cdot 10^{-4} + 7.5453 \cdot 10^{-9}i$ |

(b) $\mathcal{H}_{A/B}$ evaluate on Bethe roots

**Table 7**. Bethe roots and value of A/B-twisted handle-glue operator for quiver gauge theory (6.21) mirror to U(2) SQCD.

| $x_1$ | $x_2$ | $\mathcal{H}_A^{-1}$ | $\mathcal{H}_B^{-1}$ |
|---|---|---|---|
| $0.31175 - 1.08299i$ | $0.31175 + 1.08299i$ | $2.9932 \cdot 10^{-7}$ | $3.7916 \cdot 10^{-7}$ |
| $1.3780$ | $-0.83561$ | $4.0216 \cdot 10^{-7}$ | $3.4321 \cdot 10^{-7}$ |
| $9.2615$ | $0.00969$ | $1.7344 \cdot 10^{-4}$ | $2.1491 \cdot 10^{-10}$ |
| $-9.2651$ | $-0.00854$ | $1.8503 \cdot 10^{-4}$ | $1.3796 \cdot 10^{-10}$ |
| $0.2917 + 9.1954i$ | $-0.00029 + 0.00914i$ | $1.7909 \cdot 10^{-4} - 7.5453 \cdot 10^{-9}i$ | $1.7649 \cdot 10^{-10}$ |
| $0.2917 - 9.1954i$ | $-0.00029 - 0.00914i$ | $1.7909 \cdot 10^{-4} + 7.5453 \cdot 10^{-9}i$ | $1.7649 \cdot 10^{-10}$ |

**Table 8**. Bethe roots and value of the A/B-twisted handle-glue operator for the U(2) SQCD theory (6.21) with 4 fundamentals.

## 6.2 Higgsing and mirror symmetry

One incarnation of mirror symmetry is the exchange of Higgs and Coulomb branch in two mirror dual theories. As such, one needs to verify that the minimal Higgs branch transitions in $\mathcal{T}$ are properly mapped to the minimal Coulomb branch transitions in $\mathcal{T}^\vee$. As both type of transitions have been detailed in terms of BAE, one verifies that the prescriptions are mapped into each other, see also [21].

$A_{M-1}$ **transition.** To begin with, consider a $A_{M-1}$ Higgs branch transition of $\mathcal{T}$ on a gauge node $U(N)$ with $M \geq 2$ fundamental flavours. Denote the two adjusted flavour masses as $y_{1,M}$ such that (5.1) implies

$$y_1 = y_M q^2 . \tag{6.24}$$

In the mirror $\mathcal{T}^\vee$, there has to exist a connected sub-graph of $M-1$ balanced gauge nodes. Denote the $\epsilon$-parameter by $\epsilon_1, \ldots, \epsilon_M$ such that (5.26) implies

$$\epsilon_1 = \epsilon_M q^2 \tag{6.25}$$

These transitions are mirror dual to each other, provided the parameters are mapped as follows:

$$q \to q^{-1} , \qquad y_1 \leftrightarrow \epsilon_M , \qquad y_M \leftrightarrow \epsilon_1 . \tag{6.26}$$

$a_k$ **transition.** Thereafter, consider a $a_k$ Higgs branch transition of $\mathcal{T}$ between a chain of gauge node $U(N_j)$ with $s \leq j \leq r$ such that all $M_j = 0$ for $s < j < r$ and $M_{s,r} = 1$. For $k = r - s + 1 > 1$, the two adjusted flavour masses, say $y_1 \equiv y_1^{(s)}$ and $y_2 \equiv y_1^{(r)}$ satisfy (5.1), *i.e.*

$$y_1 = y_2 q^{k+2} . \tag{6.27}$$

In the mirror $\mathcal{T}^\vee$, there has to exist an unbalanced gauge nodes $U(N)$ with balance $e = k$. Denote the $\epsilon$-parameter by $\epsilon_{1,2}$ such that (A.35) implies

$$\epsilon_1 = \epsilon_2 q^{e+2} . \tag{6.28}$$

This is consistent, provided the mirror map is

$$q \to q^{-1} , \qquad y_1 \leftrightarrow \epsilon_2 , \qquad y_2 \leftrightarrow \epsilon_1 . \tag{6.29}$$

## 6.3 Mirror symmetry and quiver subtraction

After discussing the minimal Higgs and Coulomb branch transitions, we have the following observation: It is known that the minimal Higgs branch transitions $A_{M-1}$ and $a_k$ can be realised on the level of the quiver by *quiver subtraction* [64]. Given that we understand also the mirror dual configurations, it follows that we can propose the quiver subtraction for Coulomb branch Higgsing building on the discussion in Section 5.3.

Consider the $A_{M-1}$ transition in Figure 9a. The Higgs branch subtraction is realised by subtracting SQED with $M$ flavours. The Coulomb branch thereof is the $A_{M-1}$ ALE space $\mathbb{C}^2/\mathbb{Z}_M$; hence, the name. After subtracting the rank of the gauge nodes, one needs

to preserve the original balance by adjusting the flavour nodes. The result is precisely the quiver theory we read off from the corresponding partial Higgsing in the brane system. Consider the mirror configuration in Figure 9b. From the brane system we know that a non-abelian U($M$) flavour node leads in the S-dual to $M$ consecutive NS5 brane with identical linking numbers. In other words, there are $M - 1$ consecutive gauge nodes with vanishing balance $e_i^\vee = 0$. We propose that the Coulomb branch subtraction is then realised subtracting a *finite $A_{M-1}$ Dynkin quiver*. While the balance is not preserved in Coulomb branch Higgsing, the flavour groups are. Thus, we do obtain the correct quiver after the transition.

Consider the $a_k$ transition in Figure 10a. The Higgs branch quiver subtraction is realised by the U(1)$^k$ quiver gauge theory whose Coulomb branch is the closure of the minimal nilpotent orbit of $\mathfrak{su}(k+1)$; hence the name $a_k$. After reducing the gauge ranks appropriately and "rebalancing" to preserve the $e_i$, one obtains the correct quiver. Giving that the two relevant D5 flavour branes are separated by $k$ NS5 branes, on the mirror side, there exists a gauge node with balance $e = k$, see Figure 10b. This node is surrounded by node of positive balance. We propose that the Coulomb branch subtraction is simply realised by subtracting a U(1) node. Since balance does not need to be preserved, this subtraction immediately generates the correct quiver.

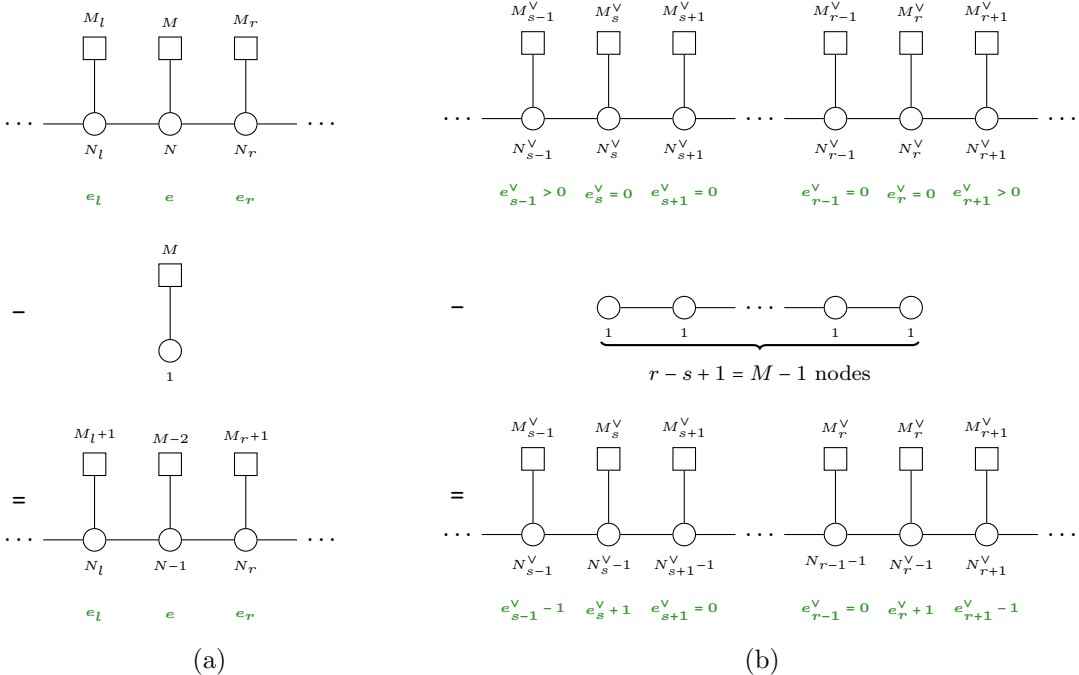

(a)          (b)

**Figure 9**. a: The $A_{M-1}$ transition on the Higgs branch is realised by subtracting the quiver of U(1) SQED with $M$ flavours and adjusting the flavour nodes such that the balance $e_i$ is preserved. b: In the mirror, the exists a sequence of $M - 1$ balance nodes and the Coulomb branch transition is realised by subtracting a finite $A_{M-1}$ Dynkin diagram.

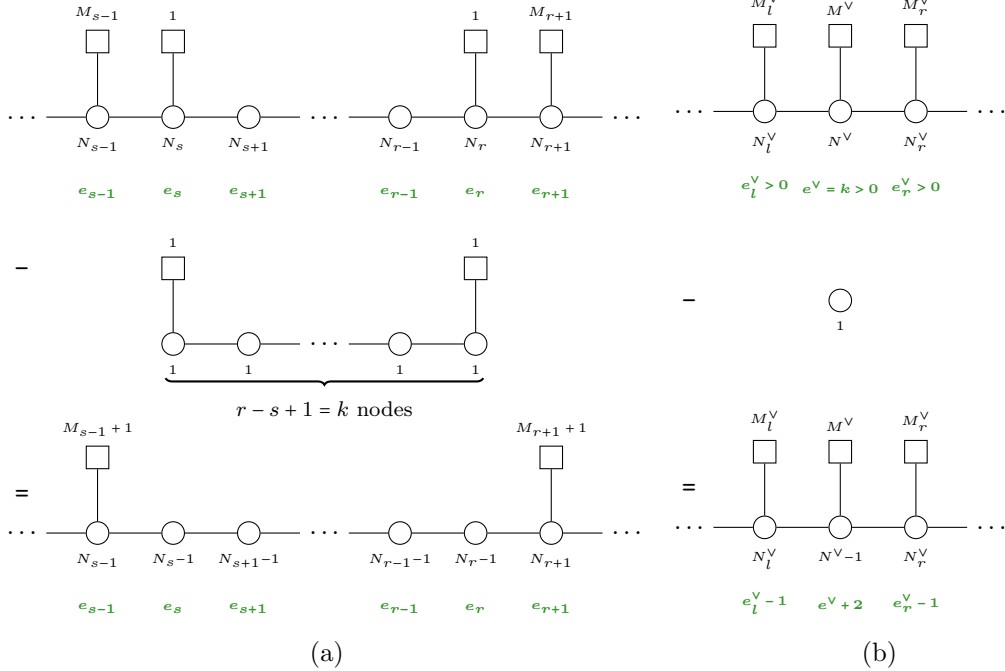

**Figure 10**. a: The $a_k$ transition on the Higgs branch is realised by subtracting the quiver of $[1] - (1) - \ldots - (1) - [1]$ with $k$ U(1) gauge factors and adjusting the flavour nodes such that the balance $e_i$ is preserved. b: In the mirror, the exists a node with balance $e^\vee = k$ surrounded by node with strictly positive balance. The Coulomb branch transition is realised by subtracting a finite $A_1$ Dynkin diagram.

One notes that this subtraction is different from the know algorithm [63–65], wherein the subtracted diagrams are of affine Dynkin type. The Coulomb branch quiver subtraction is significant for the *magnetic quiver* programme, see [66–71] and later works. For example, given a 3d $\mathcal{N} = 4$ $A$-type quiver theory $\mathcal{T}$ and suppose one knows the mirror $\mathcal{T}^\vee$. One might ask: what is the mirror after a minimal Higgs branch transition $X : \mathcal{T} \to \mathcal{T}'$? Using the corresponding Coulomb branch quiver subtraction for $X$, one straightforwardly obtains $X : \mathcal{T}^\vee \to (\mathcal{T}')^\vee$ such that $(\mathcal{T}')^\vee$ is the 3d mirror of $\mathcal{T}'$. The significance of Coulomb branch quiver subtraction is that the same logic applies to magnetic quivers[6]: given a higher-dimensional theory (8 supercharges) with known magnetic quiver, one is interested in the magnetic quiver after a partial Higgs mechanism. Applying Coulomb branch quiver subtraction (and suitable future generalisations [72]) allows to answer this.

An immediate corollary of this discussion is the following: 3d $\mathcal{N} = 4$ Sp$(k)$ SQCD with $N$ fundamentals admits a unitary $D$-type Dynkin quiver as mirror dual theory. The partial Higgs branch Higgsing of Sp$(k)$ SQCD with $N$ fundamentals to Sp$(k-1)$ SQCD with $N-2$ fundamentals is known as $d_N$ transition. By the same reasoning as above, we find that the corresponding Coulomb branch Higgsing on the $D$-type mirror quiver is realised by subtracting a *finite $D_N$ Dynkin quiver*, wherein the gauge ranks are precisely

---

[6]M.S. thanks Antoine Bourget and Zhenghao Zhong for discussion and collaboration on related projects.

the Coxeter labels.

# 7 Comments on open spin chains and orthosymplectic quivers

The setup considered so far can be naturally generalised by inclusion of O3 orientifold planes in the Type IIB brane systems. The O3 planes are parallel to the D3 branes and the low-energy world-volume theory is modified into a linear quiver gauge theory with alternating orthogonal and symplectic gauge nodes. In short, this is referred to as orthosymplectic quiver. The Bethe/Gauge correspondence relates such 3d $\mathcal{N} = 4$ theories to open spin chains [19]. In this Section, the formulation in terms of the $Q$-system is briefly discussed.

## 7.1 Brane system and 3d theory

The inclusion of an O3 plane comes with different choices, as there are four types of orientifold planes. Analogous to above, consider a stack of $n$ D3 brane parallel to an O3 plane, ending on a system of half D5 branes and half NS5 branes [26, 73]. Two partitions $\boldsymbol{\sigma}$, $\boldsymbol{\rho}$ determine how the D3 branes end on the half D5 and half NS5 branes respectively. The brane setup gives rise to the 3d $\mathcal{N} = 4$ superconformal field theories $T_{\boldsymbol{\rho}}^{\boldsymbol{\sigma}}[G]$ that are the IR fixed points of the D3 world-volume theories. Table 9 summarises the choice of orientifold, which determines $G$ and the two partitions. By construction, mirror symmetry is realised by

$$T_{\boldsymbol{\rho}}^{\boldsymbol{\sigma}}[G] \quad \longleftrightarrow \quad T_{\boldsymbol{\sigma}}^{\boldsymbol{\rho}}[G^{\vee}], \tag{7.1}$$

where $G^{\vee}$ is the GNO-dual group of $G$ [74].

In contrast to the linear unitary quivers, the IR global symmetry $G_H \times G_C$ is only partially visible in the UV description. The Cartan elements of $G_H$ are still realised by explicit mass parameters in the orthosymplectic quiver. However, the Coulomb branch global symmetry is not manifest in the UV, simply because $\mathrm{SO}(n)$ and $\mathrm{Sp}(k)$ gauge theories do not admit FI-parameter.

## 7.2 $Q$-system

The Bethe/Gauge correspondence for $\mathrm{SO}(n)$ and $\mathrm{Sp}(k)$ gauge theories have been investigated in [25]. It has been shown that the vacuum equations of these theories correspond to BAEs of integrable open spin chains with diagonal boundary conditions. Such BAEs can also been recast in terms of the rational $Q$-system. This was first done for a special case for the XXZ spin chain in [15], later it was generalized to the situation with more general diagonal boundary conditions in [16].

The rational $Q$-systems for open chain have a number of new features. First, the $QQ$-relation is modified; second, the boundary conditions are such that the $Q$-functions at the left boundary are even functions of the spectral parameter, namely $Q_{a,0}(-u) = Q_{a,0}(u)$.

| O3 | theory | $\boldsymbol{\sigma}$ partition | $\boldsymbol{\rho}$ partition |
|---|---|---|---|
| $O3^-$ | $T_{\boldsymbol{\rho}}^{\boldsymbol{\sigma}}[SO(2n)]$ | $SO(2n)$ partition | $SO(2n)$ partition |
| $\widetilde{O3}^-$ | $T_{\boldsymbol{\rho}}^{\boldsymbol{\sigma}}[SO(2n+1)]$ | $SO(2n+1)$ partition | $Sp(n)$ partition |
| $O3^+$ | $T_{\boldsymbol{\rho}}^{\boldsymbol{\sigma}}[Sp(n)]$ | $Sp(n)$ partition | $SO(2n+1)$ partition |
| $\widetilde{O3}^+$ | $T_{\boldsymbol{\rho}}^{\boldsymbol{\sigma}}[Sp'(n)]$ | $Sp(n)$ partition | $Sp(n)$ partition |

**Table 9**. The $T_{\boldsymbol{\rho}}^{\boldsymbol{\sigma}}[G]$ theories are defined by a $G$-partition $\boldsymbol{\sigma}$ and a $G^\vee$-partition $\boldsymbol{\rho}$, with $G^\vee$ the GNO-dual group of $G$.

In addition, it was shown in [16] that the corresponding $Q$-system is not unique. These observations were made by investigating rank 1 open spin chains, namely the integrable open XXX and XXZ spin chains. We expect that these features hold for higher rank models in general.

## 8 Conclusions

In this paper, we constructed the rational $Q$-system for generic BAE described by an $A_{\ell-1}$ quiver and revisit the Bethe/Gauge correspondence from the rational $Q$-system point of view. We obtained a number of new results in this study.

For integrable models, the rational $Q$-systems for $A_{\ell-1}$ BAE have been constructed for models with one momentum carrying node first for the XXX model in [13] and then for the XXZ model in [17]. Building on these works, we took one further step and generalized the framework to cases with multiple momentum carrying nodes and generic twists. Such a generalization is helpful for applications in integrability in AdS/CFT. For example, the scalar sector of ABJM theory is described by a BAE with two momentum carrying nodes [35, 75], the rational $Q$-system is expected to be more efficient to solve than the BAE. The generalization to multiple momentum carrying nodes is also necessary for applications in Bethe/Gauge correspondence where these type of BAE emerge naturally from quiver gauge theories.

For 3d $\mathcal{N} = 4$ quiver gauge theories, most of the content we discussed in the paper are known in the literature. We clarified that generic Higgs/Coulomb branch Higgs transitions are composed of elementary Kraft-Procesi transitions, for which we demonstrate the suitable reduction on the level of the BAE and verified mirror symmetry. As a corollary, we formalised Coulomb branch Higgsing in terms of quiver subtraction using finite $A$-type Dynkin quivers. These preliminaries enabled us to naturally transfer the minimal partial Higgs mechanisms into the rational $Q$-system language. As a proof of concept, we evaluated topologically twisted indices via BAE and rational $Q$-system for selected examples. The rational $Q$-system outperforms solving BAE. In this work, we demonstrated this feature

for numerical calculations of U($n$) SQCD with $n = 1, \ldots, 5$ and confirmed the validity of the results by comparing genus-0 twisted indices to known Hilbert series.

Probably the most important message of the current work is that rational $Q$-system, which is not yet well appreciated beyond integrability community, provides a natural language for the Bethe/Gauge correspondence. The first evidence is that the rational $Q$-system is naturally specified by two partitions, which can be identified nicely with the two partitions of $T_{\boldsymbol{\rho}}^{\boldsymbol{\sigma}}[\mathrm{SU}(n)]$. Moreover, the correspondence of Higgsings on both branches are realized in a more transparent way in the rational $Q$-system than the original BAE. Finally, mirror symmetry is realized in an extremely neat way by simply swapping the role of the two partitions of the $Q$-system, which specify the Young tableaux and boundary conditions. It might be possible that the $Q$-functions on the Young tableaux have more direct physical meanings in terms of quiver gauge theories.

There are several directions to pursue based on the current work. The original motivation for developing rational $Q$-system for the more general $A_{\ell-1}$ is to combine the efficiency of the $Q$-system and computational algebraic geometrical methods to compute physical quantities like the topologically twisted indices analytically. Such a strategy has already been applied in the computation of various non-trivial quantities such as partition functions of 6-vertex models [7, 9, 11] and Loschmidt echo of the integrable quantum spin chains [10]. However, these applications only involve $A_1$ model. Rational $Q$-systems of higher rank $A_{\ell-1}$ are more complicated to handle. To further improve the efficiency, we need to exploit various techniques and tricks. We will report these results in a separated publication.

It would be interesting to generalize the rational $Q$-systems even further. One immediate task is considering the cases of generic $A_{\ell-1}$ open chains, building on the comments and observations given in Section 7. An even more general case is considering higher spin representations.

Mirror symmetry is a highly non-trivial and intriguing statement from the spin chain point of view. It states that two seemingly very different BAEs/$Q$-systems are dual to each other and have the same number of solutions. It would be interesting to further understand the bispectral dualities and find potential applications in statistical mechanics and/or condensed matter physics.

**Acknowledgments.** We would like to thank the lunch seminars of SEUYC which provide nice food and inspiring atmosphere, out of which this project grows. JG is supported by the Startup Funding no. 3207022203A1 and no. 4060692201/011 of the Southeast University. YJ is supported by the Startup Funding no. 3207022217A1 of the same university. MS is supported by funding no. 4007012203.

## A  Higgsing in BAE

### A.1  Higgs branch

For completeness, the reduction of the BAE (4.19) under Higgs branch Higgsing is sketched. Consider the transition detailed in Figure 5 and recall that the parameter choice (5.1) becomes

$$y_a^{(s)} = x_{N_s}^{(s)} \cdot q \,, \quad x_{N_s}^{(s)} = x_{N_{s+1}}^{(s+1)} \cdot q \,, \quad \dots \,, \quad x_{N_{r-1}}^{(r-1)} = x_{N_r}^{(r)} \cdot q \,, \quad x_{N_r}^{(r)} = y_b^{(r)} \cdot q \qquad \text{(A.1)}$$

in terms of the complex fugacities (4.8), (4.9).

**Node $s$.**  To begin with, consider the $a \neq N_s$ BAE (4.19). The terms affected by (5.1) are:

$$P_a^{(s)} \supset \frac{x_a^{(s)} q - x_{N_s}^{(s)} q^{-1}}{x_{N_s}^{(s)} q - x_a^{(s)} q^{-1}} \cdot \frac{x_a^{(s)} - y_{M_s}^{(s)} q}{y_{M_s}^{(s)} - x_a^{(s)} q} \cdot \frac{x_a^{(s)} - x_{N_{s+1}}^{(s+1)} q}{x_{N_{s+1}}^{(s+1)} - x_a^{(s)} q}\bigg|_{(A.1)} = (-1)^3 \qquad \text{(A.2)}$$

and the sign prefactor changes as

$$\delta_s = \delta_s' + 3 \qquad \Rightarrow \qquad (-1)^{\delta_s} = (-1) \cdot (-1)^{\delta_s'} \qquad \text{(A.3)}$$

such that the appearing sign factors cancel. As a consequence, the remainder of $P_a^{(s)}$ reduces to the BAE for $U(N_s - 1)$ with the matter content as in Figure 5c.

Next, consider the $a = N_s$ BAE, the terms affected by (5.1) are:

$$P_{N_s}^{(s)} \supset \prod_{d=1}^{N_s-1} \frac{x_{N_s}^{(s)} q - x_d^{(s)} q^{-1}}{x_d^{(s)} q - x_{N_s}^{(s)} q^{-1}} \cdot \prod_{i=1}^{M_s} \frac{x_{N_s}^{(s)} - y_i^{(s)} q}{y_i^{(s)} - x_{N_s}^{(s)} q} \cdot \prod_{c=1}^{N_{s+1}} \frac{x_{N_s}^{(s)} - x_c^{(s+1)} q}{x_c^{(s+1)} - x_{N_s}^{(s)} q} \qquad \text{(A.4)}$$

which implies that this BAE becomes trivial once it is written as polynomial equation. To see this note that $y_i^{(s)} - x_{N_s}^{(s)} t = 0$ for $i = M_s$ and $x_{N_s}^{(s)} - x_c^{(s+1)} t$ for $c = N_{s+1} = 0$ due to (A.1); hence, both sides of the polynomial BAE are trivial.

**Node $j$, $s < j < r$.**  For a $U(N_j)$ node with $s \leq j \leq r$, the argument is exactly the same.

**Node $s - 1$.**  Next, consider the node $U(N_{s-1})$ and verify that the additional flavour is accommodated.

$$P_a^{(s-1)} = (-1)^{\delta_{s-1}} \frac{\epsilon_s}{\epsilon_{s-1}} \prod_{\substack{d=1 \\ d \neq a}}^{N_{s-1}} \frac{x_a^{(s-1)} q - x_d^{(s-1)} q^{-1}}{x_d^{(s-1)} q - x_a^{(s-1)} q^{-1}} \cdot \prod_{i=1}^{M_{s-1}} \frac{x_a^{(s-1)} - y_i^{(s-1)} q}{y_i^{(s-1)} - x_a^{(s-1)} q} \qquad \text{(A.5)}$$

$$\cdot \prod_{b=1}^{N_{s-2}} \frac{x_a^{(s-1)} - x_b^{(s-2)} q}{x_b^{(s-2)} - x_a^{(s-1)} q} \cdot \prod_{c=1}^{N_s-1} \frac{x_a^{(s-1)} - x_c^{(s)} q}{x_c^{(s)} - x_a^{(s-1)} q} \cdot \frac{x_a^{(s-1)} - x_{N_s}^{(s)} q}{x_{N_s}^{(s)} - x_a^{(s-1)} q}$$

and the sign factor remains invariant

$$\delta_{s-1} = \delta'_{s-1}\,. \tag{A.6}$$

Therefore, the flavour contribution for the $(s-1)$-th gauge node $U(N_{s-1})$ Figure 5c are identified with

$$\tilde{P}_a^{(s-1)} \supset \prod_{i=1}^{M_{s-1}} \frac{x_a^{(s-1)} - y_i^{(s-1)}q}{y_i^{(s-1)} - x_a^{(s-1)}q} \cdot \frac{x_a^{(s-1)} - x_{N_s}^{(s)}q}{x_{N_s}^{(s)} - x_a^{(s-1)}q} \equiv \prod_{i=1}^{M_{s-1}+1} \frac{x_a^{(s-1)} - \tilde{y}_i^{(s-1)}q}{\tilde{y}_i^{(s-1)} - x_a^{(s-1)}q}\,. \tag{A.7}$$

Using $y_a^{(s)} = x_{N_s}^{(s)}q$, the additional flavour is given by

$$\left\{\tilde{y}_i^{(s-1)}\right\}_{i=1}^{M_{s-1}+1} = \left\{\left\{y_i^{(s-1)}\right\}_{i=1}^{M_{s-1}}\,,\ y_a^{(s)}q^{-1}\right\}\,. \tag{A.8}$$

**Node $r+1$.** Similarly, the additional flavour in the $(r+1)$-th gauge node $U(N_{r+1})$ should be identified as coming from $x_{N_r}^{(r)}$. More precisely, the flavour parameter after Higgsing are given by

$$\left\{\tilde{y}_j^{(r+1)}\right\}_{j=1}^{M_{r+1}+1} = \left\{\left\{y_j^{(r+1)}\right\}_{j=1}^{M_{r+1}}\,,\ y_b^{(r)}q\right\} \tag{A.9}$$

using $x_{N_r}^{(r)} = y_b^{(r)}q$, see (A.1).

## A.2 Coulomb branch

Without loss of generality, one may consider an $A$-type quiver with a balanced $A_{r-s+1}$ subgraph (and $r \geq s$), as in Figure 7a. This means that the nodes $N_j$ for $j \in \{s, s+1, \ldots, r\}$ are balanced, $i.e.$

$$e_j = N_{j-1} + N_{j+1} + M_j - 2N_j = 0 \qquad \text{for all } j \in \{s, s+1, \ldots, r\}\,. \tag{A.10}$$

After turning on a Coulomb branch VEV, all the balanced node are partially broken $U(N_j) \to U(N_j - 1)$ for $j \in \{s, s+1, \ldots, r\}$ and the resulting theory is shown in Figure 7b. On the level of BAE, the Higgsing can be realised as follows: for each affected gauge node, selected a single complex fugacity, say, $x_{N_j}^{(j)}$. Firstly, these need to be aligned and, secondly, a limit is required

$$x_{N_s}^{(s)} = \ldots = x_{N_j}^{(j)} = \ldots = x_{N_r}^{(r)} \equiv \chi \to \infty\,. \tag{A.11}$$

It is instructive to examine the behaviour of different nodes.

**Node $s-1$.** The first node that is indirectly affected is $s-1$, and in the limit $\chi \to \infty$, the relevant terms in (4.19) are

$$\lim_{x_{N_s}^{(s)}=\chi\to\infty} \frac{x_a^{(s-1)} - x_{N_s}^{(s)}q}{x_{N_s}^{(s)} - x_a^{(s-1)}q} = -q \,. \tag{A.12}$$

In addition, the sign prefactor can be recast as

$$\delta_{s-1} = \delta'_{s-1} + 1 \qquad \Rightarrow \qquad (-1)^{\delta_{s-1}} = (-1) \cdot (-1)^{\delta'_{s-1}} \,. \tag{A.13}$$

Consequently, the BAE for this node become

$$P_a^{(s-1)} \to q(-1)^{\delta'_{s-1}} \frac{\epsilon_s}{\epsilon_{s-1}} \prod_{\substack{d=1 \\ d\neq a}}^{N_{s-1}} \frac{x_a^{(s-1)}q - x_d^{(s-1)}q^{-1}}{x_d^{(s-1)}q - x_a^{(s-1)}q^{-1}} \cdot \prod_{i=1}^{M_{s-1}} \frac{x_a^{(s-1)} - y_i^{(s-1)}q}{y_i^{(s-1)} - x_a^{(s-1)}q} \tag{A.14}$$

$$\cdot \prod_{b=1}^{N_{s-2}} \frac{x_a^{(s-1)} - x_b^{(s-2)}q}{x_b^{(s-2)} - x_a^{(s-1)}q} \cdot \prod_{c=1}^{N_s-1} \frac{x_a^{(s-1)} - x_c^{(s)}q}{x_c^{(s)} - x_a^{(s-1)}q}$$

which is the BAE for the $(s-1)$-st node of the theory after Higgsing (with $\delta'_{s-1}$), up to the choice of FI (see below).

**Node $s$.** Next, consider the left-most node that experience partial breaking. In the limit $\chi \to \infty$, the relevant terms in the $a \neq N_s$ BAE (4.19) of node $s$ are

$$\lim_{x_{N_{s+1}}^{(s+1)}=\chi\to\infty} \frac{x_a^{(s)} - x_{N_{s+1}}^{(s+1)}q}{x_{N_{s+1}}^{(s+1)} - x_a^{(s)}q} = -q \tag{A.15}$$

$$\lim_{x_{N_s}^{(s)}=\chi\to\infty} \frac{x_a^{(s)}q - x_{N_s}^{(s)}q^{-1}}{x_{N_s}^{(s)}q - x_a^{(s)}q^{-1}} = \frac{-1}{q^2} \tag{A.16}$$

and the sign prefactor behaves as

$$\delta_s = \delta'_s + 2 \qquad \Rightarrow \qquad (-1)^{\delta_s} = (-1)^{\delta'_s} \,. \tag{A.17}$$

Hence, one arrives at

$$P_{a\neq N_s}^{(s)} \to \frac{1}{q}(-1)^{\delta'_s} \frac{\epsilon_{s+1}}{\epsilon_s} \prod_{\substack{d=1 \\ d\neq a}}^{N_s-1} \frac{x_a^{(s)}q - x_d^{(s)}q^{-1}}{x_d^{(s)}q - x_a^{(s)}q^{-1}} \cdot \prod_{i=1}^{M_s} \frac{x_a^{(s)} - y_i^{(s)}q}{y_i^{(s)} - x_a^{(s)}q} \tag{A.18}$$

$$\cdot \prod_{b=1}^{N_{s-1}} \frac{x_a^{(s)} - x_b^{(s-1)}q}{x_b^{(s-1)} - x_a^{(s)}q} \cdot \prod_{c=1}^{N_{s+1}-1} \frac{x_a^{(s)} - x_c^{(s+1)}q}{x_c^{(s+1)} - x_a^{(s)}q}$$

which is the BAE of node $s$ for the theory after Higgsing (with $\delta'_s$), up to the choice of FI.

Similarly, the BAE for $a = N_s$ becomes

$$\lim_{x_{N_s}^{(s)}=x_{N_{s+1}}^{(s+1)}=\chi\to\infty} P_{a=N_s}^{(s)} = \frac{\epsilon_{s+1}}{\epsilon_s}(-1)^{-e_s-1}q^{-e_s-1} = (-1)\frac{\epsilon_{s+1}}{\epsilon_s}q^{-1} \tag{A.19}$$

using that the node is balanced, *i.e.* $e_s = 0$.

**Node $j$, $s < j < r$** Next, consider an intermediate node. Again, focus on the affected parts in the $x_{N_{j-1}}^{(j-1)} = x_{N_j}^{(j)} = x_{N_{j+1}}^{(j+1)} = \chi \to \infty$ limit. In the BAE (4.19) for $a \neq N_j$, the relevant pieces are

$$\lim_{x_{N_{j-1}}^{(j-1)}=\chi\to\infty} \frac{x_a^{(j)} - x_{N_{j-1}}^{(j-1)}q}{x_{N_{j-1}}^{(j-1)} - x_a^{(j)}q} = -q \tag{A.20a}$$

$$\lim_{x_{N_{j+1}}^{(j+1)}=\chi\to\infty} \frac{x_a^{(j)} - x_{N_{j+1}}^{(j+1)}q}{x_{N_{j+1}}^{(j+1)} - x_a^{(j)}q} = -q \tag{A.20b}$$

$$\lim_{x_{N_j}^{(j)}=\chi\to\infty} \frac{x_a^{(j)}t - x_{N_j}^{(j)}q^{-1}}{x_{N_j}^{(j)}q - x_a^{(j)}q^{-1}} = \frac{-1}{q^2} \tag{A.20c}$$

and the sign factor is changes as follows:

$$\delta_j = \delta_j' + 3 \qquad \Rightarrow \qquad (-1)^{\delta_j} = (-1) \cdot (-1)^{\delta_j'}. \tag{A.21}$$

Therefore, the limit of the BAE becomes

$$P_{a\neq N_j}^{(j)} \to (-1)^{\delta_j'} \cdot \frac{\epsilon_{j+1}}{\epsilon_j} \prod_{\substack{d=1\\d\neq a}}^{N_j-1} \frac{x_a^{(j)}q - x_d^{(j)}q^{-1}}{x_d^{(j)}q - x_a^{(j)}q^{-1}} \cdot \prod_{i=1}^{M_j} \frac{x_a^{(j)} - y_i^{(j)}q}{y_i^{(j)} - x_a^{(j)}q} \tag{A.22}$$

$$\cdot \prod_{b=1}^{N_{j-1}-1} \frac{x_a^{(j)} - x_b^{(j-1)}q}{x_b^{(j-1)} - x_a^{(j)}q} \cdot \prod_{c=1}^{N_{j+1}-1} \frac{x_a^{(j)} - x_c^{(j+1)}q}{x_c^{(j+1)} - x_a^{(j)}q}$$

which are the BAE for the node $j$ of the theory after Higgsing (with $\delta_j'$), up to the choice of FI (see below). Analogous arguments for $a = N_j$ lead to

$$\lim_{x_{N_{j-1}}^{(j-1)}=x_{N_j}^{(j)}=x_{N_{j+1}}^{(j+1)}=\chi\to\infty} P_{a=N_j}^{(j)} = \frac{\epsilon_{j+1}}{\epsilon_j}q^{-e_j} = \frac{\epsilon_{j+1}}{\epsilon_j} \tag{A.23}$$

using that the node is balanced, *i.e.* $e_j = 0$

**Node $r$.** The behaviour at node $r$ is analogous to that of node $s$. By the same reasoning as above, the limit of the $a \neq N_r$ BAE becomes

$$P^{(r)}_{a \neq N_r} \to \frac{1}{q}(-1)^{\delta'_r} \frac{\epsilon_{r+1}}{\epsilon_r} \prod_{\substack{d=1 \\ d \neq a}}^{N_r - 1} \frac{x^{(r)}_a q - x^{(r)}_d q^{-1}}{x^{(r)}_d q - x^{(r)}_a q^{-1}} \cdot \prod_{i=1}^{M_r} \frac{x^{(r)}_a - y^{(r)}_i q}{y^{(r)}_i - x^{(r)}_a q} \tag{A.24}$$

$$\cdot \prod_{b=1}^{N_{r-1}-1} \frac{x^{(r)}_a - x^{(r-1)}_b q}{x^{(r-1)}_b - x^{(r)}_a q} \cdot \prod_{c=1}^{N_{r+1}} \frac{x^{(r)}_a - x^{(r+1)}_c q}{x^{(r+1)}_c - x^{(r)}_a q}$$

which are the BAE for the $r$-th node of the theory after Higgsing (with $\delta'_r$), up to the choice of FI (see below). In contrast, the limit of the $a = N_r$ BAE reads

$$P^{(r)}_{a = N_r} \to (-1)\frac{\epsilon_{r+1}}{\epsilon_r} q^{-e_r - 1} = (-1)\frac{\epsilon_{r+1}}{\epsilon_r} q^{-1} \tag{A.25}$$

using that the node is balanced, *i.e.* $e_r = 0$.

**Node $r+1$.** Similarly, the effects on node $(r+1)$ resembles that of node $(s-1)$. The by now familiar analysis leads to

$$P^{(r+1)}_a \to q(-1)^{\delta'_{r+1}} \frac{\epsilon_{r+2}}{\epsilon_{r+1}} \prod_{\substack{d=1 \\ d \neq a}}^{N_{r+1}} \frac{x^{(r+1)}_a q - x^{(r+1)}_d q^{-1}}{x^{(r+1)}_d q - x^{(r+1)}_a q^{-1}} \cdot \prod_{i=1}^{M_{r+1}} \frac{x^{(r+1)}_a - y^{(r+1)}_i q}{y^{(r+1)}_i - x^{(r+1)}_a q} \tag{A.26}$$

$$\cdot \prod_{b=1}^{N_r - 1} \frac{x^{(r+1)}_a - x^{(r)}_b q}{x^{(r)}_b - x^{(r+1)}_a q} \cdot \prod_{c=1}^{N_{r+2}} \frac{x^{(r+1)}_a - x^{(r+2)}_c q}{x^{(r+2)}_c - x^{(r+1)}_a q} \tag{A.27}$$

which are the BAE of node $(r+1)$ of the theory after Higgsing (with $\delta'_5$), up to the choice of FI (see below).

**Fixing the FI parameter.** The above (A.14), (A.18), (A.22), (A.24), and (A.26) show that one should identify the FI parameter after Higgsing as follows:

$$\frac{\tilde{\epsilon}_s}{\tilde{\epsilon}_{s-1}} = q \frac{\epsilon_s}{\epsilon_{s-1}}, \qquad \frac{\tilde{\epsilon}_{s+1}}{\tilde{\epsilon}_s} = \frac{1}{q}\frac{\epsilon_{s+1}}{\epsilon_s}, \tag{A.28a}$$

$$\frac{\tilde{\epsilon}_{j+1}}{\tilde{\epsilon}_j} = \frac{\epsilon_{j+1}}{\epsilon_j}, \qquad \text{for } s+1 < j < r-1, \tag{A.28b}$$

$$\frac{\tilde{\epsilon}_{r+1}}{\tilde{\epsilon}_r} = \frac{1}{q}\frac{\epsilon_{r+1}}{\epsilon_r}, \qquad \frac{\tilde{\epsilon}_{r+2}}{\tilde{\epsilon}_{r+1}} = q \frac{\epsilon_{r+2}}{\epsilon_{r+1}}, \tag{A.28c}$$

such that the $\tilde{\epsilon}_a$ parameters after Higgsing are identified as

$$\tilde{\epsilon}_a = \begin{cases} \epsilon_a & a < s \\ q\epsilon_s & a = s \\ \epsilon_a & s < a < r+1 \\ q^{-1}\epsilon_{r+1} & a = r+1 \\ \epsilon_a & r < a \end{cases} \tag{A.29}$$

Moreover, (A.19), (A.23), and (A.25) imply a remaining type of constraints:

$$(-1)\frac{\epsilon_{s+1}}{\epsilon_s}q^{-1} = 1\,, \qquad \frac{\epsilon_{j+1}}{\epsilon_j} = 1 \text{ for } s < j < r\,, \qquad (-1)\frac{\epsilon_{r+1}}{\epsilon_r}q^{-1} = 1\,. \tag{A.30}$$

**Remark.** With these general considerations, one can immediately understand the mirror of the $A_k$ Higgs branch transition

$$\tag{A.31}$$

Focus on the BAE of the $\mathrm{U}(N_s)$ node. For $a \neq N_s$, For $a \neq N_2$, the relevant pieces are

$$\lim_{x_{N_s}^{(s)} \to \infty} \frac{x_a^{(s)}q - x_{N_s}^{(s)}q^{-1}}{x_{N_s}^{(s)}q - x_a^{(s)}q^{-1}} = \frac{-1}{q^2} \tag{A.32a}$$

$$\delta_s = \delta_s' + 1 \quad \Rightarrow \quad (-1)^{\delta_s} = (-1)\cdot(-1)^{\delta_s'} \tag{A.32b}$$

and therefore

$$P_{a\neq N_s}^{(s)} \to \frac{1}{q^2}(-1)^{\delta_s'}\frac{\epsilon_{s+1}}{\epsilon_s}\prod_{\substack{d=1 \\ d\neq a}}^{N_s-1}\frac{x_a^{(s)}q - x_d^{(s)}q^{-1}}{x_d^{(s)}q - x_a^{(s)}q^{-1}}\cdot\prod_{i=1}^{M_s}\frac{x_a^{(s)} - y_i^{(s)}q}{y_i^{(s)} - x_a^{(s)}q} \tag{A.33}$$

$$\cdot\prod_{b=1}^{N_{s-1}}\frac{x_a^{(s)} - x_b^{(s-1)}q}{x_b^{(s-1)} - x_a^{(s)}q}\cdot\prod_{c=1}^{N_{s+1}}\frac{x_a^{(s)} - x_c^{(s+1)}q}{x_c^{(s+1)} - x_a^{(s)}q}$$

which is the BAE for the theory after Higgsing (with $\delta_s'$), up to the choice of new FI parameter

$$\tilde{\epsilon}_a = \begin{cases} q\epsilon_s & a = s\,, \\ q^{-1}\epsilon_{s+1} & a = s+1\,, \\ \epsilon_a & \text{else}\,. \end{cases} \tag{A.34}$$

For $a = N_s$, analogous reasoning leads to

$$P^{(s)}_{a=N_s} \to \frac{\epsilon_{s+1}}{\epsilon_s} q^{-e_s-2} \quad \text{such that} \quad \epsilon_{s+1} = \epsilon_s q^{e_s+2} \,. \tag{A.35}$$

# B  Topologically twisted indices

A versatile tool for probing dualities of 3d supersymmetric theories with at least 4 super-charges (*i.e.* $\mathcal{N} \geq 2$) are topologically twisted partition functions on $\Sigma_g \times S^1$ [20, 42, 43, 58, 59, 76–79]. Focusing on 3d $\mathcal{N} = 4$, two distinct choices exist: Performing the topological twist with a Cartan subgroup of $\mathrm{SU}(2)_H$ leads to an A-twisted index, while the twist by a Cartan subgroup of $\mathrm{SU}(2)_C$ yields the so-called B-twisted index.

In this appendix, the relevant formulae for the twisted indices are summarised. The conventions follow those of [42]. For instance, the real scalar in the $\mathcal{N} = 2$ vector multiplet is $\sigma = \mathrm{diag}(\sigma_a)$ for $a = 1, \ldots, \mathrm{rk}(G)$. When compactified on $S^1$ with radius $R$, the flat connections $a_0$ for the gauge field along $S^1$ along to define a natural complexification (4.9) and the exponentiated variable $x_a$ define the complex fugacities used below. Similarly, for any global $\mathrm{U}(1)_F$ symmetry, one can turn on a background flat connection and a background real scalar $\sigma_F$ such that the combination (4.7) allows to define corresponding complex fugacity (4.8).

Supersymmetric localisation reduces the partition function on $\Sigma_g \times S^1$ to

$$I_{g,A/B} = \frac{1}{|W_G|} \sum_{m \in \Gamma^*_{G^\vee}} \oint_{\mathrm{JK}} \left[ \frac{\mathrm{d}x}{2\pi\mathrm{i}} \right]^{\mathrm{rk}(G)} Z^{A/B}_{\mathrm{cl+1-loop}}(m, x) \tag{B.1}$$

where $W_G$ denotes the Weyl group of the gauge group $G$ and $\Gamma^*_{G^\vee}$ is the weight lattice of the GNO-dual group $G^\vee$ [74]. The integrand $Z_{\mathrm{cl+1-loop}}$ is composed of a classical part

$$Z^{A/B}_{\mathrm{cl}} = \boldsymbol{\tau}^{\boldsymbol{m}} \equiv \prod_I \tau_I^{m_I} \,, \tag{B.2}$$

which is determined by FI parameters for the free subgroup $\prod_I \mathrm{U}(1)_I$ of $G$, and 1-loop determinants of the different supermultiplets.

- The 1-loop determinant for a hypermultiplet in the bifundamental representation of $G \times G'$, with variables $x$ and $y$, respectively, reads

$$Z^A_{\mathrm{hyper}} = \prod_{\gamma \in \overline{F}'} \prod_{\rho \in F} \left( \frac{x^\rho y^\gamma - q}{1 - x^\rho y^\gamma q} \right)^{\rho(m)+\gamma(n)} \tag{B.3}$$

$$Z^B_{\mathrm{hyper}} = \prod_{\gamma \in \overline{F}'} \prod_{\rho \in F} \left( \frac{x^\rho y^\gamma - q}{1 - x^\rho y^\gamma q} \right)^{\rho(m)+\gamma(n)} \left[ \frac{x^\rho y^\gamma q}{(1 - x^\rho y^\gamma q)(x^\rho y^\gamma - q)} \right]^{1-g} \tag{B.4}$$

  if $G'$ is non-dynamical, then the background flux $n$ is chosen trivial.

- A vector multiplet of a gauge group $G$ contributes with the 1-loop determinant

$$Z_{\text{vector}}^A = \left(q - q^{-1}\right)^{(g-1)\text{rk}(G)} \prod_{\alpha \in \mathfrak{g}} \left(\frac{1 - x^\alpha}{q - x^\alpha q^{-1}}\right)^{\alpha(m)-g+1} \tag{B.5}$$

$$Z_{\text{vector}}^B = \frac{1}{\left(q - q^{-1}\right)^{(g-1)\text{rk}(G)}} \prod_{\alpha \in \mathfrak{g}} \left(\frac{1 - x^\alpha}{q - x^\alpha q^{-1}}\right)^{\alpha(m)} \left[\frac{1}{(1 - x^\alpha)(q - x^\alpha q^{-1})}\right]^{g-1} \tag{B.6}$$

The contour integral can be rewritten as sum over residues at the roots of the Bethe Ansatz equations. In detail, one finds [43, 58]

$$I_{g,A/B} = \frac{(-1)^{\text{rk}(G)}}{|W_G|} \sum_{\hat{\boldsymbol{x}} \in \mathcal{S}_{\text{BE}}} Z_{\text{cl+1-loop}}^{A/B}\bigg|_{m=0} \left(\det_{ab} \frac{\partial B_a}{\partial u_b}\right)^{g-1}$$

$$\mathrm{i}B_a = \frac{\partial \log Z_{\text{cl+1-loop}}^{A/B}}{\partial m_a} \tag{B.7}$$

and, equivalently, the expression can be interpreted as [42, 59, 77]

$$I_{g,A/B} = \frac{1}{|W_G|} \sum_{\hat{\boldsymbol{x}} \in \mathcal{S}_{\text{BE}}} \mathcal{H}_{A/B}(\hat{\boldsymbol{x}})^{g-1}$$

$$\mathcal{H}_{A/B}(\boldsymbol{x}) = e^{2\pi \mathrm{i}\Omega(\boldsymbol{x})} \det_{a,b} \frac{\partial^2 \widetilde{\mathcal{W}}_{\text{eff}}}{\partial u_a \partial u_b} \tag{B.8}$$

Both formulae are insightful. The first allows a direct relation to the JK-residue expression, while the second makes contact with the effective 2d KK theory. Here, $\widetilde{\mathcal{W}}_{\text{eff}}$ denotes the effective twisted superpotential and $\Omega$ is known as effective dilaton, which accounts for the coupling of the theory to the curved 3-manifold. $\mathcal{H}_{A/B}$ is referred to as the 3d handle-gluing operator. In both, the sum is over the Bethe roots

$$\mathcal{S}_{\text{BE}} = \left\{\boldsymbol{x} \,\middle|\, P_a(\boldsymbol{x}) = 1 \quad a = 1, \ldots, \text{rk}(G), \quad w(u) \neq u, w \in \mathcal{W}_G\right\} / \mathcal{W}_G$$

$$P_a(\boldsymbol{x}) = e^{\mathrm{i}B_a} = e^{2\pi \mathrm{i}\frac{\partial \mathcal{W}}{\partial u_a}} \tag{B.9}$$

For the cases relevant here, the condition that no Weyl reflection is leaving a Bethe root invariant can be recast into the condition that the Vandermonde is non-vanishing

$$\prod_{\alpha \in G} (1 - \boldsymbol{x}^\alpha) \neq 0. \tag{B.10}$$

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
