# Peer review of "Rational $Q$-systems, Higgsing and Mirror Symmetry"

_SciPost Physics_

## Round 1 · Referee Report · Anonymous (Referee 1) · 2022-9-21

Report

Referee report

The authors have studied the correspondence between the $Q$-systems for integrable spin chains of $A_{\ell-1}$ type and 3d $\mathcal{N}=4$ quiver gauge theories known as $T_{\rho}^{\sigma}[SU(N)]$. They establish a dictionary which connects some features in the gauge theory side (such as Coulomb and Higgs branch higgsings, and mirror symmetry) to correspondent operations in the $Q$-system side.

I regard this work as extremely interesting, and as an example of high level research in the field. At the practical level, the paper is also clear and well written, and I could not find typos. I have just some minor comments and questions I would like to ask to the authors.

1) It is claimed various times in the text that is it much faster to solve for the Bethe roots by using the $Q$-system instead of the Bethe ansatz equation. This is summarized in table 1. It is not clear to me if these values refer to the XXX or XXY model. Also, the first two entries of the table are numbers in the same order of magnitude, so I personally don't know if one can use those two data points to argue for the superiority of the rational Q-system. It would be good if the authors could expand a little this section.

2) In section $2.2$ it is assumed various times that some terms appearing in eq. $(2.23)$ do not vanish, in order to divide the expression by them and have them appear in denominator. I don't understand how strong this assumption is, or if anything can be said in the vanishing case.

3) In various places in the text it is claimed that the 3d mirror dual of $T_{\rho}^{\sigma}[G]$ is given by $T^{\rho}_{\sigma}[G^{\vee}]$. I believe there are subtleties with this statement, which were overlooked in the original papers on $T_{\rho}^{\sigma}[G]$ theories by Hanany et. al. In particular for $G$ orthogonal or symplectic, there exist consistent choices of $\rho$ and $\sigma$ such that $T_{\rho}^{\sigma}[G]$ is a bad theory (here bad, in the sense of Gaiotto-Witten). As an example one can realize $USp(4)-[SO(8)]$ in this way. Now, for bad theories the concept of mirror symmetry is not unique. The Coulomb branch is not a cone with a unique singular point at which the SCFT and its mirror live at low energy. In general there are various singular higher dimensional loci, and one can define a different 3d mirror for each one of them. I was thinking maybe the authors could add this comment in a footnote somewhere in section $7$. Anyways this comment is irrelevant for the rest of the paper as for $G$ special unitary, any choice of partitions leads to good theories.

Upon clarification of the small points above, I will be happy to recommend this paper for publication on Scipost. I further add some other personal questions I have, which don't necessarily need to be addressed. It is just out of my personal interest for the paper.

1) In the case $G$ is orthosymplectic, does anything unusual or interesting happen in the $Q$-system side, in the case in which $\rho$ and/or $\sigma$ are orbits which do not lie in the image of the Spaltestein map?

2) Also in the orthogonal case, can the $Q$-system distinguish cases in which $\rho$ or $\sigma$ is doubly even? (i.e. the red/blue orbits for $so(4)$).

3) Is there any understanding of the field theory operators (like monopole and mesons) in the $Q$-system side, and how they get exchanged by mirror symmetry?

4) By glueing various $T_{\rho}[SU(N)]$ theories (for trivial $\sigma$) one can produce trinions, or in general star shaped quivers which are mirrors of class-S theories. Is there any way to extend this Bethe/Gauge correspondence to such theories?

5) Mirror symmetry is realized as the $S$-transformation in an Hanany-Witten setup. One can however use a more generic element of the IIB duality group, and have a much larger set of dualities. Like for example doing $STS$ (where here $T$ is the $T$ generator of $SL(2,Z)$). This generates linear quivers with CS levels at the nodes, as now (p,q) branes with both p and q non-trivial are present in the HW cartoon. One can of course check (i.e. HS, etc) that this theory is equivalent to the original one. Can you implement this duality at the level of the $Q$-system?

Attachment

  • validity: high
  • significance: high
  • originality: high
  • clarity: top
  • formatting: perfect
  • grammar: perfect

Author:  Marcus Sperling  on 2022-10-21  [id 2940]

(in reply to Report 1 on 2022-09-21)

We thank the referee for the careful reading of the manuscript and the detailed comments.

Regarding the points raised, we have revised the draft as follows:

Q1: We have added clarification in the revision. The comparison performed here is for the XXZ spin chain with fixed value of q. For the XXX spin chain, the efficiency of the two methods has already been compared in arXiv:1608.06504.

Q2: The assumption that (2.23) is not vanishing is necessary in order to obtained the BAE from the rational Q-system. If this is not satisfied, undesired solutions may be generated by the Q-system. At present, we understand this constraint as necessary for deriving the BAE, but we do not have an interpretation of these conditions in terms of the Q-system.
We have commented in the revised version.

Q3: We thank the referee for pointing out this subtlety for groups G other than SU(n). We have added a comment to clarify the matter to readers in Section 7.1.

We appreciate the additional questions. These are indeed interesting thoughts and we hope to report on them elsewhere.

---

## Round 1 · Referee Report · Anonymous (Referee 2) · 2022-9-25

Report

The paper is devoted to the study of the rational Q-systems that are associated with BAE corresponding to certain A-type quivers. This is relevant for various reasons. First, as shown by explicit computations by the authors, solving the rational Q-system is way more efficient than solving directly the BAE. This is very important since BAE appear in many areas of mathematical and theoretical physics, such as the computation of twisted indices of 3d theories which is one application explicitly presented in the paper.

The BAE considered are related under Bethe/Gauge correspondece to certain 3d $\mathcal{N}=4$ quiver gauge theories known as $T^\sigma_\rho[SU(n)]$ theories. The authors also discuss some new aspects of this correspondence from the perspective of the rational Q-system. Among these, they show how the partitions $\sigma$ and $\rho$ equally specify both the 3d theory and the associated rational Q-system; how the Higgsing that trigger RG flows connecting different $T^\sigma_\rho[SU(n)]$ theories similarly relate the associated rational Q-systems; how mirror symmetry, which is a 3d IR duality relating pairs of $T^\sigma_\rho[SU(n)]$ theories with swapped partitions $\sigma$ and $\rho$, is realized in the rational Q-system.

The paper is well-written and contains a good review of the necessary background material, both for the rational Q-system and for the 3d $\mathcal{N}=4$ gauge theory analysis. Because of this as well as the importance of the topic and the originality of the results, I think that this paper is suitable for publication on SciPost Physics. I only have a couple of minor comments that the authors might address before publication:

  • It seems to me, especially reading section 5.2.2, that a Higgs branch Higgsing involving a single gauge node (what the authors call "$A_{M-1}$ transition") should always decrease the number of inhomogeneities in the associated BAE by 1. This is because the specialization of the Bethe root (5.18) should be accompained by the specialization (5.19) of one of the inhomogeneities. Nevertheless, comparing the BAE before and after the Higgsing, this seems to be the case only if the node is at one of the two extremities of the quiver. For example, looking at Figure 6, in the first two transitions the number of parameters descreases by 1, but at the last one it doesn't. More precisely, it seems that starting from the BAE of the third quiver and doing the Higgsing one would get the BAE of the fourth quiver but with the parameter $\theta^{(3)}_1$ specified in terms of $\theta^{(1)}_1$. How is the missing inhomogeneity restored in such a case? I suspect that this can be done via a shift of the Bethe roots, since the problem seems related to the fact that in the 3d gauge theory an overall $U(1)$ in the flavor symmetry can be re-absorbed by a gauge transformation, but it would be good if the authors could clarify this better.

  • The sentence "for the Coulomb branch Higgsing the “smoking gun” is the presence of balanced gauge nodes" at the second paragraph of section 5.3 on page 36 sounds weird, since as the authors also explain very clearly a Coulomb branch Higgsing doesn't preserve the balanceness of gauge nodes. Moreover, I believe at the very end of the first bullet point that appears soon after there is a typo, namely $e_{M-2}$ and $e_{M-1}$ should be $e_{r+M-2}$ and $e_{r+M-1}$.

  • The authors give the mapping of parameters (6.9) under mirror symmetry, were $y_i$ parametrize a $U(N)$ flavor symmetry so that the character of the fundamental representation is $\sum_{i=1}^N y_i$ and $\epsilon_i$ parametrize topological symmetries so that the ratio $\epsilon_i/\epsilon_{i+1}$ gives the FI of the $i$-th gauge node. Is there a simple way to unerstand why this is the precise mapping of masses and FI parameters?

  • validity: high
  • significance: high
  • originality: high
  • clarity: top
  • formatting: perfect
  • grammar: excellent

Author:  Marcus Sperling  on 2022-10-21  [id 2941]

(in reply to Report 2 on 2022-09-25)

We thank the referee for the careful reading of the manuscript and the detailed comments.

Regarding the points raised, we have revised the draft as follows:

Q1: We thank the referee for posing this question! We have added clarifications in Sec 5.2.1 and 5.4.1 and App. A.3. In brief, the Q-system perceives Higgsing via the change in partition data; thus, one does not require the tuning of parameters as in the BAE. In terms of the BAE, the parameters after Higgsing can be suitably identified as demonstrated in App A. These are in principle non-generic parameters. However, if one aims to re-derive the BAE from the rational Q-system, then the parameters need to be tuned accordingly.
More conceptually, it is not yet understood how the change of partition data is equivalent to the tuning of parameters purely on the level of rational Q-system.

Q2: The statement is correct, though it may be have been written in a confusing way. Balance indicates a set of NS branes with equal linking numbers such that Coulomb Branch moduli can be opened up. This is analogous to flavours nodes in HB Higgsing: non-abelian flavour symmetries are a “smoking gun” for Higgs Branch Higgsing, even though flavour nodes are not preserved during Higgs Branch Higgsing. We have rephrased and clarified the sentence.

We thank the referee for pointing out the typo!

Q3: We added further clarifications in the revised version. The exchange of y and \epsilon upon mirror symmetry is a consequence of exchanging D5 and NS5 branes. The behaviour of q follows from recalling the definition of the U(1) global symmetry that implies $N=2^\ast$ real mass parameter. This U(1) is generated by the difference of the maximal tori of the Higgs and Coulomb branch SU(2) R-symmetry. Since mirror symmetry includes the automorphism of the N=4 algebra, the R-symmetries are exchanged and the mass parameter changes sign.

---

## Editorial Decision

resubmitted